# Learning Treatment Representations for Downstream Instrumental Variable Regression

**Shiangyi Lin** [1]   **Hui Lan** [1]   **Vasilis Syrgkanis** [2] [3]

## Abstract

Traditional instrumental variable (IV) estimators cannot accommodate more treatments than instruments, a limitation that is critical for high - dimensional, unstructured data like clinical treatment pathways. Current practice—applying unsupervised dimension reduction before IV estimation — suffers from substantial omitted treatment bias because the representation learning step ignores the instrument. We propose a novel framework that constructs treatment representations by explicitly incorporating instrumental variables. We prove that this instrument-guided approach ensures the identification of optimal outcome-prediction directions even with limited instruments. Validation on large-scale, semi-synthetic clinical data derived from a major hospital, along with other simulations, shows that our approach significantly outperforms conventional two-stage methods.

## 1. Introduction

Instrumental variable (IV) methods are a cornerstone of causal inference, enabling the estimation of causal effects in the presence of unobserved confounding (Angrist et al., 1996). However, a fundamental limitation of classical IV estimators is the *order condition* for identification, which requires that the number of instruments be at least equal to the number of endogenous treatments (Murray, 2006). This requirement poses a significant challenge in many modern applications where treatments are high-dimensional and complex—such as images, text, or detailed patient histories—while the number of available instruments remains small.

Consider, for instance, the problem of estimating the causal effect of various treatment plans on patient outcomes. Treatments are high-dimensional bundles of procedures, medications, and services, often recorded as thousands of billing codes. These treatment choices are heavily confounded by unobserved factors, such as the severity of the patient's condition or individual physician practices. While valid instruments exist, such as differential distance to hospitals or variation in hospital admission patterns by day of the week (Dong et al., 2019; Dong et al.; Qin et al., 2023), they are typically low-dimensional. With more treatment dimensions than instruments, the causal effect is not identified.

A common practice is to first apply an unsupervised dimension reduction, such as Principal Component Analysis (PCA) or an autoencoder (AE), to compress the high-dimensional treatment $X$ into a low-dimensional representation $D$. Then, a standard IV method like two-stage least squares (2SLS) is performed on $D$. However, this two-step approach is fundamentally flawed. Because the representation is learned without knowledge of the instrument $Z$, it is not guaranteed to preserve the instrument-relevant variation in the treatment. The learned representation $D$ may capture dimensions of high variance in $X$ that are unrelated to $Z$, while discarding the components of $X$ driven by the instrument. This can introduce omitted treatment bias, violating the exclusion restriction and invalidating the IV estimates (see Figure 1b). While modern non-parametric IV methods can handle high-dimensional treatments (e.g. DeepIV (Hartford et al., 2017)), as we discuss in the related work, they are still plagued by non-parametric analogues of the order condition to argue the true model is identifiable.

To address this challenge, we introduce *Instrument-Guided Representation Learning (IGRL)*, a framework for learning low-dimensional treatment representations that are explicitly optimized for downstream IV analysis. IGRL integrates instruments directly into the representation learning process. By regularizing the learner to capture the components of the treatment $X$ that are driven by the instrument $Z$, our method ensures that the learned representation $D$ satisfies the necessary conditions for valid IV regression. We provide theoretical guarantees for the identification of valid intervention directions and demonstrate empirically that IGRL

[1]Institute of Computational and Mathematical Engineering, Stanford University, Stanford, CA, USA [2]Management Science and Engineering, Stanford University, Stanford, CA, USA [3]Supported by NSF Award IIS-2337916. Correspondence to: Shiangyi Lin <shiangyi@stanford.edu>.

*Proceedings of the 43rd International Conference on Machine Learning*, Seoul, South Korea. PMLR 306, 2026. Copyright 2026 by the author(s).

outperforms traditional two-step approaches that separate dimension reduction from causal estimation. We validate our framework on a large-scale, semi-synthetic clinical dataset derived from real-world records of a major hospital and a nonlinear MNIST-based benchmark in Section 5.

**Related Work** Our work is situated at the intersection of causal inference and representation learning. We build upon several lines of research, which we briefly summarize here. Further discussion is available in Appendix A.

**Causal Representation Learning.** A growing body of literature focuses on learning representations that uncover the underlying causal structure of the data generating process (Schölkopf et al., 2021). Much of this work has focused on the identifiability of latent causal variables under various assumptions, often in the context of independent component analysis (ICA) or variational autoencoders (VAEs) (Hyvärinen, 2013; Hälvä and Hyvarinen, 2020; Monti et al., 2020; Ahuja et al., 2022; Hyvärinen et al., 2023; Jin and Syrgkanis, 2023; Hyvärinen et al., 2024; Hälvä et al., 2024). While related, the primary goal of this line of work is causal discovery, i.e., identifying the causal graph. Our objective is different: we aim to learn representations specifically for *downstream IV regression*.

**Representation Learning for Causal Inference.** Several recent works have explored using representation learning to handle high-dimensional variables in causal inference, but most do not consider unobserved confounders. For instance, Nabi et al. and Andreu et al. learn representations of high-dimensional treatments, but they assume all confounders are observed (2022, 2024). A separate line of work addresses the issue of omitted variable bias when learning representations of high-dimensional *confounders* (Vafa et al., 2024; Melnychuk et al., 2025). These methods aim to ensure that the learned representation of observed confounders is sufficient to control for confounding. Our work is distinct in that we focus on learning representations for a high-dimensional *treatment* in a setting with unobserved confounders, for which we leverage instrumental variables.

**Instrumental Variables in Linear Settings and Factor Models.** There is a classical literature on combining IV with dimension reduction, mostly in linear settings. Rao and Sabatier et al. proposed methods for PCA with respect to instrumental variables (1964,1989). More recently, Kelly et al. and Wang use instruments to help identify latent factor models (2020,2024). These methods, however, are primarily designed for linear models and do not directly address the challenge of learning nonlinear representations for causal effect estimation.

**Connections to iVAE and Rep4Ex.** Our framework has technical similarities to the identifiable VAE (iVAE) (Khemakhem et al., 2020), where the instrument $Z$ can be seen as auxiliary information that aids disentanglement. However, a key difference is that iVAE would require $Z$ as an input to the encoder, which is not feasible in our setting as we aim to learn an encoding of the treatment that can be used for intervention when $Z$ is not available. Our setup is also related to Rep4Ex (Saengkyongam et al., 2023), which learns representations for intervention extrapolation. However, Rep4Ex focuses on interventions on the auxiliary variable itself, whereas we focus on identifying directions of intervention in the latent treatment space. Furthermore, our structural model is more general, allowing for noisy and multi-component variation in the treatment.

**Machine Learning for Non-Parametric IV.** Recent ML-based IV methods (Hartford et al., 2017; Singh et al., 2019; Muandet et al., 2020b; Dikkala et al., 2020; Bennett et al., 2023a; Xu et al., 2021; Kim et al., 2025) handle high-dimensional treatments but require modeling the full conditional $P(X|Z)$ or solving complex minimax objectives. Because these methods model the effect $h(X)$ directly on the high-dimensional space, theoretical guarantees typically requires the instrument to be as "rich" as the treatment implicitly (e.g., via completeness in the high-dimensional $X$-space). In contrast, our approach identifies improving interventions by applying IV methods to a learned low-dimensional representation $D$ (which suffices for identifying improving interventions), requiring only completeness in the lower-dimensional representation space. Our contribution is orthogonal to these estimators: IGRL can be viewed as a preprocessing step that reduces dimensionality to ensure the resulting causal model on the learned representation is identifiable.

## 2. Problem Statement: Learning Interventions via Representations

We consider a setting where we are given data containing samples of variables $(Z, X, Y)$, where $X$ is a high-dimensional "treatment" variable, $Y$ is a scalar outcome of interest, and $Z$ is a low-dimensional vector of instruments. The treatment $X$ is confounded by an unobserved variable $U$ that has a causal influence on the value of $X$ and also on $Y$, as depicted in Figure 1a. Our goal is to learn a latent representation, $D$, of the highly confounded, high-dimensional treatment that preserves the validity of instrumental variable analysis, allowing us to identify outcome-improving directions of intervention in the representation space, and subsequently in the original treatment space.

**A Practical Example – Finding the Ideal Hospital Pathway.** In the context of the clinical example described in Section 1: $X$ is the observed treatment plan represented in the form of a sequence of billing items. $D$ is the low-dimensional, instrument-driven latent treatment decision (e.g., aggressive surgical approach vs. conservative management), which is influenced by the instrument $Z$. $Y$ is the patient outcome (e.g. survival, length of stay).

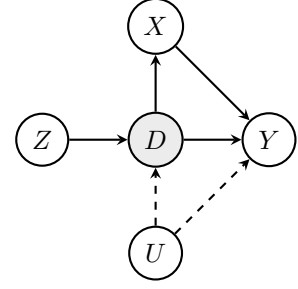

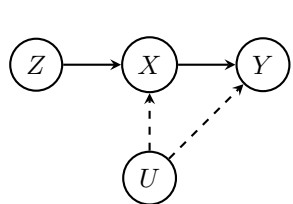

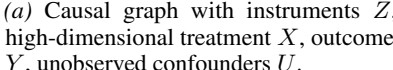

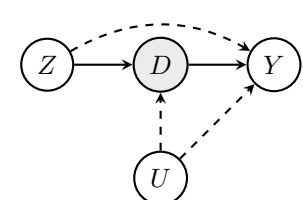

*(a)* Causal graph with instruments $Z$, high-dimensional treatment $X$, outcome $Y$, unobserved confounders $U$.

*(b)* Causal graph when high-dimensional treatment $X$ is replaced by learned representation $D$.

*(c)* Causal graph that an ideal representation $D$ of the high-dimensional treatment $X$ would satisfy.

*Figure 1.* Omitted treatment bias in instrumental variable analysis with learned treatment representations.

Naive representation learning approaches run the risk of an omitted-variable problem that can invalidate the downstream causal analysis. In particular, IV analysis assumes that the instrument $Z$, the treatment $X$, and the outcome $Y$ adhere to the causal graph depicted in Figure 1a. The instrument $Z$ only affects outcome $Y$ through its effect on treatment $X$. When the high-dimensional treatment $X$ is replaced by a learned representation $D$, we risk that the part of $X$ not captured in $D$ contains elements that are correlated with both the instrument $Z$ and the outcome $Y$. As a result, $D$ no longer absorbs the entire effect of the instrumental variable $Z$ on the outcome $Y$. This creates causal pathways from the instrument $Z$ to the outcome $Y$ that do not flow through the representation $D$, as shown in Figure 1b. For example, simply running PCA will produce representations that capture the dimensions of largest variance, rather than those influenced by instruments. Therefore, we need to regularize the representation learning process to ensure that the causal influence through these omitted paths is minimal.

An ideal latent representation $D$ should satisfy the causal graph depicted in Figure 1c, where the instrument $Z$ has no causal effect on $X$ that is not mediated by the latent representation $D$. If the representation encodes all outcome-relevant information, then no direct edge from $X$ to $Y$ should exist. However, the existence of such an edge does not invalidate the downstream instrumental variable analysis, and hence, it is not essential to exclude it.

**Structural Equation Model.** To formalize our problem, we will consider the following data generating process (structural causal model) [1] for the observed random variables in our study:

$$
\begin{aligned}
D &= A \cdot Z + U, & U &\perp\!\!\!\perp Z \\
X &= f(D, V), & V &\perp\!\!\!\perp Z \quad\quad (1) \\
Y &= h(D) + \eta(U, V, \epsilon), & \epsilon &\perp\!\!\!\perp Z
\end{aligned}
$$

The random variables $U, V, D, \epsilon$ are latent. $A$ is an $r \times k$ matrix that captures the effect of the instruments $Z \in \mathbb{R}^k$

on latent decision vector $D \in \mathbb{R}^r$. $U$ represents the unobserved confounder affecting the elements of the treatment that are also influenced by the instrument. $\epsilon$ represents an outcome noise variable and is allowed to be correlated with $U, V$. $D$ represents the aspects of the treatment $X$ affected by the instruments, while $V$ represents the remaining aspects that describe the treatment $X$, but are independent of the instrument. In the context of our *Hospital Pathway Example*, $V$ represents baseline patient covariates, such as frailty or comorbidities, while $U$ denotes unobserved confounders such as acute disease severity that simultaneously influence the treatment strategy and the outcome. The term $\epsilon$ accounts for stochastic noise in the observed outcome. For notational convenience, we will assume that $\mathbb{E}[U] = \mathbb{E}[V] = \mathbb{E}[\eta(U, V, \epsilon)] = 0$.[2] Moreover, we assume that the encoding/decoding between the latent representations and the observed treatment is invertible:

**Assumption 2.1** (Invertible Encoding). *The function $f$ is invertible. Let $e(X) = f^{-1}(X) = (D, V)$ denote the encoding function, i.e., there is a one-to-one correspondence between the high-dimensional treatment $X$ and the characteristics $(D, V)$ that describe the treatment.*

Next, we also assume that the transformation from the instruments to the latent representation $D$ is full rank.

**Assumption 2.2** (Full-Rank Latents). *The matrix $A$ has full row-rank and $\mathbb{E}[ZZ^\top] \succ 0$.*

**Discussion of Assumptions.** Our Assumption 2.1 is no stronger than those found in previous work. While previous work typically assume that the mapping from $D$ to $X$ is injective (Khemakhem et al., 2020; Saengkyongam et al., 2023), we relax this requirement by introducing a noise term $V$, assuming instead that $X = f(D, V)$ is bijective. From this perspective, $(D, V)$ can be viewed as a nonlinear decomposition of the treatment into instrument-dependent and instrument-independent components. We denote the encodings of the treatment by $e_D(X) = D$ and $e_V(X) =$

---

[1] The inclusion of covariates is discussed in Appendix D.

[2] Appropriate intercept constants need to be added to the equations in the absence of this convention.

$V$, which return these respective components.

Furthermore, requiring the matrix $A$ to have full row rank in Assumption 2.2 is equivalent to requiring $\text{rank}(\text{Cov}(D, Z)) = \dim(D)$. This condition ensures that every direction in the treatment space responds to perturbations in the instrument, which is a natural extension of the *relevance condition* to the setting of high-dimensional, continuous treatments and instruments. The last condition $\mathbb{E}[ZZ^\top] \succ 0$ can always be satisfied via a preprocessing step, such as applying a PCA transformation to the instruments to remove collinear or nearly collinear variables.

**Learning Good Interventions via Representations.** Given observations $(Z, X, Y)$ from the structural equation model, our goal is to learn an incremental intervention mapping $t(X)$ such that the expected outcome under intervention is larger than the original outcome. We use $Y^{(X \leftarrow x)}$ to denote the random outcome where we fix the treatment $X$ to be $x$. Thus we are searching for a incremental intervention $t(X)$ such that:

$$\mathbb{E}\left[Y^{(X \leftarrow t(X))}\right] > \mathbb{E}[Y] \tag{2}$$

Note that due to the one-to-one correspondence of $X$ with its decomposition, any such interventional outcome can equivalently be viewed as an intervention on the latent components of the treatment, i.e., $Y^{(D \leftarrow e_D(x), V \leftarrow e_V(x))}$. Given the structural Equation (1), the expected outcome under a incremental intervention $t(X)$ can be written as:

$$\mathbb{E}\left[Y^{(X \leftarrow t(X))}\right] = \mathbb{E}[h(e_D(t(X))) + \eta(U, e_V(t(X)), \epsilon)]$$

We identify such an intervention by acting on a learned representation. In particular, given observations, we will learn an encoding $\tilde{e}_D(X) = \tilde{D}$ that respects the properties in Equation (1) (potentially together with a learned encoding $\tilde{e}_V(X) = \tilde{V}$) and a corresponding decoder $\tilde{f}(\tilde{D})$ (potentially also taking as input $\tilde{V}$) that maps the learned encoding back to the a high-dimensional treatment. Subsequently, we will estimate an outcome-improving direction $u$ in the learned representation space via instrumental variable analysis, viewing $\tilde{D}$ as the "treatment" and $Z$ as the instrument. We will apply the direction $u$ to the learned representations, i.e., $\tilde{D} + \alpha u$, for some scalar intervention amount $\alpha$. For ease of notation, we denote by $(\cdot)_{\alpha u}$ the corresponding random variable $(\cdot)$ after this intervention. We then decode back to the high-dimensional treatment space $X_{\alpha u} = \tilde{f}(\tilde{D} + \alpha u)$ ($X_{\alpha u} = \tilde{f}(\tilde{D} + \alpha u, \tilde{V})$ if an encoding of $V$ was also learned). This process (depicted also visually in Figure 2 and described algorithmically in Algorithm 1) defines our incremental-intervention mapping:

$$t(X) = \tilde{f}(\tilde{e}_D(X) + \alpha u, \tilde{e}_V(X)), \tag{3}$$

where the second input is omitted if $\tilde{e}_V$ is not learned.

The advantage of learning encoding and decoding function explicitly is to translate any perturbation in learned latent space into human-interpretable original observed space.

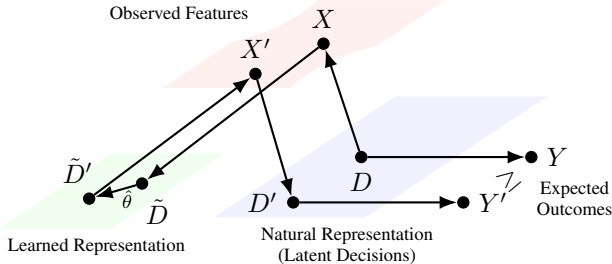

*Figure 2.* Intervention on learned representation.

---

**Algorithm 1** Intervention in Latent Representation Space and evaluation

---

1: **Autoencoder fitting.** Learn encoder $\tilde{e}$ and decoder $\tilde{f}$ of $X$ and using observed data $(Z, X, Y)$.
2: **IV analysis.** Identify causal model $\tilde{h}(\tilde{D})$ using IV regression analysis with instrument $Z$, treatment $\tilde{D} \triangleq \tilde{e}_D(X)$ and outcome $Y$. Calculate average causal derivative $u = \mathbb{E}[\nabla_D \tilde{h}(\tilde{D})]$.
3: **Encode.** Transform $X$ into latent representation $\tilde{D}$ using learned encoder $\tilde{D} = \tilde{e}_D(X)$
4: **Perturb.** Apply perturbation in the latent space: $\tilde{D}_{\alpha u} = \tilde{D} + \alpha u$ where $\alpha$ is a scalar factor controlling perturbation magnitude.
5: **Decode.** Map perturbed latent representation $\tilde{D}_{\alpha u}$ back to input space: $X_{\alpha u} = \tilde{f}(\tilde{D}_{\alpha u})$ (or $X_{\alpha u} = \tilde{f}(\tilde{D}_{\alpha u}, \tilde{e}_V(X))$ if the learned encoder also learns a representation of $V$).
6: **Evaluate.** Apply the true decomposition $e(X_{\alpha u}) = (D_{\alpha u}, V_{\alpha u})$ and evaluate outcome under intervention: $Y_{\alpha u} = h(D_{\alpha u}) + \eta(U, V_{\alpha u}, \epsilon)$.
7: Compare average original outcome $Y$ to average perturbed outcome $Y_{\alpha u}$.

---

## 3. Instrument Guided Representation Learning: The Linear Setting

We first consider the case where the structural equation model associated with the causal graph in Figure 1c contains only linear relationships:

$$\begin{aligned} D &= A \cdot Z + U, & U \perp\!\!\!\perp Z \\ X &= B \cdot D + B_\perp \cdot V, & V \perp\!\!\!\perp Z \\ Y &= \theta^\top D + \eta(U, V, \epsilon), & \epsilon \perp\!\!\!\perp Z \end{aligned} \tag{4}$$

Here, $B$ is an $m \times r$ matrix mapping the $r$-dimensional instrument-driven latent decision $D$ to the observed high-dimensional treatments $X \in \mathbb{R}^m$, and is assumed to be *full*

*column rank.* $B_\perp$ is a matrix whose column space is orthogonal to the column space of $B$ and is also assumed to be *full column rank.* $U$ corresponds to a random vector of unobserved confounders that also affect decisions and outcomes. $\theta$ is an $r$-dimensional vector capturing the direct effects of the latent decisions on the outcome. We will assume that the matrix $A$ is of *full row rank*, i.e., there are more instruments $Z$ than latent decisions $D$, and the instruments vary these latent dimensions in a full-rank manner.

Note that this setting falls under our general model since the function $f(D, V) = BD + B_\perp V$ is invertible. In particular, by the orthogonality of the column space of the two matrices and the fact that they are both full column rank, we have:

$$e_D(X) \triangleq B^+ X = D \qquad e_V(X) \triangleq B_\perp^+ X = V \quad (5)$$

where $B^+$ denotes the Moore-Penrose pseudoinverse of a matrix and is a left inverse for full column rank matrices, i.e. $B^+ = (B^\top B)^{-1} B^\top$.[3]

Our target quantity of interest is the overall effect $\theta$ of the latent factors $D$ on the outcome $Y$. If we could identify the latent factors $D, V$ from the observed variables, then the intervention direction could be taken as $u = \theta/\|\theta\|$. In this case, our improving intervention corresponds to $t(X) = B(e_D(X) + \alpha u) + B_\perp V = X + \alpha B u$, with $D_{\alpha u} = D + \alpha u$ and $V_{\alpha u} = V$, which would lead to an interventional outcome of $Y_{\alpha u} = \theta^\top(D + \alpha u) + \eta(V, U, \epsilon)$, hence:

$$\mathbb{E}[Y_{\alpha u}] = \mathbb{E}[Y] + \alpha \theta^\top u = \mathbb{E}[Y] + \alpha\|\theta\| \quad (6)$$

Here, to perform the intervention, it suffices to learn a linear encoder $e_D(X) = B^+ X$, since this allows the intervention $t(X) = X + \alpha B u$ to be performed without learning an encoding for $V$. We will take this approach in the remainder of this section.

Linear regression of $X$ on $Z$ uncovers the matrix $C = B \cdot A$, following the structural equation:

$$X = B \cdot A \cdot Z + \underbrace{B_\perp \cdot V + B \cdot U}_{\zeta}, \quad \mathbb{E}[\zeta \mid Z] = 0. \quad (7)$$

We can therefore recover this causal subspace via a thin Singular Value Decomposition (SVD) of the estimated coefficients, $C = \mathcal{U}\Sigma\mathcal{V}^\top$. Then, setting our representation to $\tilde{D} = \mathcal{U}^\top X$, we effectively isolate the instrument-relevant variation. Moreover, we show that $\tilde{D}$ is an invertible transformation of the true latent $D$ under Assumptions 2.2 and Equation 4. the proof is deferred to Appendix B.

Once the representation $\tilde{D}$ is obtained, we can obtain an instrumental variable regression estimate solving the solution

---

[3]We could have equivalently defined the SEM $X$ as:
$$X = B \cdot D + V, \qquad V \perp\!\!\!\perp Z$$
by splitting the second part into $B \cdot V + B_\perp V$ and redefine $D \to D + V$, or equivalently redefine $U \to U + V$.

to the moment restriction: $\mathbb{E}[Z(Y - \tilde{\theta}^\top \tilde{D})] = 0$. Under Assumptions 2.2 and the result above, the moment restriction has a unique solution, $\tilde{\theta} = (P^{-1})^\top \theta$. This serves as the outcome-improving intervention direction. The following theorem formalizes these arguments and provides the outcome improvement guarantee for this intervention.

**Theorem 3.1.** *Under the linear structural equation model in Equation* (4) *and assuming* $B, B_\perp$ *have full column rank and Assumption 2.2 holds, then the representation and intervention produced by the LIRR algorithm satisfy:* $\tilde{D} = PD$, *for the invertible matrix* $P \triangleq \hat{B}^\top B$. *Moreover,* $\tilde{\theta} = (P^{-1})^\top \theta$ *and the interventional outcome satisfies the guaranteed improvement property:*

$$\mathbb{E}[Y_{\alpha u}] = \mathbb{E}[Y] + \alpha\|(P^{-1})^\top \theta\|$$

*The proof is deferred to Appendix B.*

The LIRR algorithm offers improvements over standard dimensionality reduction approaches in settings characterized by large non-Gaussian and correlated noise terms, as demonstrated in Appendix G. We also note that the solution $\hat{B}\hat{\theta}$ recovered by LIRR coincides with the minimum-norm solution to the linear moment condition with $X$ as the treatment. Consequently, our approach aligns intuitively with standard linear IV, where the explicit latent learning step simply offers greater interpretability.

---

**Algorithm 2** Linear Instrument Regularized Representation (LIRR) and Intervention

---
1: **Input:** magnitude of intervention $\alpha$
2: Run linear regression of $X$ on $Z \in \mathbb{R}^k$, to estimate a coefficient matrix $C$
3: Calculate the *thin* SVD decomposition of $C = \mathcal{U}\Sigma\mathcal{V}^\top$, keeping only the top $k$ singular values
4: Define $\hat{B} = \mathcal{U}$ and $\hat{A} = \Sigma\mathcal{V}^\top$ and $\tilde{D} = \tilde{e}_D(X) = \hat{B}^\top X$
5: Run linear IV regression solving moment $\mathbb{E}[Z(Y - \tilde{\theta}^\top \tilde{D})] = 0$
6: Let $u = \tilde{\theta}/\|\tilde{\theta}\|$ and perform intervention on learned representation space $\tilde{D}_{\alpha u} = \tilde{D} + \alpha u$
7: Encode back to X-space intervention of $X_{\alpha u} = X + \alpha \hat{B} u$

---

# 4. Instrument Guided Representation Learning: The Nonlinear Setting

We now investigate the general setting introduced in Equation (1). In this nonlinear setting, we require additional assumptions. We assume that the instrument-driven latent component $D$ to be independent of the instrument-independent latent component $V$. In particular, we assume the slightly stronger property of joint independence among

$Z, U, V$, which implies that $D \perp\!\!\!\perp V$ [4].

**Assumption 4.1** (Joint Independence). Assume that $Z \perp\!\!\!\perp U \perp\!\!\!\perp V$ (jointly independent).

Moreover, we assume the regularity conditions that the mixing function $f$ is differentiable and that the instrument is supported on an open subset of $\mathbb{R}^k$.

**Assumption 4.2** (Differentiable Decoding Function). $f$ is a differentiable function with uniformly bounded derivatives.

**Assumption 4.3.** $\mathbb{E}[Z] = 0$ and the support of $Z$, $\mathcal{Z}$, is an open subset of $\mathbb{R}^k$.

We make a completeness assumption on the strength of the instrument, which is a standard assumption for non-parametric instrumental variable identification (Cui et al., 2024). We discuss sufficient conditions in the Appendix C.4. In particular, it involves characteristic function assumptions that have also been typical in the identifiable latent factor literature (Lu et al., 2021).

**Assumption 4.4** (Bounded Completeness). $D$ is bounded complete for Z, that is, for all bounded real functions $h$, we have that:

$$\mathbb{E}[h(D)|Z] = 0 \quad \text{a.s.} \quad \Rightarrow \quad h(D) = 0 \quad \text{a.s.}$$

**Theorem 4.5.** *Suppose that the data generating process follows the SEM described in Equation 1, and satisfies Assumptions 2.1 & 2.2 & 4.1 & 4.2 & 4.3 & 4.4. Let $(\tilde{D}, \tilde{V}) := (\tilde{e}_D(X), \tilde{e}_V(X)) = \tilde{e}(X)$ denote the learned representations. Consider encoder-decoder pairs with perfect reconstruction, i.e. $X = \tilde{f} \circ \tilde{e}(X)$. Then, for the solution $\tilde{e}, \tilde{f}$, and full row rank matrix $\tilde{A}$ that minimizes:*

$$\mathbb{E}[\|\tilde{e}_D(X) - \tilde{A}Z\|^2] \tag{8}$$

*subject to the following constraints: 1) $\tilde{e}$ is a differentiable function with uniformly bounded derivatives. 2) $\tilde{A}$ has full row rank. 3) $\tilde{D} = \tilde{A}Z + \tilde{U}$ with $\tilde{U} \perp\!\!\!\perp Z$ and $\mathbb{E}[\tilde{U}] = 0$.*

*we have that, with probability 1, $\tilde{D} = PD$ and $\tilde{U} = PU$, for $P = \tilde{A}A^+$. Moreover, the matrix $P$ is invertible.*

*The proof is deferred to Appendix C.1.*

Assumptions $\mathbb{E}[Z] = 0$ and $\mathbb{E}[ZZ'] \succ 0$ are without loss of generality as we can always preprocess $Z$ by centering it and removing collinear instruments. Moreover, in practice the assumption that $\tilde{D} = \tilde{A}Z + \tilde{U}$, with $\tilde{U} \perp\!\!\!\perp Z$ and $\mathbb{E}[\tilde{U}] = 0$ can be achieved by minimizing a square loss with an intercept, i.e.

$$\min_{e, f, A, c : e, f \text{ invertible}, e \circ f = \text{identity}} \mathbb{E}[\|e_D(X) - AZ - c\|^2]$$

---

[4]The joint independence assumption can be relaxed by including observed covariates (see Appendix D).

and then defining $\tilde{D} = \tilde{e}_D(X) \triangleq e_D(X) - c, \tilde{f} = f + c$.

Subsequently, we identify an intervention as described in Algorithm 1. In particular, we run an IV analysis, with $Z$ as the instrument, $\tilde{D}$ as the treatment, and $Y$ as the outcome, to estimate a causal model in representation space by finding a solution to the conditional moment restrictions:

$$\mathbb{E}[Y - \tilde{h}(\tilde{D}) \mid Z] = 0 \tag{9}$$

Since $\tilde{D} = PD$ and $\mathbb{E}[Y \mid Z] = \mathbb{E}[h(D) \mid Z]$, by the completeness assumption,

$$\mathbb{E}[h(D) - \tilde{h}(PD) \mid Z] = 0 \Rightarrow h(D) = \tilde{h}(PD) \text{ a.s.}$$
$$\implies h(P^{-1}\tilde{D}) = \tilde{h}(\tilde{D}) \text{ a.s.}$$

If for instance, $h$ is assumed to be linear, then $\tilde{h}$ is also a linear function and it suffices to run a linear instrumental variable analysis (e.g. two-stage-least-squares). If $h$ is nonlinear, then we calculate the average derivative of $\tilde{h}$, and perform the intervention $u$ as described in Algorithm 1. i.e.

$$\tilde{\theta} = \mathbb{E}[\nabla_{\tilde{D}}\tilde{h}(\tilde{D})] = (P^{-1})^\top \mathbb{E}[\nabla_D h(D)], \quad u = \tilde{\theta}/\|\tilde{\theta}\|$$

In finite samples, recently introduced doubly robust methods for estimation of average derivatives of solutions to non-parametric IV problems can be used (Bennett et al., 2022; 2023b).

**Theorem 4.6.** *Assume that:*

$$Y = h(D) + \eta(U, V, \epsilon), \quad \epsilon \perp\!\!\!\perp \{Z, U, V\}$$

*and that $h$ is twice differentiable with a bounded second derivative. Let $\tilde{e}, \tilde{f}, \tilde{A}$ be an optimal solution that minimize the objective in Equation 8, satisfying the assumptions of Theorem 4.5 with the extra constraint that: 1) $\tilde{D} \perp\!\!\!\perp \tilde{V}$. 2) $\tilde{U} \perp\!\!\!\perp \tilde{V} \perp\!\!\!\perp Z$ 3) $\tilde{e}$ is an invertible function when restricted to inputs in the image of $f$ and $\tilde{f} \circ \tilde{e}(x) = x$ for all $x \in Im(f)$. Furthermore, assume that the variable $D$ has full support in $\mathbb{R}^r$, i.e. $\mathcal{D} = \mathbb{R}^r$. Then setting $u = \tilde{\theta}/\|\tilde{\theta}\|$, in Algorithm 1, with $\tilde{\theta} = \mathbb{E}[\nabla_{\tilde{D}}\tilde{h}(\tilde{D})]$ and $\tilde{h}$ the solution to the conditional moment restriction problem in Equation (14), we have that:*

$$\mathbb{E}[Y_{\alpha u} - Y] = \alpha\|(P^{-1})^\top \mathbb{E}[\nabla_D h(D)]\| + O(\alpha^2)$$

*The proof is deferred to Appendix C.2.*

Hence, for small enough step size $\alpha$, the identified intervention will achieve a positive improvement on the outcome (assuming that $\mathbb{E}[\nabla_D h(D)] \neq 0$).

**Instrument Regularized Auto-Encoder** To achieve the positive improvement as described in Theorem 4.6, then we need to incorporate loss components that are minimized only when i) $e, f$ reconstruct the input $X$, ii) $e_D(X)$ is predicted

linearly by $Z$ with a full rank matrix $A$, iii) the residual of this regression $e_D(X) - AZ - c$, which approximates $U$, needs to be independent of $Z$, iv) $Z$ needs to be independent of $e_V(X)$ and v) $e_D(X)$ needs to be independent of $e_V(X)$, $e_D(X) - AZ - c$ needs to be independent of $e_V(X)$, and the variables $(e_D(X) - AZ - c, e_V(X), Z)$ are jointly independent. While we do not explicitly enforce $\tilde{A}$ to be full row rank, we expect this to be satisfied due to the reconstruction loss and the condition that $\tilde{D} \perp\!\!\!\perp \tilde{V}$. Note that in addition to joint independence, we also explicitly enforce pairwise independencies for computational reasons.

Denote the residual as $\delta := e_D(X) - AZ - c$. We introduce the instrument-regularized auto-encoder loss, which incorporates all these elements. Let $\mathcal{R}(A, B)$ or $\mathcal{R}(A, B, C)$, denotes any regularizer that can be evaluated on a set of $n$ samples and which takes small values when the random variables $A, B$ or $A, B, C$ are jointly independent. In our experiments, we use the kernel-based independence test statistic (HSIC) (Gretton et al., 2007) and its generalization to joint independence (d-HSIC) (Pfister et al., 2018).

$$
\begin{aligned}
\min_{e, f, A, c} \; & \mathbb{E}\left[\|X - f \circ e(X)\|^2\right] + \lambda \mathbb{E}\left[\|\delta\|^2\right] \\
& + \mu_1 \mathcal{R}(\delta, Z) + \mu_2 \mathcal{R}(Z, e_V(X)) \\
& + \mu_3 \big( c_1 \mathcal{R}(e_D(X), e_V(X)) + c_2 \mathcal{R}(\delta, e_V(X)) \\
& \qquad + c_3 \mathcal{R}(\delta, Z, e_V(X)) \big) \qquad \text{(IRAE)}
\end{aligned}
$$

In the following experiments, we denote IRAE[0] as the variant that contains only the regularizers weighted by $\lambda$, IRAE[1] as the variant that contains the parts weighted by $\lambda, \mu_1$, IRAE[2] as the variant that contains the parts weighted by $\lambda, \mu_1, \mu_2$, and IRAE as the variant that contains all regularizers.

## 5. Experimental Evaluation

We defer a comprehensive set of synthetic benchmarks—comparing our proposed framework against performing dimension reduction by PCA, Vanilla AE, Outcome-guided AE, iVAE followed by 2SLS, and fitting DeepIV directly under conditions that both strictly satisfy to Appendices G, H. These controlled experiments show that while standard representation-learning pipelines (such as PCA or Vanilla AE) perform competitively when the latent confounding noise and observed feature noise components ($U$ and $V$) are negligible, they fail when the noise are large or structured. In contrast, our approach (via LIRR and IRAE variants) maintains robust performance, matching or often exceed the performance of DeepIV while providing a interpretable lower-dimensional latent representation . In the following section, we focus on two practical applications.

**Pneumonia Recovery.** We evaluate our method on a semi-synthetic dataset ($N = 50,000$) derived from 7,200 pneu-

monia patients at a major hospital. Variables are sampled from multivariable guassian fitted on empirical mean and covariate. The instrument ($Z$) represents physician assignment (dim $= 475$), driven by shift timing rather than patient complexity. Confounders ($U$) are Elixhauser Comorbidity indicators. For treatment ($X$), we aggregate EHR codes into 815 categories based on $p(X|Z)$ with noise. The outcome ($Y$), recovery time, is generated as:

$$
\begin{aligned}
Y = \max \big( 0, \\
48 + \gamma \sum w_j^{\text{sickness}} U_j + \sum w_k^{\text{treatment}} X_k + \epsilon \big)
\end{aligned}
$$

where $w^{\text{sickness}}$, $w^{\text{treatment}}$ apply pneumonia-specific weights, and $\gamma$ controls for the outcome confounding strength. We choose a sparse $w^{\text{treatment}}$ to target treatments with high physician-level variation (see Appendix F).

We compare LIRR against PCA, OLS, LASSO, Partial least squares (PLS), and DeepIV. We first identify a low-dimensional representation: for LIRR, we follow Algorithm 2 with $r = 5$, and for PCA, we extract the top 5 principal components (taking the right singular vector matrix from the SVD decomposition as $\hat{B}$). The improvement direction $\hat{\theta}$ is then identified via 2SLS in this latent space. In contrast, $\hat{\theta}$ is estimated directly on the observed features $X$ for OLS, LASSO, PLS, and DeepIV, where the regularization weight of LASSO is chosen by grid search to minimize MSE. The perturbed treatment is then $X' = \mathbb{I}(X - \alpha u > \tau)$, where the intervention direction $u$ corresponds to $\hat{B}^\top \hat{\theta}$ for latent models, and the direct coefficient vector $\hat{\theta}$ for the regressions. Setting $\alpha = 10$ and $\tau = 0.2$, improvement is defined as $Y(X) - Y(X')$.

Table 1 and Figure 3 summarize our empirical findings. While PCA provides a competitive baseline, its variance-maximization objective prioritizes encoding confounding components over instrumental variations, causing it to overlook true causal directions. OLS fails due to presence of unobserved confounding. Similarly, PLS identifies components correlated with the outcome that heavily align with these confounding directions, resulting in worse performance than OLS under high $\gamma$ noise regimes. LASSO's failure further highlights the limits of high-dimensional selection: sparsity penalties frequently select statistically convenient but sometimes physically impossible treatment combinations, resulting in no improvement. DeepIV also underperforms in this scenario; its highly complex architecture is not suited for the linear DGP, requiring significantly more data to generalize effectively. In contrast, LIRR successfully isolates the instrumental subspace to recover the true causal coordinates, and lead to the largest improvements overall.

We note that while physician assignment is a strong instrument in theory, real-world clinical settings present potential violations. For instance, senior physicians often manage

severe cases where acute severity ($V$, not simulated here) interacts with pre-existing conditions ($U$). This would violate the joint independence required for IRAE, but not needed for LIRR.

*Table 1.* Average Test Improvement in Recovery Time Across Confounding Strengths. ($\alpha = 10$)

| | Saved time ($\pm$ SD) under Confounding Strength ($\gamma$) | | |
|---|---|---|---|
| Model | $\gamma = 1.0$ | $\gamma = 2.0$ | $\gamma = 3.0$ |
| DeepIV | $0.30 \pm 1.04$ | $0.86 \pm 1.19$ | $0.58 \pm 0.79$ |
| LASSO | $0.00 \pm 0.00$ | $0.00 \pm 0.00$ | $-1.33 \pm 2.21$ |
| PLS | $\mathbf{40.95 \pm 8.08}$ | $6.67 \pm 9.94$ | $3.22 \pm 10.83$ |
| OLS | $19.86 \pm 7.50$ | $3.84 \pm 3.25$ | $0.86 \pm 5.89$ |
| PCA + 2SLS | $33.00 \pm 9.53$ | $33.04 \pm 9.54$ | $33.05 \pm 9.55$ |
| LIRR (Ours) | $34.64 \pm 11.21$ | $\mathbf{34.74 \pm 11.25}$ | $\mathbf{34.99 \pm 11.20}$ |

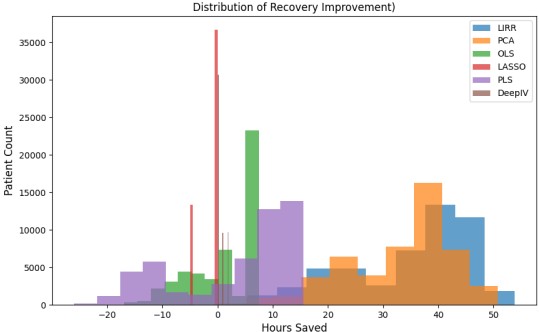

*Figure 3.* Comparison of Distribution of Recovery Time improvements for $\gamma = 3.0$.

**MNIST Experiment** We investigate a nonlinear setting using colored MNIST digits as treatment $X$. Instruments $Z$ and confounders $U$ are randomly generated 2D vectors, while color features $D = (R, G, B)$ are influenced by both $Z$ and $U$; the outcome is defined as $Y = R + G + B$, meaning better outcomes correspond to brighter digits. The grayscale digit structure $V$ serves as a baseline, and $X$ is obtained by multiplying the grayscale image with the color features. This setup represents a realistic scenario where minor violations to the assumptions are presented: continuous color gradients prevent perfect recovery of the underlying representation (violating invertibility), and $\dim(D) > \dim(Z)$ ensures the mixing matrix is not full row rank.

We compare IRAE ith different regularization weights against a vanilla Autoencoder (AE). Bottleneck sizes are set to $\dim(\tilde{D}) = \dim(Z)$ for most methods to satisfy downstream IV order conditions, while IRAE and IRAE[2] use a total dimension of 10 to accommodate both color features and digit morphology ($\dim(\tilde{D}) + \dim(\tilde{V}) = 10$). Following representation learning, we apply 2SLS to identify the outcome-improving intervention direction and decode back to $X$-space for evaluation (see Appendix I, J for details, including ablation studies on different bottle neck sizes,

*Table 2.* Average Test Improvement Comparison of 5 Methods on MNIST Data (Mean $\pm$ Std) with Sample Size = 10,000 (Mean $\pm$ Std).

| Method | reconstructed ($\alpha = 0$) | intervened ($\alpha = 0.2$) | intervened ($\alpha = 1.0$) |
|---|---|---|---|
| Vanilla AE | -0.51 $\pm$ 0.03 | -0.50 $\pm$ 0.03 | -0.47 $\pm$ 0.04 |
| IRAE[0] | -0.73 $\pm$ 0.04 | -0.07 $\pm$ 0.15 | 0.04 $\pm$ 0.20 |
| IRAE[1] | -0.73 $\pm$ 0.04 | 0.07 $\pm$ 0.26 | 0.15 $\pm$ 0.29 |
| IRAE[2] | **-0.33 $\pm$ 0.05** | 0.72 $\pm$ 0.48 | 0.92 $\pm$ 0.47 |
| IRAE (Ours) | **-0.33 $\pm$ 0.04** | **0.87 $\pm$ 0.43** | **1.10 $\pm$ 0.46** |

independence test statistics, and training hyperparameter selection).

Our experiments highlight the critical role of instrument regularization in shaping the latent space. The vanilla AE, lacking such regularization, primarily captures digit morphology and ignores color features (Figure 4e). Consequently, IV regression fails to identify meaningful intervention directions, yielding minimal improvement in $Y$ (Figure 4f). In contrast, the instrument regularization in IRAE[0] and IRAE[1] encourages the latent space to capture color information, improving intervention quality at a slight cost to reconstruction fidelity. By expanding the latent dimension in IRAE and IRAE[2], the model effectively separates instrument-driven color features from confounding structures. As shown in Figures 4a and 4c, IRAE successfully encodes instrument-relevant information while maintaining high reconstruction quality; the intervened images contain visually brighter digits.

## 6. Discussion

Our theoretical framework provides a formal foundation for identification. Most of our requirements, such as instrument relevance, unconfounded instrument, and bounded completeness, are standard in the IV literature and can often be assessed via statistical tests or domain knowledge. For Assumptions 2.1 & 2.2, we provide a discussion in Section 2. Our Pneumonia and MNIST experiments further support this practical applicability, demonstrating that the method is robust to minor violations of invertibiltiy and mismatch between natural dimensions of instrument and latent treatment.

Furthermore, Assumption 4.1 captures two standard relationships: $U \perp\!\!\!\perp Z$, the standard IV unconfoundedness condition, and $D \perp\!\!\!\perp V$, the separation of endogenous and exogenous variation. While joint independence is a stronger condition, it is essential for formal identification in this work. To illustrate this limitation, we include a naive linear example in Appendix G where LIRR underperforms when this assumption is excessively violated, similar to standard IV method in low-dimension regime. We discuss a relaxation, allowing for conditional joint independence by incorporating observed covariates, in Appendix D. Developing robust sensitivity analysis frameworks to quantify the impact of such violation or removing the need of such assumption

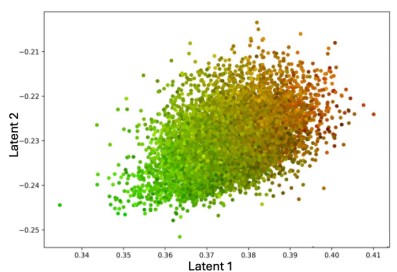

*(a)* Scatter plot of IRAE representation, colored by instruments.

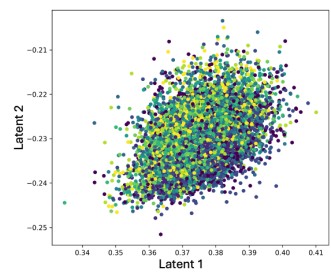

*(b)* Scatter plot of IRAE representation, colored by digit.

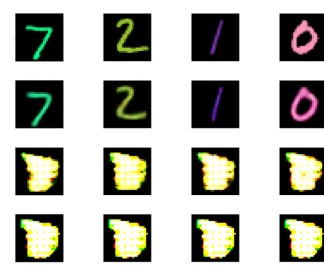

*(c)* From top to bottom: Original, reconstructed, and intervened ($\alpha = 0.2, 1.0$) digits fitted on IRAE.

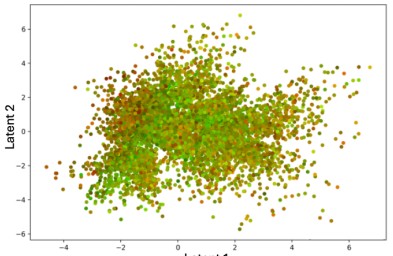

*(d)* Scatter plot of Vanilla AE representation, colored by instruments.

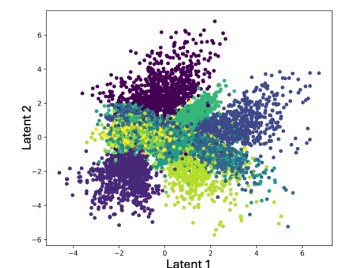

*(e)* Scatter plot of Vanilla AE representation, colored by digit.



*(f)* From top to bottom: Original, reconstructed, and intervened ($\alpha = 0.2, 1.0$) digits fitted on Vanilla.

*Figure 4.* Comparison of latent space representations and incremental intervention for MNIST experiment 1. The top row illustrates the proposed **IRAE** framework, while the bottom row demonstrates the baseline **Vanilla AE**. More figures and ablation studies included in Appendix I, J.

altogether is an important direction for future work.

The transition from these theoretical conditions to a practical model involves a flexible specification of the latent space. Increasing the dimensionality of $\tilde{V}$ renders the invertible encoding assumption more plausible. By allowing the representation to capture exogenous variation independent of the instrument, the model achieves better reconstruction and improved interpretability. In this sense, the complexity of $\tilde{V}$ can be treated as a standard hyperparameter during training.

Other computational considerations include choices of regularization weights and independence test statistics. In our experiments, tuning the regularization weights proved straightforward. We follow a magnitude balancing heuristic: weights are adjusted so that the reconstruction loss and independence penalties maintain a relatively similar magnitude during training iterations. To further assist practitioners, we also included results of setting various weights to 0s in the experiments. This allows users to select the level of regularization that suits their task and prior knowledge.

Lastly, we compare our method to applying flexible Instrumental Variable (IV) methods such as DeepIV directly directly to raw, high-dimensional variables. The later often introduces estimation vulnerabilities: Models optimized directly on high dimensional features require large sample sizes to satisfy support and order conditions. Manipulating raw variables in isolation may also risk yielding perturbations that are off-manifold, representing feature combinations that are physically impossible. Mapping inputs to a low-dimensional causal representation isolates the underlying manifold effectively spanned by the instrument, which is more likely to satisfy the requisite support and order assumptions. When this latent bottleneck is paired with a decoder to map configurations back to the observed space space, any counterfactual perturbation remains consistent by the data manifold. This ensures that latent modifications are physically grounded, and the decoder directly translate the perturbation back into interpretable interventions.

In conclusion, this work establishes instrument guidance as a necessary paradigm within causal representation learning under unobserved confounding. By validating the proposed LIRR and IRAE models, we demonstrate that incorporating instrumental variables directly into the bottleneck is a viable, theoretically grounded approach to resolving unobserved confounding in high-dimensional, unstructured treatment pathways.

## Impact Statement

This work advances causal machine learning by studying how learned representations can enable causal identification in IV settings where treatment is high-dimensional. By integrating representation learning with instrumental variable methods, the approach may expand the applicability of causal inference to domains involving unstructured or complex data. At the same time, the use of learned representations introduces additional modeling assumptions, underscoring the importance of careful validation and responsible use in practice.

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

# A. Further Related Work

In this section we provide a more discussion on related work that is not covered in the main text.

**Further Discussion on Identifying Representations for Intervention Extrapolation** Similar to our work, Saengkyongam et al. proposed the *Rep4Ex* approach which tries to solve the task of interventional outcome prediction by identifying the SCM. Importantly, although they work with a similar SCM as we do (Equation 1), the level of intervention differs - our work considers interventions on the latent treatment space ($D$), while Saengkyongam et al. considers intervening on $Z$ (using notations in Equation 1). Moreover, our work is motivated by the presence of unobserved confounding between the latent representation of the treatment and the outcome, whereas their work is motivated by the need to extrapolate to unseen interventions, while the treatment that they consider is fully exogenous. Like our approach, they employ autoencoders to learn latent representations from potentially high-dimensional observed features, but use maximum moment restriction (MMR) regularization (Muandet et al., 2020a) to enforce the constraint $E[e_D(X) - AZ|Z] = 0$. This can be achieved when $E[e_D(X) - AZ] = 0$ and $e_D(X) - AZ \perp\!\!\!\perp Z$, corresponding to our $\lambda$ and $\mu_1$ term in Equation (IRAE). Additionally, while *Rep4Ex* assumes a deterministic mixing function from the latent representation to the observables $X$, our method explicitly handles noisy observations of X through $e_V(X)$, which allows for broader generalization.

**Further Discussion on Representation Learning and OMV bias of Observed Confounding** Prior work on OMV bias assumes that all confounders are present in the high-dimensional treatment data and focuses on learning representations that retain this confounding information. For example, Vafa et al. add a group-membership prediction loss to the wage-prediction objective to encourage the representation to preserve confounding features. In *GraphITE*, a separate model learns molecular-structure representations and employs HSIC penalties to disentangle treatment and confounding components before feeding them into the outcome model (Harada and Kashima, 2021). Melnychuk et al. address OMV bias using orthogonal (OR) learners; however, these methods still require that all confounders be observed so that the unconstrained nuisance estimators in OR-learners can correct bias. In contrast, our work studies the setting where confounders are unobserved but instruments are available, enabling identification by leveraging IV structure rather than assuming the treatment contains all confounding information.

**Other Dimensionality Reduction for High Dimensional Treatments** When learning a representation for the treatment, it is important for the learned representation to capture all causal factors so that the causal relationship is preserved for downstream estimation tasks like treatment effect estimation. Nabi et al. utilize semi-parametric inference theory for structural models to provide a generalized the sufficient dimension reduction approach for learning lower-dimensional representation for treatment, while capturing the relationship between the treatment and the mean counterfactual outcome. Andreu et al. employed a contrastive approach to learn a representation of the high-dimensional treatments. These works studied settings that did not involve the presence of unobserved confounders of the treatment, while we focus on heavily confounded high dimensional structured treatments. Moreover, in these works, the selection of causally relevant factors are guided by the outcome, where as we take an inherently different approach that learns the latent representations using auxiliary information from instrumental variables instead of the treatment.

**Independence Conditions** In our work, we show that independence between certain variables (for more details, see Theorem 4.5) is desirable for identification. We enforce the independence condition by incorporating a Hilbert-Schmidt Independence Criterion (HSIC) (Gretton et al., 2007) regularizer. This approach has also been adopted in prior research: for instance, Lopez et al. employed HSIC regularization to mitigate bias in observational datasets for applications in counterfactual policy optimization, while Harada and Kashima use it to learn a representations of the treatment that is independent with the target individual in order to mitigate selection bias.

# B. Proof of Linear Identification

Before proving the main theorem, we first present some useful lemma.

**Lemma B.1.** *Suppose $A$ is a $n \times k$ matrix with full row rank ($k > n$), and $B$ is a $m \times n$ matrix, with full column rank ($m > n$). Then the columns of $C = BA$ spans the same space as the columns of $B$.*

*Proof of Lemma B.1.* Let $\mathcal{R}(\cdot)$ denote the column space of a matrix.

For any $x \in \mathcal{R}(B)$, there exist vector $y$ such that $x = By$. Since $A$ is full row rank, we know that $AA^+ = I_n$, and $x = By = BAA^+y = C(A^+y)$. Therefore $x \in \mathcal{R}(C)$, so $\mathcal{R}(B) \subseteq \mathcal{R}(C)$.

Similarly, for any $x \in \mathcal{R}(C)$, there exist vector $y$ such that $x = BAy = B(Ay)$. So $x \in \mathcal{R}(B)$, and we have $\mathcal{R}(C) \subseteq \mathcal{R}(B)$.

Together, we have $\mathcal{R}(C) = \mathcal{R}(B)$.

$\square$

Now we proceed to prove Theorem 3.1.

*Proof of Theorem 3.1.* From Equation 7, we have that:

$$X = BAZ + B_\perp V + BU$$

Then taking the conditional expectation over $Z$, we have:

$$\begin{aligned}
\mathbb{E}[X|Z] &= BAZ + \mathbb{E}[B_\perp V + BU] \\
&= BAZ + \mathbb{E}[B_\perp \mathbb{E}[V|Z]] + \mathbb{E}[B\mathbb{E}[U|Z]] \\
&= BAZ + B_\perp \mathbb{E}[V] + B\mathbb{E}[U] \qquad\qquad \text{(Since } V \perp\!\!\!\perp Z \text{ and } U \perp\!\!\!\perp Z) \\
&= BAZ
\end{aligned}$$

Thus $C := BA$ can be uniquely identified as the solution to the linear regression problem, regressing $X$ on $Z$. Consider the SVD decomposition of $C = \mathcal{U}\Sigma\mathcal{V}^\top$. Let $\hat{B} = \mathcal{U}$, and $\hat{A} = \Sigma\mathcal{V}^\top$. Then by Lemma B.1, we have that the columns of $\hat{B}$ spans the same space as the columns of $B$. In other words, there exist an invertible change of basis matrix $P$ such that $B = \hat{B}P$. Since $\hat{B}$ is orthonormal (by construction of SVD), we have that $\hat{B}^T\hat{B} = I_r$, and $P = \hat{B}^T B$. As a result, we also have:

$$\begin{aligned}
D = B^+ X &= (B^T B)^{-1} B^T X \\
&= (P^T \hat{B}^T \hat{B} P)^{-1} P^T \hat{B}^T X \\
&= (P^T P)^{-1} P^T \hat{B}^T X \\
&= P^{-1} \hat{B}^T X = P^{-1}\tilde{D}
\end{aligned}$$

Next, we show that $\tilde{\theta} = (P^{-1})^T \theta$. The LIRR algorithm solves for $\tilde{\theta}$ from the following moment equation:

$$\begin{aligned}
0 &= \mathbb{E}[Z(Y - \tilde{\theta}^T \hat{D})] \\
&= \mathbb{E}[Z(\theta^T D + \eta(V, U, \epsilon) - \hat{\theta}^T PD)] \\
&= \mathbb{E}[Z(\theta^T D - \tilde{\theta}^T PD)] \qquad\qquad \text{(Since } U, V, \epsilon \perp\!\!\!\perp Z \text{ and } \mathbb{E}[\eta(U, v, \epsilon)] = 0) \\
&= \mathbb{E}[ZD^T](\theta - P^T\tilde{\theta}) \\
&= \mathbb{E}[Z(Z^T A^T + U^T)](\theta - P^T\tilde{\theta}) \\
&= \mathbb{E}[ZZ^T]A^T(\theta - P^T\tilde{\theta})
\end{aligned}$$

Since the instruments are not co-linear, we have that $\mathbb{E}[ZZ^T] \succ 0$, i.e. $\mathbb{E}[ZZ^T]$ is invertible. Thus $\mathbb{E}[ZZ^T]A^T(\theta - P^T\tilde{\theta}) = 0$ if and only if $A^T(\theta - P^T\tilde{\theta}) = 0$. Since $A^T$ has full column rank, then by the Rank-Nullity theorem, the null space of $A^T = 0$. Together, this shows that $\tilde{\theta} = (P^{-1})^T\theta$ is the unique solution to the moment condition.

Lastly, we show that the intervened outcome is guaranteed improvement in expectation. Consider an intervention in the direction of $u = \tilde{\theta}/\|\tilde{\theta}\|$ in the $\tilde{D}$ space, this maps to an intervention in the $D$ space as:

$$e_D(t(X)) = B^+ t(X) = D + \alpha B^+ \hat{B} \tilde{\theta}$$

$$= D + \alpha P^{-1} \frac{\tilde{\theta}}{\|\tilde{\theta}\|} = D + \alpha P^{-1} \frac{(P^{-1})^\top \theta}{\|(P^{-1})^\top \theta\|}$$

Since, we intervene only in $D$, $e_V(t(X)) = V$. Then, we can compute the intervened outcome:

$$\begin{aligned}
\mathbb{E}[Y_{\alpha u}] &= \mathbb{E}[\theta^T e_D(t(X)) + \eta(e_V(t(X)), U, \epsilon)] \\
&= \mathbb{E}[\theta^T e_D(t(X))] && (e_V(t(X)) = V, \text{ and } \mathbb{E}[\eta(U, v, \epsilon)] = 0) \\
&= \mathbb{E}\left[\theta^T \left(D + \alpha P^{-1} \frac{(P^{-1})^\top \theta}{\|(P^{-1})^\top \theta\|}\right)\right] \\
&= \mathbb{E}[\theta^T D + \alpha \|(P^{-1})^\top \theta\|] = \mathbb{E}[Y] + \alpha \|(P^{-1})^\top \theta\|
\end{aligned}$$

$\square$

# C. Proof of Nonlinear Identification

## C.1. Theorem 4.5

*Proof of Theorem 4.5.* By definition of $(\tilde{D}, \tilde{V})$, we have:

$$(\tilde{D}, \tilde{V}) = \tilde{e}(X) = \tilde{e} \circ f(D, V) =: q(D, V)$$

Denote with $q_1(D, V)$ the $\tilde{D}$ component of the output of $q$ and $q_2$ the $\tilde{V}$ component.

Since we have that $\tilde{D} = \tilde{A}Z + \tilde{U}$, with $\tilde{U} \perp\!\!\!\perp Z$ and $\mathbb{E}[\tilde{U}] = 0$, we can write:

$$\mathbb{E}[\tilde{D} \mid Z = z] = \mathbb{E}[\tilde{A}Z + \tilde{U} \mid Z = z] = \tilde{A}z$$

Moreover:

$$
\begin{aligned}
\mathbb{E}[\tilde{D} \mid Z = z] &= \mathbb{E}[q_1(D, V) \mid Z = z] \\
&= \mathbb{E}[q_1(Az + U, V) \mid Z = z] \\
&= \mathbb{E}[q_1(Az + U, V)] && (Z \perp\!\!\!\perp \{U, V\}) \\
&= \mathbb{E}_U[\mathbb{E}_V[q_1(Az + U, V)]] && (U \perp\!\!\!\perp V) \\
&= \mathbb{E}_U[\tilde{q}_1(Az + U)] && (\tilde{q}_1(d) \triangleq \mathbb{E}_V[q_1(d, V)]) \\
&= \mathbb{E}[\tilde{q}_1(Az + U)]
\end{aligned}
$$

Thus we can conclude that:

$$\tilde{A}z = \mathbb{E}[\tilde{D} \mid Z = z] = \mathbb{E}[\tilde{q}_1(Az + U)]$$

Since this holds for all $z \in \mathcal{Z}$ and since $\mathcal{Z}$ is an open set, we can take the derivative with respect to $z$, to derive:

$$\forall z \in \mathcal{Z} : \tilde{A} = \partial_z \mathbb{E}[\tilde{q}_1(Az + U)]$$

Since $q_1$ is continuously differentiable with bounded derivatives, the same holds for $\tilde{q}_1$ and therefore we can exchange the order of differentiation and expectation:

$$\tilde{A} = \mathbb{E}[\partial_z \tilde{q}_1(Az + U)]$$

Letting $\tilde{q}_1^{(1)}$ denote the Jacobian of the function $\tilde{q}_1(d)$, we can write by the chain rule:

$$
\begin{aligned}
\tilde{A} &= \mathbb{E}[\tilde{q}_1^{(1)}(Az + U)A] \\
&= \mathbb{E}[\tilde{q}_1^{(1)}(Az + U)]A \\
&= \mathbb{E}[\tilde{q}_1^{(1)}(Az + U) \mid Z = z]A && (Z \perp\!\!\!\perp U) \\
&= \mathbb{E}[\tilde{q}_1^{(1)}(AZ + U) \mid Z = z]A \\
&= \mathbb{E}[\tilde{q}_1^{(1)}(D) \mid Z = z]A
\end{aligned}
$$

Since $A$ is full row rank, we have that $AA^+$ is invertible. Thus we can write:

$$\tilde{A}A^+ = \mathbb{E}[\tilde{q}^{(1)}(D) \mid Z = z]$$

or equivalently:

$$\forall z \in \mathcal{Z} : \mathbb{E}[\tilde{q}_1^{(1)}(D) - \tilde{A}A^+ \mid Z = z] = 0$$

By the bounded completeness assumption and since both $\tilde{A}A^+$ and $\tilde{q}_1^{(1)}$ are bounded, the latter implies that:

$$\forall d \in \mathcal{D} : \tilde{q}_1^{(1)}(d) = \tilde{A}A^+$$

or equivalently that:

$$\tilde{q}_1(d) = \tilde{A}A^+ d + \tilde{\nu}$$

for some constant vector $\nu$. Moreover,

$$
\begin{aligned}
\mathbb{E}[\tilde{D}] &= \mathbb{E}[\tilde{q}_1(D)] \\
&= \tilde{A}A^+ \mathbb{E}[D] + \tilde{\nu} \\
&= \tilde{A}A^+ A\mathbb{E}[Z] + \tilde{A}A^+ \mathbb{E}[U] + \tilde{\nu} \\
&= \tilde{\nu}
\end{aligned}
$$

But we also have $\mathbb{E}[\tilde{D}] = A\mathbb{E}[Z] + \mathbb{E}[\tilde{U}] = 0$. Hence, we have that $\tilde{\nu} = 0$. Thus:

$$\forall d \in \mathcal{D} : \tilde{q}_1(d) = \tilde{A}A^+ d$$

Next, we argue that $\tilde{A}A^+$ is an invertible matrix. Note that:

$$
\begin{aligned}
\mathbb{E}[\tilde{D}Z^\top] &= \mathbb{E}[(\tilde{A}Z + \tilde{U})Z^\top] \\
&= \tilde{A}\mathbb{E}[ZZ^\top] && (\tilde{U} \perp\!\!\!\perp Z, \mathbb{E}[Z] = 0, \mathbb{E}[\tilde{U}] = 0)
\end{aligned}
$$

Moreover:

$$
\begin{aligned}
\mathbb{E}[\tilde{D}Z^\top] &= \mathbb{E}[q_1(D,V)Z^\top] \\
&= \mathbb{E}[q_1(AZ + U, V)Z^\top] \\
&= \mathbb{E}[\mathbb{E}[q_1(AZ + U, V) \mid Z, U]Z^\top] \\
&= \mathbb{E}[\tilde{q}_1(AZ + U)Z^\top] && (Z \perp\!\!\!\perp U \perp\!\!\!\perp V) \\
&= \mathbb{E}[\tilde{q}_1(D)Z^\top] \\
&= \mathbb{E}[(\tilde{A}A^+ D)Z^\top] \\
&= \tilde{A}A^+ \mathbb{E}[DZ^\top] \\
&= \tilde{A}A^+ \mathbb{E}[(AZ + U)Z^\top] \\
&= \tilde{A}A^+ A\mathbb{E}[ZZ^\top] && (Z \perp\!\!\!\perp U, \mathbb{E}[U] = 0)
\end{aligned}
$$

Thus we have concluded that:

$$\tilde{A}\mathbb{E}[ZZ^\top] = \mathbb{E}[\tilde{D}Z^\top] = \tilde{A}A^+ A\mathbb{E}[ZZ^\top]$$

Since $\mathbb{E}[ZZ^\top]$ is assumed to be invertible, the latter implies that:

$$\tilde{A} = \tilde{A}A^+ A$$

By Lemma C.2 in Appendix C.3, since $\tilde{A}$ and $A$ have full row rank, the row span of $\tilde{A}$ is equal to the row span of $A$ and the matrix $\tilde{A}A^+$ is invertible.

We have thus concluded that:

$$\forall d \in \mathcal{D} : \tilde{q}_1(d) = \tilde{A}A^+ d$$

and $\tilde{A}A^+$ is invertible.

Consider any solution with perfect encoder-decoder pair $(\tilde{e}, \tilde{f})$, and $\tilde{A}$ that satisfies the conditions of the theorem and minimizes the objective function:

$$\mathbb{E}[\|\tilde{e}_D(X) - \tilde{A}Z\|^2] = \mathbb{E}[\|\tilde{D} - \tilde{A}Z\|^2]$$

For any feasible solution, we can decompose this objective into two components by centering around

$$\mu_{\tilde{A}}(d) \triangleq \tilde{A}A^+ d$$

i.e.:

$$\mathbb{E}[\|\tilde{D} - \tilde{A}Z\|^2] = \mathbb{E}[\|\tilde{D} - \mu_{\tilde{A}}(D) + \mu_{\tilde{A}}(D) - \tilde{A}Z\|^2]$$
$$= \mathbb{E}[\|\tilde{D} - \mu_{\tilde{A}}(D)\|^2 + \|\mu_{\tilde{A}}(D) - \tilde{A}Z\|^2] + 2\mathbb{E}[(\tilde{D} - \mu_{\tilde{A}}(D))^\top (\mu_{\tilde{A}}(D) - \tilde{A}Z)]$$

Consider the inner product term. Since we have that:

$$\mathbb{E}[\tilde{D} - \mu_{\tilde{A}}(D) \mid D, Z] = \mathbb{E}[q_1(D, V) - \tilde{q}_1(D) \mid D, Z]$$
$$= \mathbb{E}[q_1(D, V) \mid D, Z] - \tilde{q}_1(D)$$
$$= \mathbb{E}[q_1(D, V) \mid D, Z] - \mathbb{E}[q_1(D, V) \mid D]$$

Since $Z \perp\!\!\!\perp U \perp\!\!\!\perp V$, we have by Lemma C.3 in Appendix C.3 that $Z \perp\!\!\!\perp V \mid \mathbb{1}\{AZ + U = d\}$:

$$\mathbb{E}[q_1(D, V) \mid D = d, Z] = \mathbb{E}[q_1(d, V) \mid D = d, Z] = \mathbb{E}[q_1(d, V) \mid D = d] = \mathbb{E}[q_1(D, V) \mid D = d]$$

Thus:

$$\mathbb{E}[\tilde{D} - \mu_{\tilde{A}}(D) \mid D, Z] = 0$$

From this we conclude that for any feasible solution $\tilde{e}, \tilde{f}, \tilde{A}$, we have that the objective can be decomposed as:

$$\mathbb{E}[\|\tilde{e}_D(X) - \tilde{A}Z\|^2] = \mathbb{E}[\|\tilde{D} - \mu_{\tilde{A}}(D)\|^2] + \mathbb{E}[\|\mu_{\tilde{A}}(D) - \tilde{A}Z\|^2]$$
$$= \mathbb{E}[\|q_1(D, V) - \mu_{\tilde{A}}(D)\|^2] + \mathbb{E}[\|\mu_{\tilde{A}}(D) - \tilde{A}Z\|^2]$$

Suppose that with positive probability, we have that $q_1(D, V) \neq \mu_{\tilde{A}}(D) = \tilde{A}A^+ D$. Then we have that:

$$\mathbb{E}[\|q_1(D, V) - \mu_{\tilde{A}}(D)\|^2] > 0$$

In this case, we will provide an alternative feasible solution, which achieves smaller objective than $\tilde{e}, \tilde{f}, \tilde{A}$. Consider the solution:

$$\tilde{e}'(x) = (\tilde{A}A^+ e_D(x), e_V(x))$$
$$\tilde{f}'(d, v) = f((\tilde{A}A^+)^{-1}d, v)$$

Note that we used the fact that for any feasible solution, we have already shown that $\tilde{A}A^+$ is invertible. Moreover, note that for this solution we have that, $\tilde{e}', \tilde{f}'$ is invertible, since $e, f$ is invertible and $\tilde{A}A^+$ is invertible. Finally,

$$\tilde{f}' \circ \tilde{e}'(x) = x$$
$$\tilde{e}'(f(d, v)) = (\tilde{A}A^+ e_D(f(d, v)), e_V(f(d, v))) = (\tilde{A}A^+ d, v)$$

Thus:

$$\tilde{D} = \tilde{e}'_D(X) = \tilde{e}'_D(f(D, V)) = \tilde{A}A^+ D = \tilde{A}A^+ AZ + \tilde{A}A^+ U = \tilde{A}Z + \tilde{A}A^+ U$$

Where we used the fact that we have already shown (in the proof of Theorem 4.5) that $\tilde{A}A^+ A = \tilde{A}$. Thus, we also have that:

$$\tilde{D} = \tilde{A}Z + \tilde{U}$$

where $\tilde{U} = \tilde{A}A^+ U$ and satisfies $\tilde{U} \perp\!\!\!\perp Z$ and $\mathbb{E}[\tilde{U}] = 0$.

Therefore, this new solution is a feasible solution. Moreover, since under this solution we have that $\tilde{D} = \tilde{A}A^+ D = \mu_{\tilde{A}}(D)$, the first part of the objective vanishes and the objective takes the value:

$$\mathbb{E}[\|\mu_{\tilde{A}}(D) - \tilde{A}Z\|^2] < \mathbb{E}[\|q_1(D, V) - \mu_{\tilde{A}}(D)\|^2] + \mathbb{E}[\|\mu_{\tilde{A}}(D) - \tilde{A}Z\|^2]$$

contradicting the optimality of the original solution.

Thus we have derived that for any optimal feasible solution, it must hold that with probability 1:

$$\tilde{D} = q_1(D, V) = \mu_{\tilde{A}}(D) = \tilde{A}A^+ D \tag{10}$$

with $\tilde{A}A^+$ an invertible matrix. Moreover, this implies that $\tilde{U} = \tilde{D} - \tilde{A}Z = \tilde{A}A^+ U$. $\qquad \square$

## C.2. Proof of Positive Improvement

*Proof of Theorem 4.6.* In this proof, we show that intervention in the direction of average derivatives of $\tilde{h}$ guarantees positive improvement for sufficiently small $\alpha$, assuming that $h$ is twice differentiable. If we perform the intervention $\tilde{D} + \alpha u$, then we have by Lemma C.1 that:

$$D_{\alpha u}, V_{\alpha u} = (D + \alpha P^{-1}u, q_2^{-1}(\tilde{D} + \alpha u, \tilde{V}))$$

Since $\mathcal{D} = \mathbb{R}^r$ and since $P$ is an invertible matrix, we have that $\tilde{\mathcal{D}} = \mathbb{R}^r$. Thus, for all $d \in \tilde{\mathcal{D}}$, we also have that $d + au \in \tilde{\mathcal{D}}$. By Lemma C.1, have that for all $d \in \tilde{\mathcal{D}}$:

$$\text{Law}(q_2^{-1}(d, \tilde{V})) = \text{Law}(q_2^{-1}(d + \alpha u, \tilde{V}))$$

By Theorem 4.5, we have that, with probability 1, $\tilde{D} = PD$ and $\tilde{U} = PU$. Moreover, by assumption, we have that $\tilde{V} \perp\!\!\!\perp \tilde{U} \perp\!\!\!\perp Z$, which implies

$$\tilde{V} \perp\!\!\!\perp \{\tilde{A}Z + \tilde{U}, P^{-1}\tilde{U}\} \implies \tilde{V} \perp\!\!\!\perp \{\tilde{D}, U\},$$

By Lemma C.1, we also have that:

$$\text{Law}(q_2^{-1}(d, \tilde{V}) \mid \tilde{D} = d, U) = \text{Law}(q_2^{-1}(d + \alpha u, \tilde{V}) \mid \tilde{D} = d, U)$$
$$\implies \text{Law}(q_2^{-1}(\tilde{D}, \tilde{V}) \mid \tilde{D}, U) = \text{Law}(q_2^{-1}(\tilde{D} + \alpha u, \tilde{V}) \mid \tilde{D}, U)$$

which by the definition of $V$ and $V_\alpha$ is equivalent to:;

$$\text{Law}(V \mid \tilde{D}, U) = \text{Law}(V_{\alpha u} \mid \tilde{D}, U) \implies \text{Law}(V \mid U) = \text{Law}(V_{\alpha u} \mid U)$$

By the outcome structural equation

$$Y = h(D) + \eta(U, V, \epsilon)$$

we have that:

$$Y_{\alpha u} = h(D + \alpha P^{-1}u) + \eta(U, V_{\alpha u}, \epsilon_Y)$$

and that:

$$\mathbb{E}[Y_{\alpha u} - Y] = \mathbb{E}[h(D + \alpha P^{-1}u) - h(D)] + \mathbb{E}[\eta(U, V_{\alpha u}, \epsilon) - \eta(U, V, \epsilon)]$$

Since $\epsilon \perp\!\!\!\perp \{Z, U, V\}$ and since $V_{\alpha u}$ is a measurable function of these random variables, we have that $\epsilon \perp\!\!\!\perp \{V_{\alpha u}, V, U\}$. Letting $\tilde{\eta}(u, v) = \mathbb{E}_\epsilon[\eta(u, v, \epsilon)]$, we can write:

$$\mathbb{E}[Y_{\alpha u} - Y] = \mathbb{E}[h(D + \alpha P^{-1}u) - h(D)] + \mathbb{E}[\tilde{\eta}(U, V_{\alpha u}) - \tilde{\eta}(U, V)]$$
$$= \mathbb{E}[h(D + \alpha P^{-1}u) - h(D)] + \mathbb{E}[\mathbb{E}[\tilde{\eta}(U, V_{\alpha u}) - \tilde{\eta}(U, V) \mid U]]$$
$$= \mathbb{E}[h(D + \alpha P^{-1}u) - h(D)] \qquad\qquad (\text{Law}(V \mid U) = \text{Law}(V_{\alpha u} \mid U))$$

By a first-order Taylor expansion and since $h$ is twice differentiable with bounded first and second derivatives:

$$\mathbb{E}[Y_{\alpha u} - Y] = \mathbb{E}[\alpha \nabla_D h(D)^\top P^{-1}u] + O(\alpha^2) = \alpha \|(P^{-1})^\top \mathbb{E}[\nabla_D h(D)]\| + O(\alpha^2)$$

$\square$

## C.3. Auxiliary Lemmas

**Lemma C.1.** *Suppose the assumptions of Theorem 4.5 hold, and additionally impose the following constraints on the learned functions $\tilde{e}, \tilde{f}, \tilde{A}$ that minimize the objective in Equation 8:*

- $\tilde{D} \perp\!\!\!\perp \tilde{V}$

- $\tilde{e}$ is an invertible function when restricted to inputs in the image of $f$ and $\tilde{f} \circ \tilde{e}(x) = x$ for all $x \in Im(f)$.

Let $q \triangleq \tilde{e} \circ f$ and $q^{-1} \triangleq e \circ \tilde{f}$. Let $q_2$ denote the $\tilde{V}$-component of the output of $q$ and $q_2^{-1}$ the $V$-component of the output of $q^{-1}$. Then we have that $(\tilde{D}, \tilde{V}) = q(D, V)$ and $(D, V) = q^{-1}(\tilde{D}, \tilde{V})$, almost surely. Moreover, it must also hold with probability 1 that:

- $\tilde{V} = q_2(D, V) = (\tilde{e} \circ f)_2(D, V)^5$ with the property that for all $d, d' \in \mathcal{D}$:

$$Law(q_2(d, V)) = Law(q_2(d', V)).$$

- $V = q_2^{-1}(D, V) = (e \circ \tilde{f})_2(\tilde{D}, \tilde{V})$ with the property that for all $d, d' \in \tilde{\mathcal{D}} \triangleq \{Pd : d \in \mathcal{D}\}$:

$$Law(q_2^{-1}(d, \tilde{V})) = Law(q_2^{-1}(d', \tilde{V})).$$

*Proof of Lemma C.1.* In this proof, we argue about the properties of the second part of the function $q$. Note that since $\tilde{D} \perp\!\!\!\perp \tilde{V}$ and since $\tilde{D} = PD$, for some invertible $P$, with probability 1, we have that $\tilde{V} \perp\!\!\!\perp D$. Thus:

$$q_2(D, V) \equiv \tilde{V} \perp\!\!\!\perp D$$

Since, $D \perp\!\!\!\perp V$, this implies that $Law(q_2(d, V)) = Law(q_2(d', V))$ for all $d, d' \in \mathcal{D}$.

Since $\tilde{e}$ is a bijection when restricted to inputs that are outputs of $f$ and since $f$ is an injection, we have that $\tilde{e} \circ f$ is an injection. Thus there exists a well-defined inverse function $q^{-1} = e \circ \tilde{f}$, such that $D, V = q^{-1}(D, V)$. Let $q_2^{-1}$ be the $V$ component of its output. Since $D \perp\!\!\!\perp V$ and $\tilde{D} = PD$, we have that:

$$V \equiv q_2^{-1}(\tilde{D}, \tilde{V}) \perp\!\!\!\perp \tilde{D}$$

Since $\tilde{D} \perp\!\!\!\perp \tilde{V}$, this implies that $Law(q_2^{-1}(d, \tilde{V})) = Law(q_2^{-1}(d', \tilde{V}))$ for all $d, d' \in \tilde{\mathcal{D}}$ □

**Lemma C.2.** *Suppose $A$ and $B$ are $r \times k$ matrices with full row rank. If $A = AB^+B$, then $rowspan(A) = rowspan(B)$ and $AB^+$ is invertible.*

*Proof.* Consider the thin SVDs of $B = U_B \Sigma_B V_B^\top$ and $A = U_A \Sigma_A V_A^\top$. Then

$$B^+B = V_B \Sigma_B^{-1} U_B^\top U_B \Sigma_B V_B^\top = V_B V_B^\top$$

is the projection onto the row space of $B$. Then, we have:

$$A = AB^+B \quad \Leftrightarrow \quad AV_B V_B^\top = A \tag{11}$$

First, we prove by contradiction that $rowspan(A) = rowspan(B)$. Suppose $x \in rowspan(A) = span(V_A)$, but $x \notin rowspan(B) = span(V_B)$. Let $V_B^\perp$ denote an orthogonal completion of $V_B$, then

$$x \notin span(V_B) \Rightarrow x = V_B V_B^\top x + V_B^\perp (V_B^\perp)^\top x$$

where $V_B^\perp (V_B^\perp)^\top x \neq 0$, which implies $(V_B^\perp)^\top x \neq 0$ as $V_B^\perp$ is orthogonal. Hence, we have the following:

$$\|x\|^2 = x^\top x = x^\top V_B V_B^\top x + x^\top V_B^\perp (V_B^\perp)^\top x = \|V_B^\top x\|^2 + \|(V_B^\perp)^\top x\|^2 > \|V_B^\top x\|^2$$

However, we also have that $AV_B V_B^\top x - Ax = 0$, which implies $u \triangleq V_B V_B^\top x - x \in null\text{-}space(A) = span(V_A^\perp)$. Thus, it should be orthogonal with $x \in span(V_A)$.

$$0 = x^\top u = x^\top (V_B V_B^\top x - x) = \|V_B^\top x\|^2 - \|x\|^2 \neq 0$$

This yields a contradiction! Thus, $rowspan(A) \subseteq rowspan(B)$. Since, both matrices have full row rank, then $A$ and $B$ have the same row space.

---

[5]With $(\tilde{e} \circ f)_2$ we denote the $V$ component of the output of the function $\tilde{e} \circ f_0$.

Now we show that $AB^+$ is invertible:

$$AB^+ = U_A \Sigma_A V_A^\top V_B \Sigma_B^{-1} U_B^\top$$

Since $A, B$ are full row rank, $U_A, U_B, \Sigma_A, \Sigma_B$ are $r \times r$ invertible matrices. So it suffices to show that $V_A^\top V_B$ is invertible. Since $\text{span}(V_A) = \text{span}(V_B)$, there exists an invertible change-of-basis matrix $P$ such that

$$V_B = V_A P \Rightarrow V_A^\top V_B = V_A^\top V_A P = I_r P = P \Rightarrow AB^+ \text{ is invertible.}$$

$\square$

**Lemma C.3.** *If $U \perp\!\!\!\perp V \perp\!\!\!\perp Z$ (jointly independent), then $V \perp\!\!\!\perp Z \mid f(Z, U)$, for any measurable function $U$.*

Let $W = f(Z, U)$. Then:

$$
\begin{aligned}
p(v \mid w, z) &= \int_u p(v \mid w, z, u) p(u \mid w, z) du \\
&= \int_u p(v \mid z, u) p(u \mid w, z) du \\
&= \int_u p(v) p(u \mid w, z) du && (V \perp\!\!\!\perp Z \perp\!\!\!\perp U) \\
&= p(v) \int_u p(u \mid w, z) du \\
&= p(v)
\end{aligned}
$$

**Lemma C.4** (Sufficient Conditions for Bounded Completeness). *Consider $D = A \cdot Z + U, \quad U \perp\!\!\!\perp Z$. $D$ is bounded complete for $Z$ if the following holds:*

- *The measure of $AZ$ is continuous and is supported on $\mathbb{R}^r$.*

- *The density of $U$ is continuous.*

- *The characteristic function of the distribution of $U$ is infinitely often differentiable and does not vanish on the real line.*

*Proof of Lemma C.4.* This result follows as a Corollary of Theorem 2.1 in D'Haultfoeuille, where we consider the special case of linear mappings from $Z$ to $D$. $\square$

Another set of sufficient conditions is given by Saengkyongam et al., requiring only that $AZ$ be supported on an open subset of $\mathbb{R}^r$ rather than on the full space.

**Lemma C.5** (Sufficient Conditions for Bounded Completeness 2). *Consider $D = A \cdot Z + U, \quad U \perp\!\!\!\perp Z$. $D$ is bounded complete for $Z$ if the following holds:*

- *The measure of $AZ$ is continuous and is supported on an open subset of $\mathbb{R}^r$.*

- *The density of $U$ is analytic.*

- *The characteristic function of the distribution of $U$ has no zeros.*

*Proof of Lemma C.5.* This result follows part of the proof of Theorem 6 in Saengkyongam et al.. $\square$

# D. Non-linear setting with covariates

We will expand the nonlinear setting to include covariates. Specifically, the data generating process follows:

$$
\begin{aligned}
D &= A_W \cdot Z + U, & U &\perp\!\!\!\perp Z \mid W \\
X &= f_W(D, V), & V &\perp\!\!\!\perp Z \mid W \\
Y &= h_W(D) + \eta(U, V, \epsilon), & \epsilon &\perp\!\!\!\perp Z \mid W
\end{aligned}
\tag{12}
$$

Where $W$ is dimension $p$ observed covariates or exogenous variables. Denote the domain of $W$ be $\mathcal{W}$. All of $Z, U, V, \epsilon$ can be dependent on $W$.

The assumption required mostly follows the nonlinear setting, with additional conditioning on $W$.

**Assumption D.1** (Invertible Encoding). For every $w \in \mathcal{W}$, the function $f_w$ is invertible. Given $W = w$, write the encoding function $e_w(X) = f_w^{-1}(X) = (D, V)$, i.e. there is a one-to-one correspondence between the high-dimensional treatment $X$ and the characteristics $(D, V)$ that describe the treatment given covariates are $W = w$.

**Assumption D.2** (Full-Rank Latents). Assume that the matrix $A_w$ has full row-rank and $\mathbb{E}[ZZ^\top \mid W = w] \succ 0$ for every $w \in \mathcal{W}$.

**Assumption D.3** (Conditional Joint Independence). Assume that $Z \perp\!\!\!\perp U \perp\!\!\!\perp V \mid W$ (conditionally jointly independent).

**Assumption D.4** (Differentiable Decoding Function). For every $w \in \mathcal{W}$, $f_w$ is a differentiable function with uniformly bounded derivatives.

**Assumption D.5.** $\mathbb{E}[Z \mid W] = 0$, $\mathbb{E}[U \mid W] = 0$, $\mathbb{E}[\eta(U, V, \epsilon) \mid W] = 0$ and the support of $Z$, $\mathcal{Z}$, is an open subset of $\mathbb{R}^k$.

We discussed the sufficient conditions for bounded completeness in D.6, which is a direct result of Theorem 2.1 in D'Haultfoeuille.

**Assumption D.6** (Bounded Completeness). $D$ is bounded complete for Z, that is, for all bounded real functions $h$, we have that:

$$
\mathbb{E}[h_W(D) \mid Z, W = w] = 0 \quad \text{a.s.} \quad \Rightarrow \quad h_w(D) = 0 \quad \text{a.s.}
$$

**Theorem D.7.** *Suppose that the data generating process follows the SEM described in Equation 12, and satisfies Assumptions D.1 & D.2 & D.3 & D.4 & D.5 & D.6. Let $(\tilde{D}_W, \tilde{V}_W) := (\tilde{e}_{W,D}(X), \tilde{e}_{W,V}(X)) = \tilde{e}_W(X)$ denote the learned representations. Consider encoder-decoder pairs with perfect reconstruction, i.e. $X = \tilde{f}_W \circ \tilde{e}_W(X)$. Then, for the solution $\tilde{e}_W, \tilde{f}_W$, and full row rank matrix $\tilde{A}_W$ that minimizes the objective function*

$$
\mathbb{E}[\|\tilde{e}_{W,D}(X) - \tilde{A}_W Z\|^2]
\tag{13}
$$

*subject to the following constraints:*

- *$\tilde{e}_w$ is a differentiable function with uniformly bounded derivatives for all $w \in \mathcal{W}$.*

- *$\tilde{A}_w$ has full row rank for all $w \in \mathcal{W}$.*

- *$\tilde{D}_w = \tilde{A}_w Z + \tilde{U}_w$ for all $w \in \mathcal{W}$ with $\tilde{U}_W \perp\!\!\!\perp Z \mid W$ and $\mathbb{E}[\tilde{U}_W \mid W] = 0$.*

*we have that, with probability 1, $\tilde{D}_W = P_W D$ and $\tilde{U}_W = P_W U$, for $P_W = \tilde{A}_W A_W^+$. Moreover, the matrix $P_W$ is invertible.*

**Lemma D.8.** *Suppose the assumptions of Theorem D.7 hold, and additionally impose the following constraints on the learned functions $\tilde{e}_W, \tilde{f}_W, \tilde{A}_W$ that minimize the objective in Equation 13:*

- *$\tilde{D}_W \perp\!\!\!\perp \tilde{V}_W \mid W$*

- *$\tilde{e}_w$ is an invertible function when restricted to inputs in the image of $f_w$ and $\tilde{f}_w \circ \tilde{e}_w(x) = x$ for all $x \in Im(f)$ and corresponding $w \in \mathcal{W}$.*

Let $q_W \triangleq \tilde{e}_W \circ f_W$ and $q_W^{-1} \triangleq e_W \circ \tilde{f}_W$. Let $q_{W,1}$ denote the $\tilde{D}$-component of the output of $q_W$ and $q_{W,1}^{-1}$ the $D$-component of the output of $q_W^{-1}$.

Then we have that $(\tilde{D}_W, \tilde{V}_W) = q_W(D, V)$ and $(D, V) = q_W^{-1}(\tilde{D}_W, \tilde{V}_W)$, almost surely. Moreover, it must also hold with probability 1 that:

- $\tilde{V}_W = q_{W,1}(D, V) = (\tilde{e}_W \circ f_W)_2(D, V)$ with the property that for given $w$ and all $d, d' \in \mathcal{D}_w := support(D \mid W = w)$:

$$Law(q_{W,2}(d, V) \mid W = w) = Law(q_{W,2}(d', V) \mid W = w).$$

- $V = q_{W,2}^{-1}(D, V) = (e_W \circ \tilde{f}_W)_2(\tilde{D}_W, \tilde{V}_W)$ with the property that for given $w$ and all $d, d' \in \tilde{\mathcal{D}}_w := \{P_w d : d \in \mathcal{D}_w\}$:

$$Law(q_{w,2}^{-1}(d, \tilde{V}_W) \mid W = w) = Law(q_{w,2}^{-1}(d', \tilde{V}_W) \mid W = w).$$

*Remark* D.9. Note that the assumptions that $\mathbb{E}[Z \mid W] = 0$ and $D = A_W Z + U$ are without loss of generality as we can always pre-process Z by centering it. In practice, the assumption that $\tilde{D}_W = \tilde{A}_W Z + \tilde{U}$, with $\tilde{U}_W \perp\!\!\!\perp Z \mid W$ and $\mathbb{E}[\tilde{U}_W \mid W] = 0$ can be achieved by minimizing a square loss with an covariate-specific intercept $C_W$, i.e.

$$\min_{e, f, A, C: e, f \text{ invertible}, e \circ f = \text{identity}} \mathbb{E}[\|e_{W,D}(X) - A_W Z - C_W\|^2]$$

and then defining $\tilde{D}_W = \tilde{e}_D(X) - C_W \triangleq e_{W,D}(X)$, $\tilde{f}_W = f_W + C_W$.

The intervention is carried out in close accordance with Algorithm 1. In particular, we will run an IV analysis, with $Z$ as the instrument, $\tilde{D}$ as the treatment, $W$ as the covariates, and $Y$ as the outcome, to estimate a causal model in representation space by finding a solution to the conditional moment restrictions:

$$\mathbb{E}[Y - \tilde{h}_W(\tilde{D}_W) \mid Z, W = w] = 0 \tag{14}$$

Note that since $\tilde{D}_W = P_W D$ and since $\mathbb{E}[Y \mid Z, W = w] = \mathbb{E}[h_W(D) \mid Z, W = w]$, we have by the completeness assumption that:

$$\mathbb{E}[h_W(D) - \tilde{h}_W(P_W D) \mid Z, W = w] = 0 \Rightarrow h_w(D) = \tilde{h}_w(P_w D) \text{ a.s.}$$
$$\Rightarrow h_w(P_w^{-1} \tilde{D}) = \tilde{h}_w(\tilde{D}_w) \text{ a.s.}$$

**Theorem D.10.** *Assume that:*

$$Y = h_W(D) + \eta(U, V, \epsilon), \quad \epsilon \perp\!\!\!\perp \{Z, U, V\} \mid W$$

*and that $h_W$ is twice differentiable with a bounded second derivative. Let $\tilde{e}_W, \tilde{f}_W, \tilde{A}_W$ be an optimal solution as prescribed in Lemma D.8 with the extra constraint that:*

$$\tilde{U}_W \perp\!\!\!\perp \tilde{V}_W \perp\!\!\!\perp Z \mid W \qquad \text{(conditional joint independence)}$$

*and assume that the assumptions of Lemma D.8 are satisfied. Furthermore, assume that the variable D has full support in $\mathbb{R}^r$ conditional on $W = w$, for all $w \in \mathbb{R}^p$. Then setting $u_w = \tilde{\theta}_w / \|\tilde{\theta}_w\|$, in Algorithm 1, with $\tilde{\theta}_w = \mathbb{E}[\nabla_{\tilde{D}_W} \tilde{h}_W(\tilde{D}_W) \mid W = w]$ and $\tilde{h}$ the solution to the conditional moment restriction problem in Equation (14), we have that:*

$$\mathbb{E}[Y_{\alpha u} - Y \mid W] = \alpha \|(P_W^{-1})^\top \mathbb{E}[\nabla_D h_W(D) \mid W]\| + O(\alpha^2)$$

*Hence, for small enough step size $\alpha$, the identified intervention will achieve a positive improvement on the outcome (assuming that $\mathbb{E}[\nabla_D h_W(D) \mid W] \neq 0$).*

**Instrument Regularized Auto-Encoder**  To achieve the positive improvement as described in Theorem D.10, then we need to incorporate loss components that are minimized only when

- $e_W, f_W$ reconstruct the input $X$,

- $e_{W,D}(X)$ is predicted linearly by $Z$ given $W$ with a full rank matrix $A_W$,

- The residual of this regression $e_{W,D}(X) - A_W Z - C_W$, which approximates $U$, is independent of $Z$ given $W$,

- $Z$ is independent of $e_{W,V}(X)$ given $W$,

- Conditioning on $W$, we have that:

$$e_{W,D} \perp\!\!\!\perp e_{W,V}(X) \mid W$$
$$e_{W,D}(X) - A_W Z - C_W \perp\!\!\!\perp e_{W,V}(X) \mid W$$
$$(e_{W,D}(X) - A_W Z - C_W) \perp\!\!\!\perp e_{W,V}(X) \perp\!\!\!\perp Z \mid W$$

While we do not explicitly enforce $\tilde{A}_W$ to be full row rank, we expect this to be satisfied due to the reconstruction loss and the condition that $\tilde{D}_W \perp\!\!\!\perp \tilde{V}_W \mid W$. Moreover, note that instead of only conditional joint independence of $Z, \tilde{U}_W, \tilde{V}_W$ we also explicitly enforce pairwise independencies for computational reasons.

We introduce the instrument-regularized auto-encoder loss, which incorporates all these elements:

$$
\begin{aligned}
\min_{e,f,A,C} \; & \mathbb{E}\left[\|X - f_W \circ e_W(X)\|^2\right] + \lambda \mathbb{E}\left[\|e_{W,D}(X) - A_W Z - C_W\|^2\right] \\
& + \mu_1 \mathcal{R}_W(e_{W,D}(X) - A_W Z - C_W, Z) \\
& + \mu_2 \mathcal{R}_W(Z, e_{W,V}(X)) \\
& + \mu_3 \big( c_1 \mathcal{R}_W(e_{W,D}(X), e_{W,V}(X)) \\
& \qquad + c_2 \mathcal{R}_W(e_{W,D}(X) - A_W Z - C_W, e_{W,V}(X)) \\
& \qquad + c_3 \mathcal{R}_W(e_{W,D}(X) - A_W Z - C_W, Z, e_{W,V}(X)) \big)
\end{aligned}
\qquad \text{(IRAE with covar)}
$$

$\mathcal{R}_W(A, B)$ or $\mathcal{R}_W(A, B, C)$, denotes any regularizer that can be evaluated on a set of $n$ samples and which takes small values when the random variables $A, B$ or $A, B, C$ are jointly independent condition on $W$. For example, the procedure introduced in (Pogodin et al., 2024). Computing conditional independence statistics is typically more challenging than evaluating unconditional independence, and may introduce additional computational complexity.

# E. Proof of Non-linear Identification with covariates

## E.1. Auxiliary Lemmas

**Lemma E.1.** *If $U \perp\!\!\!\perp V \perp\!\!\!\perp Z \mid W$ (conditionally jointly independent), then $V \perp\!\!\!\perp Z \mid f(Z, U, W), W$, for any measurable function $U$.*

Let $S = f(Z, U, W)$. Then:

$$
\begin{aligned}
p(v \mid s, z, w) &= \int_u p(v \mid s, z, u, w) p(u \mid s, z, w) du \\
&= \int_u p(v \mid z, u, w) p(u \mid s, z, w) du \\
&= \int_u p(v \mid w) p(u \mid s, z, w) du \qquad\qquad (V \perp\!\!\!\perp Z \perp\!\!\!\perp U \mid W) \\
&= p(v \mid w) \int_u p(u \mid s, z, w) du \\
&= p(v \mid w)
\end{aligned}
$$

**Lemma E.2** (Sufficient Conditions for Bounded Completeness)**.** *Consider $D = A_W \cdot Z + U, \quad U \perp\!\!\!\perp Z \mid W$. $D$ is bounded complete for $Z$ if the following holds:*

- *$Z \perp\!\!\!\perp U \mid W$*

- *The measure of $A_W Z$ is continuous and is supported on $\mathbb{R}^r$.*

- *The conditional density of $U \mid W = w$ is continuous for all $w \in \mathcal{W}$.*

- *The conditional characteristic function of the distribution of $U \mid W = w$ is infinitely often differentiable and does not vanish on the real line for all $w \in \mathcal{W}$.*

*Proof of Lemma E.2.* This result follows as a Corollary of Theorem 2.1 in D'Haultfoeuille, where we consider the special case of linear mappings from $Z$ to $D$. $\qquad\square$

## E.2. Theorem D.7 and Lemma D.8

*Proof of Theorem 4.5.* By definition of $(\tilde{D}_W, \tilde{V}_W)$, and given $W = w$, we have:

$$
(\tilde{D}_W, \tilde{V}_W) = \tilde{e}_w(X) = \tilde{e}_w \circ f_w(D, V) =: q_w(D, V)
$$

Denote with $q_{w,1}(D, V)$ the $\tilde{D}_w$ component of the output of $q$ and $q_{w,2}$ the $\tilde{V}_w$ component.

Since we have that $\tilde{D}_w = \tilde{A}_w Z + \tilde{U}_w$, with $\tilde{U}_W \perp\!\!\!\perp Z \mid W$ and $\mathbb{E}[\tilde{U}_W \mid W] = 0$, we can write:

$$
\mathbb{E}[\tilde{D}_W \mid Z = z, W = w] = \mathbb{E}[\tilde{A}_w Z + \tilde{U}_W \mid Z = z, W = w] = \tilde{A}_w z
$$

Moreover:

$$
\begin{aligned}
\mathbb{E}[\tilde{D}_W \mid Z = z, W = w] &= \mathbb{E}[q_{w,1}(D, V) \mid Z = z, W = w] \\
&= \mathbb{E}[q_{w,1}(A_w z + U, V) \mid Z = z, W = w] \\
&= \mathbb{E}[q_{w,1}(A_w z + U, V) \mid W = w] \qquad\qquad (Z \perp\!\!\!\perp \{U, V\} \mid W) \\
&= \mathbb{E}_U[\mathbb{E}_V[q_{w,1}(A_w z + U, V) \mid W = w] \mid W = w] \qquad\qquad (U \perp\!\!\!\perp V \mid W) \\
&= \mathbb{E}_U[\tilde{q}_{w,1}(A_w z + U) \mid W = w] \qquad (\tilde{q}_{w,1}(d) \triangleq \mathbb{E}_V[q_{w,1}(d, V) \mid W = w]) \\
&= \mathbb{E}[\tilde{q}_{w,1}(A_w z + U) \mid W = w]
\end{aligned}
$$

Thus we can conclude that:

$$\tilde{A}_w z = \mathbb{E}[\tilde{D}_W \mid Z = z, W = w] = \mathbb{E}[\tilde{q}_{w,1}(A_w z + U) \mid W = w]$$

Since this holds for all $z \in \mathcal{Z}$ and since $\mathcal{Z}$ is an open set, we can take the derivative with respect to $z$, to derive:

$$\forall z \in \mathcal{Z} : \tilde{A}_w = \partial_z \mathbb{E}[\tilde{q}_{w,1}(A_w z + U) \mid W = w]$$

Since $q_{w,1}$ is continuously differentiable with bounded derivatives, the same holds for $\tilde{q}_{w,1}$ and therefore we can exchange the order of differentiation and expectation:

$$\tilde{A}_w = \mathbb{E}[\partial_z \tilde{q}_{w,1}(A_w z + U) \mid W = w]$$

Letting $\tilde{q}_{w,1}^{(1)}$ denote the Jacobian of the function $\tilde{q}_{w,1}(d)$, we can write by the chain rule:

$$
\begin{aligned}
\tilde{A}_w &= \mathbb{E}[\tilde{q}_{w,1}^{(1)}(A_w z + U) A_w \mid W = w] \\
&= \mathbb{E}[\tilde{q}_{w,1}^{(1)}(A_w z + U) \mid W = w] A_w \\
&= \mathbb{E}[\tilde{q}_{w,1}^{(1)}(A_w z + U) \mid Z = z, W = w] A_w && (Z \perp\!\!\!\perp U \mid W) \\
&= \mathbb{E}[\tilde{q}_{w,1}^{(1)}(A_W Z + U) \mid Z = z, W = w] A_w \\
&= \mathbb{E}[\tilde{q}_{w,1}^{(1)}(D) \mid Z = z, W = w] A_w
\end{aligned}
$$

Since $A_w$ is full row rank, we have that $A_w A_w^+$ is invertible. Thus we can write:

$$\tilde{A}_w A_w^+ = \mathbb{E}[\tilde{q}_w^{(1)}(D) \mid Z = z, W = w]$$

or equivalently:

$$\forall z \in \mathcal{Z} : \mathbb{E}[\tilde{q}_{w,1}^{(1)}(D) - \tilde{A}_w A_w^+ \mid Z = z, W = w] = 0$$

By the bounded completeness assumption and since both $\tilde{A}_w A_w^+$ and $\tilde{q}_{w,1}^{(1)}$ are bounded, the latter implies that:

$$\forall d \in \mathcal{D} : \tilde{q}_{w,1}^{(1)}(d) = \tilde{A}_w A_w^+$$

or equivalently that:

$$\tilde{q}_{w,1}(d) = \tilde{A}_w A_w^+ d + \tilde{\nu}$$

for some constant vector $\nu$. Moreover,

$$
\begin{aligned}
\mathbb{E}[\tilde{D}_W \mid W = w] &= \mathbb{E}[\tilde{q}_{w,1}(D)] \\
&= \tilde{A}_w A_w^+ \mathbb{E}[D \mid W = w] + \tilde{\nu} \\
&= \tilde{\nu}
\end{aligned}
$$

But we also have $\mathbb{E}[\tilde{D}_W \mid W = w] = 0$. Hence, we have that $\tilde{\nu} = 0$. Thus:

$$\forall d \in \mathcal{D} : \tilde{q}_{w,1}(d) = \tilde{A}_w A_w^+ d$$

Next, we argue that $\tilde{A}_w A_w^+$ is an invertible matrix. Note that:

$$
\begin{aligned}
\mathbb{E}[\tilde{D}_W Z^\top \mid W = w] &= \mathbb{E}[(\tilde{A}_w Z + \tilde{U}_W) Z^\top \mid W = w] \\
&= \tilde{A}_w \mathbb{E}[ZZ^\top \mid W = w] && (\tilde{U}_W \perp\!\!\!\perp Z \mid W, \mathbb{E}[Z \mid W] = 0, \mathbb{E}[\tilde{U}_W \mid W] = 0)
\end{aligned}
$$

Moreover:

$$
\begin{aligned}
\mathbb{E}[\tilde{D}_W Z^\top \mid W = w] &= \mathbb{E}[q_{w,1}(D, V) Z^\top \mid W = w] \\
&= \mathbb{E}[q_{w,1}(A_w Z + U, V) Z^\top \mid W = w] \\
&= \mathbb{E}[\mathbb{E}[q_{w,1}(A_w Z + U, V) \mid Z, U, W = w] Z^\top \mid W = w] \\
&= \mathbb{E}[\tilde{q}_{w,1}(A_w Z + U) Z^\top \mid W = w] \qquad\qquad (Z \perp\!\!\!\perp U \perp\!\!\!\perp V \mid W) \\
&= \mathbb{E}[\tilde{q}_{w,1}(D) Z^\top \mid W = w] \\
&= \mathbb{E}[(\tilde{A}_w A_w^+ D) Z^\top \mid W = w] \\
&= \tilde{A}_w A_w^+ \mathbb{E}[D Z^\top \mid W = w] \\
&= \tilde{A}_w A_w^+ \mathbb{E}[(A_w Z + U) Z^\top \mid W = w] \\
&= \tilde{A}_w A_w^+ A_w \mathbb{E}[Z Z^\top \mid W = w] \qquad (Z \perp\!\!\!\perp U \mid W, \mathbb{E}[Z \mid W] = 0, \mathbb{E}[U \mid W] = 0)
\end{aligned}
$$

Thus we have concluded that:

$$
\tilde{A}_w \mathbb{E}[Z Z^\top \mid W = w] = \mathbb{E}[\tilde{D}_W Z^\top \mid W = w] = \tilde{A}_w A_w^+ A_w \mathbb{E}[Z Z^\top \mid W = w]
$$

Since $\mathbb{E}[Z Z^\top \mid W = w]$ is assumed to be invertible, the latter implies that:

$$
\tilde{A}_w = \tilde{A}_w A_w^+ A_w
$$

By Lemma C.2 in Appendix C.3, since $\tilde{A}_w$ and $A_w$ have full row rank, the row span of $\tilde{A}_w$ is equal to the row span of $A_w$ and the matrix $\tilde{A}_w A_w^+$ is invertible.

We have thus concluded that:

$$
\forall d \in \mathcal{D} : \tilde{q}_1(d) = \tilde{A}_w A_w^+ d
$$

and $\tilde{A}_w A_w^+$ is invertible.

Consider any solution with perfect encoder-decoder pair $(\tilde{e}_W, \tilde{f}_W)$, and $\tilde{A}_W$ that satisfies the conditions of the theorem and minimizes the objective function:

$$
\mathbb{E}[\|\tilde{e}_{W,D}(X) - \tilde{A} Z\|^2] = \mathbb{E}[\|\tilde{D}_W - \tilde{A}_W Z\|^2]
$$

Given $W = w$, for any feasible solution, we can decompose this objective into two components by centering around

$$
\mu_{\tilde{A}_w}(d) \triangleq \tilde{A}_w A_w^+ d
$$

i.e.:

$$
\begin{aligned}
\mathbb{E}[\|\tilde{D}_W - \tilde{A}_W Z\|^2] &= \mathbb{E}[\|\tilde{D}_W - \mu_{\tilde{A}_W}(D) + \mu_{\tilde{A}_W}(D) - \tilde{A}_W Z\|^2] \\
&= \mathbb{E}[\|\tilde{D}_W - \mu_{\tilde{A}_W}(D)\|^2 + \|\mu_{\tilde{A}_W}(D) - \tilde{A}_W Z\|^2] \\
&\quad + 2\mathbb{E}[(\tilde{D}_W - \mu_{\tilde{A}_W}(D))^\top (\mu_{\tilde{A}_W}(D) - \tilde{A}_W Z)]
\end{aligned}
$$

Consider the inner product term. Since we have that:

$$
\begin{aligned}
\mathbb{E}[\tilde{D}_W - \mu_{\tilde{A}_W}(D) \mid D, Z, W] &= \mathbb{E}[q_{W,1}(D, V) - \tilde{q}_{W,1}(D) \mid D, Z, W] \\
&= \mathbb{E}[q_{W,1}(D, V) \mid D, Z, W] - \tilde{q}_{W,1}(D) \\
&= \mathbb{E}[q_{W,1}(D, V) \mid D, Z, W] - \mathbb{E}[q_{W,1}(D, V) \mid D, W]
\end{aligned}
$$

Since $Z \perp\!\!\!\perp U \perp\!\!\!\perp V \mid W$, we have by Lemma E.1 that $Z \perp\!\!\!\perp V \mid \{\mathbb{1}\{A_w Z + U = d\}, \mathbb{1}\{W = w\}\}$:

$$
\begin{aligned}
\mathbb{E}[q_{W,1}(D, V) \mid D = d, Z, W = w] &= \mathbb{E}[q_{W,1}(d, V) \mid D = d, Z, W = w] \\
&= \mathbb{E}[q_{W,1}(d, V) \mid D = d, W = w] \\
&= \mathbb{E}[q_{W,1}(D, V) \mid D = d, W = w]
\end{aligned}
$$

Thus:

$$\mathbb{E}[\tilde{D}_W - \mu_{\tilde{A}_W}(D) \mid D, Z, W] = 0$$

From this we conclude that for any feasible solution $\tilde{e}_W, \tilde{f}_W, \tilde{A}_W$, we have that the objective can be decomposed as:

$$\mathbb{E}[\|\tilde{e}_{W,D}(X) - \tilde{A}_W Z\|^2] = \mathbb{E}[\|\tilde{D}_W - \mu_{\tilde{A}_W}(D)\|^2] + \mathbb{E}[\|\mu_{\tilde{A}_W}(D) - \tilde{A}_W Z\|^2]$$
$$= \mathbb{E}[\|q_{W,1}(D, V) - \mu_{\tilde{A}_W}(D)\|^2] + \mathbb{E}[\|\mu_{\tilde{A}_W}(D) - \tilde{A}_W Z\|^2]$$

Suppose that with positive probability, we have that $q_{W,1}(D, V) \neq \mu_{\tilde{A}_W}(D) = \tilde{A}_W A_W^+ D$. Then we have that:

$$\mathbb{E}[\|q_{W,1}(D, V) - \mu_{\tilde{A}_W}(D)\|^2] > 0$$

In this case, we will provide an alternative feasible solution, which achieves smaller objective than $\tilde{e}_W, \tilde{f}_W, \tilde{A}_W$. Consider the solution:

$$\tilde{e}'_W(x) = (\tilde{A}_W A_W^+ e_{W,D}(x), e_{W,V}(x))$$
$$\tilde{f}'_W(d, v) = f((\tilde{A}_W A_W^+)^{-1} d, v)$$

Note that we used the fact that for any feasible solution, we have already shown that $\tilde{A}_W A_W^+$ is invertible. Moreover, note that for this solution we have that, $\tilde{e}'_W, \tilde{f}'_W$ is invertible, since $e_W, f_W$ is invertible and $\tilde{A}_W A_W^+$ is invertible. Finally,

$$\tilde{f}'_W \circ \tilde{e}'_W(x) = x$$
$$\tilde{e}'_W(f(d, v)) = (\tilde{A}_W A_W^+ e_{W,D}(f_W(d, v)), e_{W,V}(f_W(d, v))) = (\tilde{A}_W A_W^+ d, v)$$

Thus:

$$\tilde{D}_W = \tilde{e}'_{W,D}(X) = \tilde{e}'_{W,D}(f_W(D, V))$$
$$= \tilde{A}_W A_W^+ D$$
$$= \tilde{A}_W A_W^+ A_W Z + \tilde{A}_W A_W^+ U$$
$$= \tilde{A}_W Z + \tilde{A}_W A_W^+ U$$

Where we used the fact that we have already shown (in the proof of Theorem D.7) that $\tilde{A}_w A_w^+ A_w = \tilde{A}_w$ for all $w \in \mathcal{W}$. Thus, we also have that:

$$\tilde{D}_W = \tilde{A}_W Z + \tilde{U}_W$$

where $\tilde{U}_W = \tilde{A}_W A_W^+ U$ and satisfies $\tilde{U}_W \perp\!\!\!\perp Z \mid W$ and $\mathbb{E}[\tilde{U}_W \mid W] = 0$.

Therefore, this new solution is a feasible solution. Moreover, since under this solution we have that $\tilde{D}_W = \tilde{A}_W A_W^+ D = \mu_{\tilde{A}_W}(D)$, the first part of the objective vanishes and the objective takes the value:

$$\mathbb{E}[\|\mu_{\tilde{A}_W}(D) - \tilde{A}_W Z\|^2] < \mathbb{E}[\|q_{W,1}(D, V) - \mu_{\tilde{A}_W}(D)\|^2] + \mathbb{E}[\|\mu_{\tilde{A}_W}(D) - \tilde{A}_W Z\|^2]$$

contradicting the optimality of the original solution.

Thus we have derived that for any optimal feasible solution, it must hold that with probability 1:

$$\tilde{D}_W = q_{W,1}(D, V) = \mu_{\tilde{A}_W}(D) = \tilde{A}_W A_W^+ D \tag{15}$$

with $\tilde{A}_W A_W^+$ an invertible matrix. Moreover, this implies that $\tilde{U}_W = \tilde{D}_W - \tilde{A}_W Z = \tilde{A}_W A_W^+ U$. $\qquad\square$

*Proof of Lemma D.8.* In this proof, we argue about the properties of the second part of the function $q_W$. Note that since $\tilde{D}_W \perp\!\!\!\perp \tilde{V}_W \mid W$ and since $\tilde{D}_W = P_W D$, for some invertible $P_W$, with probability 1, we have that $\tilde{V}_W \perp\!\!\!\perp D \mid W$. Thus:

$$q_{w,2}(D, V) \equiv \tilde{V}_w \perp\!\!\!\perp D \mid W = w$$

Since, $D \perp\!\!\!\perp V \mid W$, this implies that $\text{Law}(q_{w,2}(d, V) \mid W = w) = \text{Law}(q_{w,2}(d', V) \mid W = w)$ for all $d, d' \in \mathcal{D}_w$.

Fixed $W = w$. Since $\tilde{e}_w$ is a bijection when restricted to inputs that are outputs of $f_w$ and since $f_w$ is an injection, we have that $\tilde{e}_w \circ f_w$ is an injection. Thus there exists a well-defined inverse function $q_w^{-1} = e_w \circ \tilde{f}_w$, such that $D, V = q_w^{-1}(D, V)$. Let $q_{w,2}^{-1}$ be the $V$ component of its output. Since $D \perp\!\!\!\perp V \mid W$ and $\tilde{D}_W = P_W D$, we have that:

$$V \equiv q_{w,2}^{-1}(\tilde{D}_w, \tilde{V}_w) \perp\!\!\!\perp \tilde{D} \mid W = w$$

Since $\tilde{D}_W \perp\!\!\!\perp \tilde{V}_W \mid W$, this implies that $\text{Law}(q_{w,2}^{-1}(d, \tilde{V}_w) \mid W = w) = \text{Law}(q_{w,2}^{-1}(d', \tilde{V}_w) \mid W = w)$ for all $d, d' \in \tilde{\mathcal{D}}_w$ $\qquad\square$

### E.3. Proof of Positive Improvement

*Proof of Theorem D.10.* In this proof, we show that intervention in the direction of average derivatives of $\tilde{h}_W$ guarantees positive improvement for sufficiently small $\alpha$, assuming that $h_W$ is twice differentiable. If we perform the intervention $\tilde{D}_W + \alpha u_W$, then we have by Lemma D.8 that:

$$D_{\alpha u_W}, V_{\alpha u_W} = (D + \alpha P^{-1} u_W, q_2^{-1}(\tilde{D}_W + \alpha u_W, \tilde{V}_W))$$

Since $\mathcal{D}_w = \mathbb{R}^r$ and since $P_w$ is an invertible matrix, we have that $\tilde{\mathcal{D}}_w = \mathbb{R}^r$ for every $w \in \mathcal{W}$. Thus, for all $d \in \tilde{\mathcal{D}}_W$, we also have that $d + a u_W \in \tilde{\mathcal{D}}$. By Lemma C.1, have that for all $d \in \tilde{\mathcal{D}}_w$:

$$\text{Law}(q_{W,2}^{-1}(d, \tilde{V}_W) \mid W) = \text{Law}(q_{W,2}^{-1}(d + \alpha u_W, \tilde{V}_W) \mid W)$$

By Theorem D.7, we have that, with probability 1, $\tilde{D}_W = P_W D$ and $\tilde{U}_W = P_W U$. Moreover, by assumption, we have that $\tilde{V}_W \perp\!\!\!\perp \tilde{U}_W \perp\!\!\!\perp Z \mid W$, which implies

$$\tilde{V}_W \perp\!\!\!\perp \{\tilde{A}_W Z + \tilde{U}_W, P^{-1} \tilde{U}_W\} \mid W \implies \tilde{V}_W \perp\!\!\!\perp \{\tilde{D}_W, U\} \mid W$$

we also have that:

$$\text{Law}(q_{W,2}^{-1}(d, \tilde{V}_W) \mid U, W = w, \tilde{D}_W = d) = \text{Law}(q_{W,2}^{-1}(d + \alpha u_W, \tilde{V}_W) \mid U, W = w, \tilde{D}_W = d)$$
$$\implies \text{Law}(q_{W,2}^{-1}(\tilde{D}_W, \tilde{V}_W) \mid \tilde{D}_W, U, W = w) = \text{Law}(q_{W,2}^{-1}(\tilde{D}_W + \alpha u_W, \tilde{V}_W) \mid \tilde{D}_W, U, W = w)$$

which by the definition of $V$ and $V_\alpha$ is equivalent to:;

$$\text{Law}(V \mid \tilde{D}_W, U, W) = \text{Law}(V_{\alpha u} \mid \tilde{D}_W, U, W) \implies \text{Law}(V \mid U, W) = \text{Law}(V_{\alpha u} \mid U, W)$$

By the outcome structural equation

$$Y = h_W(D) + \eta(U, V, \epsilon)$$

we have that:

$$Y_{\alpha u} = h_W(D + \alpha P_W^{-1} u_W) + \eta(U, V_{\alpha u_W}, \epsilon_Y)$$

and that:

$$\mathbb{E}[Y_{\alpha u_W} - Y \mid W] = \mathbb{E}[h_W(D + \alpha P_W^{-1} u_W) - h_W(D) \mid W] + \mathbb{E}[\eta(U, V_{\alpha u_W}, \epsilon) - \eta(U, V, \epsilon) \mid W]$$

Since $\epsilon \perp\!\!\!\perp \{Z, U, V\} \mid W$ and since $V_{\alpha u_W}$ is a measurable function of these random variables, we have that $\epsilon \perp\!\!\!\perp \{V, V_{\alpha u}, U\} \mid W$. Letting $\tilde{\eta}(u, v) = \mathbb{E}_\epsilon[\eta(u, v, \epsilon) \mid W]$, we can write:

$$\mathbb{E}[Y_{\alpha u} - Y \mid W] = \mathbb{E}[h_W(D + \alpha P_W^{-1} u_W) - h_W(D) \mid W] + \mathbb{E}[\tilde{\eta}(U, V_{\alpha u_W}) - \tilde{\eta}(U, V) \mid W]$$
$$= \mathbb{E}[h_W(D + \alpha P_W^{-1} u_W) - h_W(D) \mid W] + \mathbb{E}[\mathbb{E}[\tilde{\eta}(U, V_{\alpha u}) - \tilde{\eta}(U, V) \mid U, W]] \mid W]$$
$$= \mathbb{E}[h_W(D + \alpha P_W^{-1} u) - h_W(D) \mid W] \qquad (\text{Law}(V \mid U, W) = \text{Law}(V_{\alpha u} \mid U, W))$$

By a first-order Taylor expansion and since $h$ is twice differentiable with bounded first and second derivatives:

$$\mathbb{E}[Y_{\alpha u} - Y \mid W] = \mathbb{E}[\alpha \nabla_D h_W(D)^\top P^{-1} u_W \mid W] + O(\alpha^2)$$
$$= \alpha \|(P_W^{-1})^\top \mathbb{E}[\nabla_D h_W(D) \mid W]\| + O(\alpha^2)$$

$\qquad\square$

# F. Semi-Synthetic Experiment Details

This section provides a detailed description of the semi-synthetic experiment based on real-world EHR data from a major hospital, corresponding to Section 5.

## F.1. Data Source and Cohort

The original dataset consists of 7,200 pneumonia patients admitted through the Emergency Department (ED) at a major hospital. The ED setting provides a structural advantage for causal inference: because patients are typically assigned to the next available attending physician based on shift timing rather than medical complexity, the assignment of a Physician ID ($Z$) can be reasonably treated as a quasi-random instrument. This minimizes the risk of selection bias and strengthens the validity of the Physician ID as an instrument.

Due to the sensitive nature of the patient records, the data are private. Researchers interested in access for replication purposes should contact the authors to discuss data-sharing agreements.

## F.2. Variable Construction

From the raw EHR data, we constructed the following variables for our semi-synthetic experiment:

**Instrument (Z):**   The instrument is derived from the attending Physician ID assigned to each patient in the ED. The raw data contains a large number of physicians, many of whom have treated only a few patients in the cohort. To create a more stable instrument, we group physicians who have treated fewer than 10 patients into a single `no_pref` category. The final instrument $Z$ is a one-hot encoded vector of these 475 physician IDs.

**Baseline Covariates (V):**   To account for baseline patient health status, we mapped diagnosis codes from the Electronic Health Records (EHR) to the Elixhauser Comorbidity Index, using these categories as nuisance covariates ($V$).

**High-Dimensional Treatment (X):**   The raw treatment vector is initially composed of 6,393 unique clinical codes, including specific medication identifiers (e.g., `abacavir 300 mg`) and procedural codes (e.g., `ultrasonography`). To mitigate feature sparsity and clinical redundancy, we manually aggregated these codes into 815 semantically distinct categories based on therapeutic class and procedural similarity. The final treatment matrix $X$ is represented as a binary indicator matrix across these 815 consolidated features.

## F.3. Synthetic Outcome Generation

To create a ground-truth causal effect, we generate a synthetic outcome variable, $Y$, representing the patient's recovery time in hours. The outcome is designed to be a function of the patient's baseline health and the treatments they receive, following a clinically-informed structural equation.

$$Y = \text{Baseline} + \gamma f_{\text{sickness}}(V) + f_{\text{treatment}}(X) + \mathcal{N}(0, \sigma^2) \tag{16}$$

**1. Baseline Sickness Score ($f_{\textbf{sickness}}(V)$):**   A patient's total sickness score is defined as the sum of the weights of their pre-existing comorbidities. Specifically, the baseline patient sickness is calculated as a weighted sum of the Elixhauser categories $V$. We diverge from the standard Elixhauser Index weights in favor of a pneumonia-specific scheme, assigning higher values to comorbidities—such as examples in Table 3. The additional scaler weight $\gamma$ controls for the confounding strength.

**2. Treatment Effect Score ($f_{\textbf{treatment}}(X)$):**   We define the causal effect of different treatments by assigning a negative weight (representing a reduction in recovery time) to a curated list of common pneumonia treatments in $X$ as listed in Table 4. The weights are chosen to reflect clinical reality, where broad-spectrum antibiotics and steroids have a larger effect than symptomatic treatments. The total treatment effect for a patient is the sum of the effects of all treatments they received.

**3. Final Outcome ($Y$):**   The final outcome $Y$ is calculated by starting with a baseline recovery time of 48 hours, adding the patient's sickness score, and incorporating the total treatment effect. To ground the simulation in realistic clinical variance,

*Table 3.* Example of Pneumonia-Specific Weighting for Elixhauser Comorbidity Categories.

| Clinical Theme | Comorbidity Variable | Weight |
|---|---|---|
| **Respiratory & Hemodynamic** | Chronic Lung Disease (`LUNG_CHRONIC`) | 22 |
| | Heart Failure (`HF`) | 20 |
| | Pulmonary Circulation (`PULMCIRC`) | 18 |
| | Valvular Disease (`VALVE`) | 12 |
| | Hypertension with Complications (`HTN_CX`) | 6 |
| **Immune & Metabolic Exhaustion** | Metastatic Cancer (`CANCER_METS`) | 25 |
| | AIDS/HIV (`AIDS`) | 20 |
| | Leukemia/Lymphoma (`CANCER_LEUK`) | 15 |
| | Weight Loss/Malnutrition (`WGHTLOSS`) | 12 |
| | Diabetes with Complications (`DIAB_CX`) | 12 |
| | Alcohol Abuse (`ALCOHOL`) | 10 |
| | Diabetes Uncomplicated (`DIAB_UNCX`) | 5 |
| **Organ Failure (Kidney/Liver)** | Severe Liver Disease (`LIVER_SEV`) | 20 |
| | Severe Renal Failure (`RENLFL_SEV`) | 18 |
| | Moderate Renal Failure (`RENLFL_MOD`) | 10 |
| | Mild/Moderate Liver Disease (`LIVER_MLD`) | 7 |
| **Neurological (Aspiration Risk)** | Dementia (`DEMENTIA`) | 15 |
| | Paralysis (`PARALYSIS`) | 15 |
| | Cerebrovascular Disease (`CBVD_POA`) | 12 |
| | Neurological/Seizures (`NEURO_SEIZ`) | 8 |
| **Systemic Stress** | Coagulation Deficiency (`COAG`) | 10 |
| | Deficiency Anemia (`ANEMDEF`) | 5 |
| **Low Pneumonia Impact** | Obesity (`OBESE`) | 3 |
| | Peptic Ulcer Disease (`ULCER_PEPTIC`) | 3 |
| | Hypothyroidism (`THYROID_HYPO`) | 2 |
| | Depression (`DEPRESS`) | 2 |
| | Psychoses (`PSYCHOSES`) | 2 |

we add a noise term $\alpha \cdot \epsilon$, where $\epsilon$ is sampled from a Kernel Density Estimation (KDE) fitted to the empirical Length of Stay (LOS) residuals from the original EHR records. We apply a scaling factor $\alpha$ to align the noise magnitude with our causal factors. This setup allows the outcome to inherit the characteristic heavy right-tail of real-world patient data—representing outlier complications—while maintaining a signal-to-noise ratio suitable for evaluating representation recovery. The final value is clipped at zero to ensure non-negative recovery times.

**4. Mechanistic Sample Generation Process:** To clarify the process, the generation of a single data point $(Z_i, V_i, X_i, Y_i)$ for a synthetic patient $i$ proceeds as follows:

1. **Generate Covariates ($V_i$):** A synthetic covariate vector $V_i$ is generated by sampling from a multivariate normal fitted with the global mean and covariate. This ensures that the distribution of comorbidities is realistic, but critically, it is generated independently of the physician assignment.
2. **Generate Instrument ($Z_i$):** A physician ID is randomly drawn from the set of real physician IDs, and the corresponding one-hot vector is assigned as the instrument $Z_i$.
3. **Generate Treatment ($X_i$):** The treatment vector $X_i$ is generated from a probabilistic model that depends on both $Z_i$ and $V_i$. First, we compute the average treatment probability for each physician from the real data (the *physician style*). This forms the base probability of treatment. Then, we add a *sickness nudge* based on the synthetic patient's comorbidities $W_i$ (e.g., a patient with chronic lung disease is more likely to receive respiratory medication). An example is summarized in Table 5. This sum forms the final treatment probability, and the binary vector $X_i$ is drawn from a Bernoulli distribution with these probabilities. This step explicitly introduces the confounding, as treatment

*Table 4.* Treatment Effects on Pneumonia Recovery Time. Effects represent the reduction in recovery time (hours) associated with each clinical intervention in the semi-synthetic outcome generation.

| Clinical Category | Treatment Variable (Representative Code) | Effect ($\Delta$ Hours) |
|---|---|---|
| **Antibiotics** | Levofloxacin (`levofloxacin_po`) | -10.0 |
| | Azithromycin (`azithromycin_po`) | -8.0 |
| **Steroids & Anti-inflammatory** | Prednisone (`prednisone_po`) | -6.0 |
| | Dexamethasone (`dexamethasone_sodium_phosphate_inj`) | -5.5 |
| | Dexamethasone Oral (`dexamethasone_po`) | -5.0 |
| | Ibuprofen (`ibuprofen_po`) | -1.0 |
| | Aspirin (`aspirin_po`) | -1.0 |
| **Respiratory Support** | Ipratropium-Albuterol (`ipratropium-albuterol_unspecified`) | -4.5 |
| | Albuterol Sulfate (`albuterol_sulfate_unspecified`) | -4.0 |
| | Nebulizer Therapy (`hc_stat_nebulizer`) | -3.5 |
| | Inhalation Treatment (`hc_inhalation_tx`) | -3.0 |
| | Metered Dose Inhaler (`hc_metered_dose`) | -2.0 |
| **Supportive Care & Anticoagulants** | Heparin IV (`heparin_iv`) | -5.0 |
| | Heparin Injection (`heparin_inj`) | -4.0 |
| | Dextrose IV (`dextrose_iv`) | -2.0 |
| | Therapy/IVP (`hc_therapypro/dx_ivp`) | -2.0 |
| **Symptomatic** | Acetaminophen (`acetaminophen_po`) | -1.0 |
| | Stat Therapist Treatment (`hc_stat:thrpst_trtmnt`) | -1.0 |
| | Guaifenesin (`guaifenesin_po`) | -0.5 |
| **Diagnostics & Monitoring** | Blood Culture (`hc_blood_culture`) | -1.5 |
| | Venipuncture/CBC/Metabolic Panel (`hc_cbc_w/auto`) | -1.0 |
| | Arterial Blood Gas/CO2 (`hc_pc02-02_(cg4)`) | -0.8 |
| | Lactate/Sepsis Monitoring (`hc_pclact-lactate_(cg4)`) | -0.5 |
| | Chest X-Ray (`hc_chest_2v`) | -0.5 |

choice $X_i$ is influenced by both the instrument $Z_i$ and the confounder $W_i$.

4. **Generate Outcome ($Y_i$):** The patient's baseline sickness score, $S_i$, is calculated from their synthetic covariates $V_i$. The total treatment effect, $T_i$, is calculated from their synthetic treatment vector $X_i$. The final outcome is then computed as $Y_i = \max(0, 48 + \gamma S_i + T_i + \epsilon_i)$, where $\epsilon_i$ is drawn from a right-tail distribution.

This process yields a complete semi-synthetic dataset where the confounding structure is explicitly modeled and the ground-truth causal effects are known, allowing for rigorous evaluation.

*Table 5.* Logic for Sickness-Driven Treatment Nudges. For each patient, the probability of receiving a treatment is increased by up to 99% if they possess the corresponding driving comorbidity.

| Treatment Category | Driving Comorbidities ($V$) | Affected Treatments ($X$) |
|---|---|---|
| **Respiratory** | `LUNG_CHRONIC, HF, PULMCIRC, VALVE` | Nebulizers, Albuterol Sulfate, Ipratropium-Albuterol |
| **Antibiotics** | `AIDS, CANCER_METS, CANCER_LEUK, WGHTLOSS` | Levofloxacin, Azithromycin, Ceftriaxone |
| **Steroids** | `LUNG_CHRONIC, AUTOIMMUNE` | Prednisone, Dexamethasone |

### F.4. Evaluation

To quantify the downstream utility of the learned representations, we simulate a treatment optimization task.

For the latent-space models, we first extract a $k$-dimensional representation ($k = 5$). For the PCA baseline, we use the top $k$ principal components; for our proposed LIRR, we use the learned representation $\tilde{D}$ as described in Algorithm 2. In both cases, we identify the causal direction $\hat{\theta}$ by performing 2SLS regression on the latent space.

For direct-regression models, we fitted Ordinary Least Squares (OLS), LASSO regression, and Partial Least Squares (PLS) regression fitted on the full high-dimensional treatment vector $X$ as non-instrumental baselines.

From each fitted model, we then construct a perturbed treatment vector $X'$ by shifting the original treatment toward the identified causal direction. For latent models, which utilize a decoding matrix $\hat{B}$, the perturbation is defined as:

$$X' = \mathbb{I}(X - \alpha \hat{B}^\top \hat{\theta} > \tau)$$

For direct regression models (OLS/LASSO), the perturbation is performed in the same way without decoding factor. Here, $\alpha = 10$ is a scaling factor, and $\tau = 0.2$ is the decision threshold for treatment activation. We evaluate the improvement in recovery time by calculating the difference between the noise-free ground-truth outcomes $Y(X')$ and $Y(X)$.

# G. Linear Experiment

We evaluate the performance of LIRR (Algorithm 2) under a high-dimensional linear data-generating process (DGP) characterized by unobserved structural confounding and high measurement noise. In addition, we provided examples with independence violations.

## G.1. Data Generating Process

We evaluate the models across three distinct DGP that meet the structural assumptions:

- *DGP 1 (Ideal Baseline):* The latent confounder $U$ and the instrument-independent measurement noise $V$ are drawn from independent standard Gaussian distributions. This represents an idealized setup where the noise layers do not possess independent internal structural correlations.

- *DGP 2 (Structured Latent Confounding):* The components of the treatment confounder $U$ are derived from a correlated Uniform distribution mapped via a mixing matrix $E$. This simulates scenarios where multi-dimensional hidden traits collectively influence the latent treatment assignation.

- *DGP 3 (Fully Structured Confounding and Noise):* Confounder $U$ remains structured as in DGP 2, while the high-dimensional noise component $V$ is drawn from a highly correlated Gaussian distribution mapped via a mixing matrix $F$. The dominant variance directions in the observed space are heavily confounded with the target outcome.

The categorical configurations (DGPs A–D) introduce dependency violations between instrument $Z$ and confounders $U, V$:

- *DGP A (Confounder-to-Instrument Leakage):* The latent confounder $U$ back-propagates directly into the observed instruments $Z$.

- *DGP B (Instrument-to-Noise Confounding):* The instrument directly drives variations within the orthogonal noise subspace $V$.

- *DGP C (Confounder-to-Noise Leakage):* Models a link between the confounder $U$ and $V$.

- *DGP D (Joint Interaction):* Enforces a multiplicative, non-linear interaction ($U \odot Z$) that feeds directly into the measurement noise field $V$, violating joint independence.

## G.2. Structural Parameters

For each DGP, we also tested the following parameters ($V = v$ indicates the parameter $V$ is fixed at the given value $v$):

- **Instrument Strength** ($\gamma_A = 1.0, \gamma_u$)**:** Controls the Signal-to-Confounder ratio in the latent space $D$. As $\gamma_A \to 0, \gamma_u \to \infty$, the latent representation becomes structurally dominated by the confounder $U$, creating a weak instrument regime.

- **Observed Treatment Strength** ($\gamma_B = 1.0, \gamma_v$)**:** Scales the relative magnitude of the causal projection subspace ($B$) against the non-causal noise subspace ($B_\perp V$) within the observed variable $X$. As $\gamma_B \to 0, \gamma_v \to \infty$, the true treatment will be dominated by noise $V$.

- **Feature Dimension** ($m = 100$)**:** The dimensionality of the observed treatment vector $X_i \in \mathbb{R}^m$.

- **Confounding Intensity** ($\nu_u = \nu_v = -10.0$)**:** Weighting coefficients mapping the hidden subspaces to the outcome $Y$. Specifically, $\nu_u$ dictates latent treatment confounding (motivating the use of Instrumental Variables), and $\nu_v$ dictates noise-subspace confounding (motivating Instrument-Guided Representation Learning).

- **Dependence Strength** ($\rho = 5$)**:** Dictates the cross-subspace leakage magnitude across the structural violations introduced in DGPs A–D.

The generating equations are detailed below:

**Linear DGP 1**

Draw DGP parameters:

$$A \sim \{\mathcal{N}(0, \gamma_A^2)\}^{r \times k}, \qquad W \sim \{\mathcal{N}(0,1)\}^{m \times m}, \qquad \theta \sim \{\mathcal{N}(0,1)\}^{r \times 1}$$

$$B = \gamma_B * Q[:,:r] \in \mathbb{R}^{m \times r}, \qquad B_\perp = Q[:,r:] \in \mathbb{R}^{m \times (m-r)} \qquad \text{where } Q, R = \text{qr}(W)$$

Draw Noise:

$$U_i \sim \gamma_u \mathcal{N}(0, I_r) \qquad \text{(Confounder for } D)$$
$$V_i \sim \gamma_v \mathcal{N}(0, I_{m-r}) \qquad \text{(Confounder for } X)$$
$$\epsilon_i \sim \mathcal{N}(0,1) \qquad \text{(Irreducible Error)}$$
$$\eta_i(U_i, V_i) = \eta_u \sum U_i + \eta_v \sum V_i + \epsilon_i \qquad \text{(Confounder for } Y)$$

Construct Variables:

$$Z_i \sim \mathcal{N}(0, I_k) \qquad \text{(Instrument)}$$
$$D_i = AZ_i + U_i \qquad \text{(Latent Treatment)}$$
$$X_i = BD_i + B_\perp V_i \qquad \text{(Observed Treatment)}$$
$$Y_i = \theta^\top D_i + \eta_i(U_i, V_i) \qquad \text{(Outcome)}$$

With dimensions $n = 10000$, $r = k = 4$, and $m = 100$.

---

**Linear DGP 2: Structured $U$**

Draw additional DGP parameters:

$$E \sim \mathcal{N}(0, 2^2)^{r \times h}$$

Draw updated noise:

$$\mathcal{U}_i \sim \gamma_u \text{Uniform}(-1,1)^{h \times 1}, \qquad U_i = E\mathcal{U}_i \in \mathbb{R}^r \qquad \text{(Confounder for } D)$$

With $h = 3$. Everything else remains the same as DGP 1.

---

**Linear DGP 3: Structured $U$, $V$**

Draw additional DGP parameters:

$$E \sim \mathcal{N}(0, 2^2)^{r \times h} \qquad\qquad F \sim \mathcal{N}(0, 0.5^2)^{(m-r) \times h_2}$$

Draw updated noise:

$$\mathcal{U}_i \sim \gamma_u \text{Uniform}(-1,1)^{h \times 1}, \qquad U_i = E\mathcal{U}_i \in \mathbb{R}^r \qquad \text{(Confounder for } D)$$
$$\mathcal{V}i \sim \gamma_v \mathcal{N}(0, I_{h_2}), \qquad V_i = F\mathcal{V}_i \in \mathbb{R}^{m-r} \qquad \text{(Confounder for } X)$$

With $h = 3$, $h_2 = 5$. Everything else remains the same as DGP 1.

---

**Structural Dependency Violations (DGPs A–D)**

For the following formulations, baseline distributions are configured with standard components: $\tilde{Z}_i \sim \mathcal{N}(0, I_k)$, $\tilde{U}_i \sim \gamma_u \cdot \mathcal{N}(0, I_r)$, and $\tilde{V}_i \sim \gamma_v \cdot \mathcal{N}(0, I_{m-r})$. Let $G$ specify a cross-subspace coupling matrix populated via $\rho \cdot \mathcal{N}(0, I)$.

**Linear DGP A (Confounder $\rightarrow$ Instrument Leakage):** Draw $G \sim \{\mathcal{N}(0, \rho^2)\}^{k \times r}$:

$$U_i = \tilde{U}_i, \qquad\qquad Z_i = \tilde{Z}_i + GU_i, \qquad\qquad V_i = B_\perp \tilde{V}_i$$

**Linear DGP B (Instrument $\rightarrow$ Noise Confounding):** Draw $G \sim \{\mathcal{N}(0, \rho^2)\}^{(m-r) \times k}$:

$$U_i = \tilde{U}_i, \qquad\qquad Z_i = \tilde{Z}_i, \qquad\qquad V_i = B_\perp \left( \tilde{V}_i + GZ_i \right)$$

**Linear DGP C (Confounder $\rightarrow$ Noise Leakage):** Draw $G \sim \{\mathcal{N}(0, \rho^2)\}^{(m-r) \times r}$:

$$U_i = \tilde{U}_i, \qquad\qquad Z_i = \tilde{Z}_i, \qquad\qquad V_i = B_\perp \left( \tilde{V}_i + GU_i \right)$$

**Linear DGP D (Non-linear Instrument-Confounder Interaction):** Requires $r = k$. Draw $G \sim \{\mathcal{N}(0, \rho^2)\}^{(m-r) \times r}$:

$$U_i = \tilde{U}_i, \qquad\qquad Z_i = \tilde{Z}_i, \qquad\qquad V_i = B_\perp \left( \tilde{V}_i + G(U_i \odot Z_i) \right)$$

where $\odot$ represents the element-wise (Hadamard) vector product.

## G.3. Evaluation

As a baseline, we use PCA to extract the top $k = 4$ components of $X$ as a latent representation, followed by 2SLS using $Z$ as the instrument to identify the intervention direction. LIRR uses the same procedure with its instrument-aligned representation. For both methods, the perturbed treatment is computed using

$$\tilde{X}_{\alpha u} = X + \alpha \hat{B} u$$

where $u$ and $\hat{B}$ are the normalized learned 2SLS coefficients and decoding matrix respectively. The remaining methods (OLS and PLS) learn a direction on the original $X$-space, and hence

$$\tilde{X}_{\alpha u} = X + \alpha u$$

where $u$ is the normalized coefficients for each corresponding methods.

To determine the true outcome after perturbation, we used the true generating formula

$$Y_{\alpha u} = \theta^T (B^\dagger X_{\alpha u}).$$

Each experiment was repeated 100 times with different random seeds, each containing a sample size of 10000 with 80-20 train-test split.

## G.4. Results

The results are included in Table 6, 7. LIRR consistently yields positive outcome improvements across the standard environments (DGPs 1–3). We discuss the results by the structural parameters tested:

**Instrument Strength (DGPs 1–3):** A reduction in instrument strength (increasing $\gamma_u$) leads to higher variance for all methods, spanning zero in some cases. This shows that all methods suffer from weak instruments.

**Observed Treatment Strength (DGPs 1–3):** For the same instrument strength ($\gamma_u$), PCA performance degrades as $\gamma_v$ increases. This is expected: if the observed treatment mostly contains instrument-relevant information, naive PCA will recover the relevant dimension. In particular, under confounded distributions (DGP 1, DGP 2), the performance of PCA can be comparable to LIRR in some settings.

**High-Dimensional Scaling:** While feature dimensionality was kept fixed across these tracking runs, the evaluated setting ($m = 100$) represents a significantly underdetermined regime relative to the true treatment dimension ($k = 4$), presenting an inherently challenging subspace isolation task.

**Confounding Intensity (DGPs 1–3):** To stress-test the methods against heavy, directional omitted-variable bias, we set the confounding magnitudes to ($\nu_u = \nu_v = -10.0$). Consequently, both OLS and PLS fail, as they do not account for confounding.

*Table 6.* Results for Linear DGPs 1–3 ($m = 100$, $\nu_u = \nu_v = -10.0$, $\rho = 0$).

| $\gamma_u$ | $\gamma_v$ | Method | Improvement $\Delta Y$ ($\pm$ SD) | | |
| --- | --- | --- | --- | --- | --- |
| | | | DGP 1 | DGP 2 | DGP 3 |
| 0.1 | 0.1 | LIRR | **0.67 $\pm$ 0.34** | **0.67 $\pm$ 0.34** | **0.68 $\pm$ 0.34** |
| | | OLS | 0.02 $\pm$ 0.09 | 0.02 $\pm$ 0.10 | 0.09 $\pm$ 0.14 |
| | | PCA | 0.54 $\pm$ 0.36 | 0.57 $\pm$ 0.35 | 0.36 $\pm$ 0.38 |
| | | PLS | 0.03 $\pm$ 0.07 | 0.03 $\pm$ 0.09 | 0.09 $\pm$ 0.11 |
| | 1.0 | LIRR | **1.13 $\pm$ 1.10** | **1.05 $\pm$ 1.08** | **0.97 $\pm$ 1.01** |
| | | OLS | 0.05 $\pm$ 0.06 | 0.04 $\pm$ 0.12 | 0.15 $\pm$ 0.30 |
| | | PCA | 0.16 $\pm$ 0.39 | 0.21 $\pm$ 0.47 | 0.02 $\pm$ 0.03 |
| | | PLS | 0.04 $\pm$ 0.04 | 0.04 $\pm$ 0.04 | 0.11 $\pm$ 0.13 |
| | 10.0 | LIRR | **2.13 $\pm$ 7.10** | **2.27 $\pm$ 6.87** | **1.92 $\pm$ 6.38** |
| | | OLS | 0.40 $\pm$ 0.51 | 0.38 $\pm$ 1.05 | 1.08 $\pm$ 2.56 |
| | | PCA | 0.01 $\pm$ 0.03 | 0.01 $\pm$ 0.02 | 0.00 $\pm$ 0.01 |
| | | PLS | 0.16 $\pm$ 0.20 | 0.17 $\pm$ 0.21 | 0.28 $\pm$ 0.70 |
| 1.0 | 0.1 | LIRR | **0.71 $\pm$ 0.39** | **0.71 $\pm$ 0.57** | **0.82 $\pm$ 0.48** |
| | | OLS | 0.02 $\pm$ 0.10 | 0.01 $\pm$ 0.17 | 0.04 $\pm$ 0.34 |
| | | PCA | **0.71 $\pm$ 0.39** | **0.70 $\pm$ 0.58** | 0.78 $\pm$ 0.50 |
| | | PLS | 0.03 $\pm$ 0.12 | 0.03 $\pm$ 0.26 | 0.04 $\pm$ 0.35 |
| | 1.0 | LIRR | **1.12 $\pm$ 1.12** | **1.02 $\pm$ 1.20** | **1.13 $\pm$ 1.08** |
| | | OLS | 0.04 $\pm$ 0.12 | 0.03 $\pm$ 0.21 | 0.07 $\pm$ 0.55 |
| | | PCA | 0.67 $\pm$ 0.90 | 0.58 $\pm$ 1.03 | 0.06 $\pm$ 0.13 |
| | | PLS | 0.04 $\pm$ 0.12 | 0.03 $\pm$ 0.19 | 0.06 $\pm$ 0.53 |
| | 10.0 | LIRR | **2.12 $\pm$ 7.09** | **2.26 $\pm$ 6.89** | **1.99 $\pm$ 6.47** |
| | | OLS | 0.30 $\pm$ 1.00 | 0.25 $\pm$ 1.60 | 0.41 $\pm$ 3.96 |
| | | PCA | 0.01 $\pm$ 0.04 | 0.01 $\pm$ 0.04 | 0.00 $\pm$ 0.01 |
| | | PLS | 0.26 $\pm$ 0.53 | 0.16 $\pm$ 0.82 | 0.24 $\pm$ 2.23 |
| 10.0 | 0.1 | LIRR | **1.33 $\pm$ 2.18** | **0.97 $\pm$ 3.93** | **1.29 $\pm$ 3.73** |
| | | OLS | 0.07 $\pm$ 0.58 | 0.03 $\pm$ 1.51 | 0.00 $\pm$ 2.84 |
| | | PCA | **1.33 $\pm$ 2.23** | 0.90 $\pm$ 3.95 | 1.20 $\pm$ 3.88 |
| | | PLS | 0.08 $\pm$ 1.30 | 0.06 $\pm$ 3.00 | 0.01 $\pm$ 3.01 |
| | 1.0 | LIRR | **1.45 $\pm$ 2.24** | **0.97 $\pm$ 3.89** | **1.58 $\pm$ 3.41** |
| | | OLS | 0.05 $\pm$ 0.63 | 0.03 $\pm$ 1.57 | 0.06 $\pm$ 2.88 |
| | | PCA | **1.47 $\pm$ 2.55** | 0.81 $\pm$ 3.45 | 0.66 $\pm$ 2.05 |
| | | PLS | 0.06 $\pm$ 0.71 | 0.12 $\pm$ 1.70 | -0.03 $\pm$ 2.89 |
| | 10.0 | LIRR | **1.90 $\pm$ 7.25** | **2.39 $\pm$ 8.02** | **2.10 $\pm$ 7.80** |
| | | OLS | 0.14 $\pm$ 1.83 | 0.20 $\pm$ 2.30 | 0.30 $\pm$ 5.95 |
| | | PCA | 0.33 $\pm$ 1.64 | 0.37 $\pm$ 2.74 | 0.25 $\pm$ 1.02 |
| | | PLS | 0.14 $\pm$ 1.83 | 0.03 $\pm$ 1.82 | -0.10 $\pm$ 5.61 |

*Table 7.* Results for DGPs A–D ($m = 100$, $\gamma_u = \gamma_v = 0.1$, $\nu_u = \nu_v = 0.0$, and $\rho = 5.0$).

| Method | Improvement $\Delta Y$ ($\pm$ SD) | | | |
| --- | --- | --- | --- | --- |
| | DGP A | DGP B | DGP C | DGP D |
| LIRR | $0.93 \pm 0.46$ | $0.70 \pm 0.41$ | $1.66 \pm 0.69$ | $1.65 \pm 0.69$ |
| OLS | $\mathbf{1.15 \pm 0.66}$ | $\mathbf{15.65 \pm 7.82}$ | $\mathbf{1.77 \pm 0.88}$ | $\mathbf{1.78 \pm 0.89}$ |
| PCA | $0.72 \pm 0.50$ | $0.70 \pm 0.41$ | $0.03 \pm 0.06$ | $0.03 \pm 0.07$ |
| PLS | $1.05 \pm 0.53$ | $11.41 \pm 6.06$ | $1.55 \pm 0.69$ | $1.56 \pm 0.70$ |

**Dependency Violations:** LIRR performance under the structural dependency violation regimes (DGPs A–D) exhibits smaller improvements, as anticipated given the intentional violations. In fact, in DGPs A–B, $Z$ is no longer a valid instrument for $D$, and with no unobserved confounding ($\nu_u = \nu_v = 0$), standard observed method OLS and PLS perform better trivially.

# H. Quadratic Experiment with DeepIV

To highlight the advantages of instrument-guided dimensionality reduction, we compare IRAE against DeepIV (Hartford et al., 2017) applied to the raw high-dimensional features. Furthermore, we evaluate a hybrid IRAE+DeepIV approach—applying a non-parametric second stage to our learned representations—to determine if it outperforms a standard 2SLS final stage. This hybrid configuration is motivated by the potential for residual non-linearities in the mapping between recovered and true latent factors. To capture these effects, we employ a quadratic mapping $f$ from latent components $(D, V)$ to the observed treatment $X$, while maintaining a linear outcome function $h(D) = \theta^\top D$ as per Equation (1). The noise scenarios follow the linear settings, with the instrument dimension fixed at $k = 4$.

## H.1. DGP

The data generating process closely mirrors the linear setting, with the extension of incorporating quadratic terms. We did not evaluate settings involving instrument violations (e.g., DGP A–D) again, as they do not constitute a valid instrumental variable data generating process. See the linear experiments for intuitions in such settings.

---

**Quadratic DGP 1**

Draw DGP parameters:

$$A \sim \{\mathcal{N}(0, \gamma_A^2)\}^{r \times k}, \qquad W \sim \{\mathcal{N}(0,1)\}^{m \times m}, \qquad \theta \sim \{\mathcal{N}(0,1)\}^{r \times 1}$$

$$B = \gamma_B \cdot Q[:,:p] \in \mathbb{R}^{m \times p}, \qquad B_\perp = Q[:,p:] \in \mathbb{R}^{m \times (m-p)} \qquad \text{where } Q, R = \text{qr}(W)$$

Draw Noise:

$$U_i \sim \gamma_u \mathcal{N}(0, I_r) \qquad\qquad \text{(Confounder for } D\text{)}$$
$$V_i \sim \gamma_v \mathcal{N}(0, I_{m-p}) \qquad\qquad \text{(Confounder for } X\text{)}$$
$$\epsilon_i \sim \mathcal{N}(0,1) \qquad\qquad \text{(Irreducible Error)}$$
$$\eta_i(U_i, V_i) = \eta_u \sum U_i + \eta_v \sum V_i + \epsilon_i \qquad\qquad \text{(Confounder for } Y\text{)}$$

Construct Variables:

$$Z_i \sim \mathcal{N}(0, I_k) \qquad\qquad \text{(Instrument)}$$
$$D_i = A Z_i + U_i \qquad\qquad \text{(Latent Treatment)}$$
$$X_i = B \Phi(D_i) + B_\perp V_i \qquad\qquad \text{(Observed Non-linear Treatment)}$$
$$Y_i = \theta^\top D_i + \eta_i(U_i, V_i) \qquad\qquad \text{(Outcome)}$$

Where $\Phi(D_i)$ denotes the polynomial expansion vector of degree 2, dimension dimensions are $n = 10000$, $r = k = 4$, and $m = 100$, with $p$ representing the expanded polynomial feature dimension.

---

**Quadratic DGP 2: Structured $U$**

Draw additional DGP parameters:

$$E \sim \{\mathcal{N}(0, 2^2)\}^{r \times h}$$

Draw updated noise:

$$\mathcal{U}_i \sim \gamma_u \text{Uniform}(-1,1)^{h \times 1}, \qquad\qquad U_i = E \mathcal{U}_i \in \mathbb{R}^r \qquad \text{(Confounder for } D\text{)}$$

With $h = 3$. Everything else remains identical to Quadratic DGP 1.

---

---

**Quadratic DGP 3: Structured $U, V$**

Draw additional DGP parameters:

$$E \sim \{\mathcal{N}(0, 2^2)\}^{r \times h} \qquad\qquad F \sim \{\mathcal{N}(0, 0.5^2)\}^{(m-p) \times h_2}$$

Draw updated noise:

$$\mathcal{U}_i \sim \gamma_u \text{Uniform}(-1, 1)^{h \times 1}, \qquad U_i = E\mathcal{U}_i \in \mathbb{R}^r \qquad \text{(Confounder for } D\text{)}$$
$$\mathcal{V}_i \sim \gamma_v \mathcal{N}(0, I_{h_2}), \qquad V_i = F\mathcal{V}_i \in \mathbb{R}^{m-p} \qquad \text{(Confounder for } X\text{)}$$

With $h = 3$, $h_2 = 5$. Everything else remains identical to Quadratic DGP 1.

---

### H.2. Models

In this benchmark, we test the following models as shown in Table 8.

*Table 8.* Regularization Configurations Across IRAE Method Variants

| Method Key | *latent_v* | $\lambda$ | $\mu_1$ | $\mu_2$ | $\mu_3$ | $c_{1,2,3}$ |
|---|---|---|---|---|---|---|
| Vanilla AE | 0 | 0.0 | 0.0 | 0.0 | 0.0 | 0.0 |
| Outcome AE | N/A | N/A | N/A | N/A | N/A | N/A |
| iVAE | N/A | N/A | N/A | N/A | N/A | N/A |
| IRAE-0 | 0 | 1.0 | 0.0 | 0.0 | 0.0 | 0.0 |
| IRAE-1 | 0 | 1.0 | 1.0 | 0.0 | 0.0 | 0.0 |
| IRAE-medium-2 | 16 | 1.0 | 1.0 | 1.0 | 0.0 | 0.0 |
| IRAE-small | 8 | 1.0 | 1.0 | 1.0 | 1.0 | 0.1 |
| IRAE-medium | 16 | 1.0 | 1.0 | 1.0 | 1.0 | 0.1 |
| IRAE-large | 24 | 1.0 | 1.0 | 1.0 | 1.0 | 0.1 |
| DeepIV | NA | NA | NA | NA | NA | NA |
| IRAE-DeepIV | 16 | 1.0 | 1.0 | 1.0 | 1.0 | 0.1 |

Below, we include an illustration of the model structures benchmarked in this study:

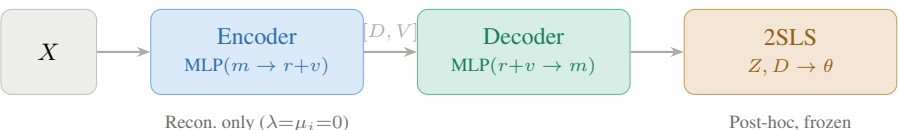

Vanilla AE minimizes reconstruction loss exclusively.

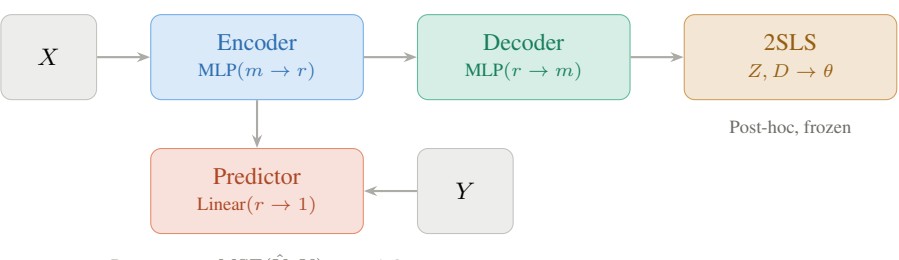

**Outcome AE** jointly minimizes reconstruction loss and downstream prediction loss from the learned representation to the outcome, representing an outcome-guided representation learning process.

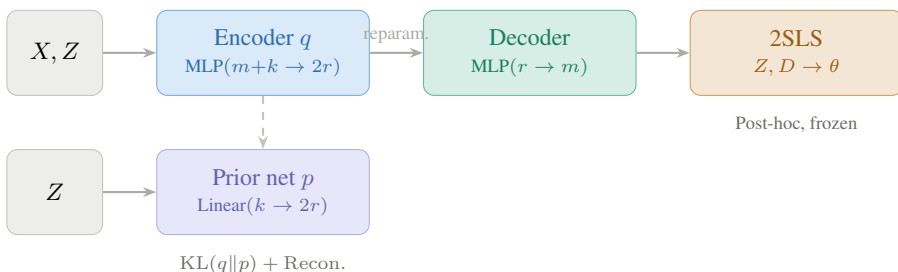

**iVAE** maximizes the conditional likelihood $p(X|Z)$ by incorporating the instrument $Z$ into both the encoding and decoding processes. While incorpering $Z$ in encoding and decoding process help learned instrument relevant dimension, in practice, $Z$ may not be available in test time.

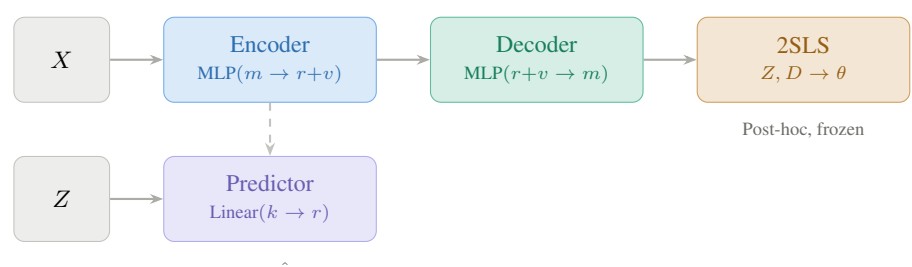

**IRAE**, as introduced in the main paper, jointly minimizes reconstruction loss and prediction loss from the instrument to the learned representation alongside targeted independence regularizations. Each of the numbered IRAE model is obtained with different hyperparameter, as described in Table 8.

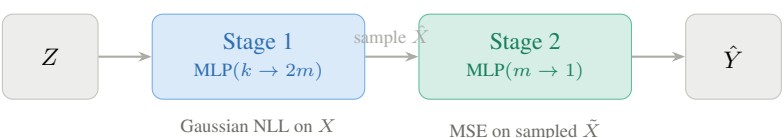

**DeepIV** implements the deep instrumental variable framework described by (Hartford et al., 2017).

**IRAE-DeepIV** refers to the model that applies DeepIV onto the representations learned by the IRAE model (as opposed to 2SLS).

All encoder and decoder networks, including DeepIV's treatment net and outcome net, consist of a linear layer projecting to a hidden size of 64, followed by LayerNorm and LeakyReLU(0.2), and a final linear projection to the specified output dimension. The bottleneck dimension is $r +$ `latent_v` where $r = 4$ is the true causal latent dimension. The treatment input dimension $m = 50$, corresponding to a linear transformation of second-degree polynomial features from 4 raw features. The iVAE encoder additionally conditions on the instrument $Z$, giving an input dimension of 54. its decoder takes only the latent $D$. The predictor (used in IRAE and Outcome AE) is a single linear layer. See Table 9 for remaining training hyperparameters.

*Table 9.* Architecture and training hyperparameters

| Model | Networks | Architecture | Optimizer | LR / Epochs / Patience |
|---|---|---|---|---|
| Vanilla AE | Encoder, Decoder | `MLPBlock` | AdamW | 5e-3 / 60 / 15 |
| Outcome AE | Encoder, Decoder, Predictor[†] | `MLPBlock` | AdamW | 5e-3 / 60 / 15 |
| iVAE | Encoder $q$, Decoder, Prior net[†] | `MLPBlock` | AdamW | 3e-3 / 60 / 10 |
| IRAE | Encoder, Decoder, Predictor[†] | `MLPBlock` | AdamW | 3e-3 / 60 / 10 |
| DeepIV | Treatment Net (Q-net) | `MLPBlock` | Adam | 1e-3 / 60 / 10 |
| | Outcome Net (O-net) | `MLPBlock` | Adam | 1e-3 / 60 / 10 |
| IRAE-DeepIV | Stage 1: IRAE | Same as above | | |
| | Stage 2: Deep IV | Same as above | | |

[†]: `nn.Linear` — no hidden layer, no activation.

`MLPBlock`s: `Linear`($d_{\text{in}}$→64) → `LayerNorm`(64) → `LeakyReLU`(0.2) → `Linear`(64→$d_{\text{out}}$).

## H.3. Evaluation

To determine the true outcome after perturbation, we used the formula

$$Y_{\alpha u} = \theta^T((B^\dagger X_{\alpha u})[:r]),$$

where $[:r]$ index into the first-order terms (excluding the quadratic and cross terms) of $D$.

## H.4. Results

Under low regularizing conditions ($\gamma_u = 0.1, \gamma_v = 0.1$), IRAE-small and IRAE-DeepIV significantly outperform alternative models in terms of the improvement metric ($\Delta Y$). The superior performance of IRAE-small under low $\gamma_v$ is intuitive: when the confounding influence of $V$ is minor, a lower-capacity bottleneck is sufficient, avoiding an overly complex latent space that could otherwise destabilize training. The strong performance of IRAE-DeepIV indicates the presence of remaining non-linearities not captured by the smaller variants.

Similar to the linear setting, scaling the regularization strength parameters up to $\gamma_u = 1.0$ and/or $\gamma_v = 1.0$ causes the standard deviations (SD) of almost all IRAE variants to increase significantly, often spanning zero. While the variance of standard DeepIV also increases under these conditions, its SD remains relatively small—though this comes at the cost of a severely limited total improvement. This could also be due to underspecified model. To ensure a fair baseline comparison, we restricted the MLPBlock to a simple two-layer linear network for all models, which is slightly underspecified. The Linear Instrumental Variable baseline (LIRR) follows similar trend: it consistently yields positive improvements. However, it fails to achieve the high peak performance of the IRAE models due to structural architecture mismatch against the quadratic DGP.

In contrast, standard representation frameworks (Vanilla AE, iVAE, and Outcome AE) consistently fail to show meaningful improvements, with $\Delta Y$ hovering between 0 and 1. This starkly demonstrates that optimizing purely for data reconstruction or raw downstream outcome prediction without explicit instrumental bounds fails to isolate unobserved confounding.

*Table 10.* Quadratic DGP 1–3 Results ($m = 50$, $\nu_u = \nu_v = -10.0$, and $\rho = 0$).

| $\gamma_u$ | $\gamma_v$ | Method | Improvement $\Delta Y$ ($\pm$ SD) | | |
|---|---|---|---|---|---|
| | | | DGP 1 | DGP 2 | DGP 3 |
| 0.1 | 0.1 | LIRR | $3.22 \pm 1.93$ | $3.31 \pm 1.89$ | $3.25 \pm 1.92$ |
| | | VANILLA_AE | $0.16 \pm 0.55$ | $0.36 \pm 1.17$ | $0.04 \pm 0.76$ |
| | | OUTCOME_AE | $0.67 \pm 0.88$ | $0.15 \pm 0.70$ | $0.35 \pm 0.82$ |
| | | IVAE | $0.86 \pm 0.79$ | $0.82 \pm 0.72$ | $0.98 \pm 0.73$ |
| | | IRAE-0 | $0.81 \pm 2.78$ | $1.00 \pm 3.03$ | $1.16 \pm 2.89$ |
| | | IRAE-1 | $0.58 \pm 4.09$ | $0.78 \pm 3.14$ | $1.29 \pm 2.23$ |
| | | IRAE-MEDIUM-2 | $5.41 \pm 5.76$ | $6.07 \pm 5.68$ | $6.67 \pm 5.86$ |
| | | IRAE-SMALL | $\mathbf{11.87 \pm 11.18}$ | $\mathbf{9.37 \pm 8.60}$ | $\mathbf{8.06 \pm 7.92}$ |
| | | IRAE-MEDIUM | $6.93 \pm 6.82$ | $5.52 \pm 3.93$ | $7.95 \pm 7.48$ |
| | | IRAE-LARGE | $4.57 \pm 4.79$ | $5.57 \pm 5.26$ | $3.93 \pm 3.28$ |
| | | IRAE-DEEPIV | $\mathbf{8.86 \pm 5.41}$ | $\mathbf{9.55 \pm 5.81}$ | $\mathbf{9.84 \pm 7.11}$ |
| | | DEEPIV | $2.24 \pm 1.91$ | $2.32 \pm 1.93$ | $2.20 \pm 2.01$ |
| 0.1 | 1.0 | LIRR | $2.67 \pm 2.23$ | $2.69 \pm 2.23$ | $2.45 \pm 2.48$ |
| | | VANILLA_AE | $0.21 \pm 1.00$ | $-0.02 \pm 0.78$ | $0.38 \pm 1.08$ |
| | | OUTCOME_AE | $0.20 \pm 0.56$ | $0.05 \pm 0.58$ | $-0.05 \pm 0.52$ |
| | | IVAE | $0.51 \pm 0.74$ | $0.43 \pm 0.61$ | $0.50 \pm 0.66$ |
| | | IRAE-0 | $0.94 \pm 3.25$ | $0.57 \pm 2.18$ | $0.68 \pm 3.43$ |
| | | IRAE-1 | $0.89 \pm 3.28$ | $0.49 \pm 2.78$ | $0.25 \pm 2.21$ |
| | | IRAE-MEDIUM-2 | $5.53 \pm 8.61$ | $1.94 \pm 3.06$ | $\mathbf{4.13 \pm 5.72}$ |
| | | IRAE-SMALL | $\mathbf{6.03 \pm 9.15}$ | $4.13 \pm 6.88$ | $2.18 \pm 4.34$ |
| | | IRAE-MEDIUM | $\mathbf{6.10 \pm 7.82}$ | $\mathbf{5.46 \pm 8.26}$ | $\mathbf{3.68 \pm 5.11}$ |
| | | IRAE-LARGE | $4.51 \pm 6.32$ | $5.12 \pm 6.01$ | $3.06 \pm 4.78$ |
| | | IRAE-DEEPIV | $4.69 \pm 9.67$ | $\mathbf{5.62 \pm 8.26}$ | $3.13 \pm 7.18$ |
| | | DEEPIV | $0.92 \pm 1.19$ | $0.82 \pm 0.90$ | $0.87 \pm 1.14$ |
| 1.0 | 0.1 | LIRR | $3.64 \pm 2.43$ | $\mathbf{5.01 \pm 3.29}$ | $4.93 \pm 3.57$ |
| | | VANILLA_AE | $0.52 \pm 0.72$ | $0.19 \pm 1.03$ | $0.25 \pm 0.87$ |
| | | OUTCOME_AE | $0.33 \pm 0.64$ | $0.21 \pm 0.46$ | $0.15 \pm 0.59$ |
| | | IVAE | $0.66 \pm 0.95$ | $0.36 \pm 0.61$ | $0.54 \pm 0.65$ |
| | | IRAE-0 | $0.64 \pm 2.38$ | $2.48 \pm 8.35$ | $1.56 \pm 6.14$ |
| | | IRAE-1 | $1.51 \pm 6.19$ | $2.03 \pm 5.71$ | $2.58 \pm 6.45$ |
| | | IRAE-MEDIUM-2 | $5.46 \pm 7.03$ | $1.56 \pm 3.97$ | $2.13 \pm 3.85$ |
| | | IRAE-SMALL | $\mathbf{15.31 \pm 15.66}$ | $\mathbf{5.49 \pm 12.32}$ | $\mathbf{11.68 \pm 15.45}$ |
| | | IRAE-MEDIUM | $6.26 \pm 7.23$ | $2.80 \pm 5.64$ | $3.16 \pm 6.46$ |
| | | IRAE-LARGE | $4.33 \pm 4.99$ | $-0.36 \pm 2.89$ | $1.88 \pm 4.50$ |
| | | IRAE-DEEPIV | $\mathbf{7.45 \pm 12.15}$ | $4.98 \pm 7.32$ | $\mathbf{7.84 \pm 9.08}$ |
| | | DEEPIV | $2.47 \pm 1.85$ | $3.70 \pm 2.65$ | $3.68 \pm 2.95$ |
| 1.0 | 1.0 | LIRR | $3.15 \pm 2.63$ | $3.98 \pm 3.41$ | $\mathbf{4.14 \pm 3.97}$ |
| | | VANILLA_AE | $0.07 \pm 0.62$ | $0.43 \pm 1.03$ | $0.09 \pm 0.94$ |
| | | OUTCOME_AE | $0.20 \pm 0.65$ | $0.20 \pm 0.57$ | $0.08 \pm 0.57$ |
| | | IVAE | $0.53 \pm 0.73$ | $0.29 \pm 0.53$ | $0.31 \pm 0.35$ |
| | | IRAE-0 | $0.72 \pm 2.52$ | $0.79 \pm 4.41$ | $1.19 \pm 5.14$ |
| | | IRAE-1 | $0.22 \pm 3.87$ | $-0.42 \pm 6.79$ | $0.91 \pm 4.33$ |
| | | IRAE-MEDIUM-2 | $\mathbf{9.87 \pm 11.45}$ | $2.69 \pm 5.33$ | $1.68 \pm 6.23$ |
| | | IRAE-SMALL | $8.29 \pm 10.58$ | $\mathbf{4.82 \pm 13.46}$ | $3.43 \pm 9.69$ |
| | | IRAE-MEDIUM | $\mathbf{12.26 \pm 13.15}$ | $\mathbf{4.44 \pm 7.11}$ | $\mathbf{5.01 \pm 5.90}$ |
| | | IRAE-LARGE | $8.23 \pm 10.37$ | $0.87 \pm 2.94$ | $1.58 \pm 3.68$ |
| | | IRAE-DEEPIV | $5.02 \pm 15.48$ | $3.71 \pm 11.61$ | $3.89 \pm 12.56$ |
| | | DEEPIV | $1.66 \pm 1.47$ | $2.79 \pm 2.51$ | $2.67 \pm 2.35$ |

# I. MNIST Experiment 1

This section provides details of the MNIST experiments briefly described in Section 5 of the main paper. Here we included detailed data generating equations, model hyperparameter, and plots for IRAE[0], IRAE[1], IRAE[2] that were not included in the main paper.

The data for MNIST experiment is generated using *Case 1 DGP*.

---

**Case 1 DGP**

Draw DGP parameters $\alpha, \beta \sim \text{Unif}(0.1, 0.7)$. Then generate samples as:

$$G_i \in [0, 1]^{28 \times 28} \qquad \text{(grayscale MNIST image)}$$

$$Z_i, \ U_i \sim \mathcal{N}(0, I_2), \qquad Z_i \perp\!\!\!\perp U_i \qquad \text{(instrument \& confounder)}$$

$$r_i = \text{clip}(0.5 + \alpha \, Z_{i1} + \beta \, U_{i1}, \, 0, \, 1) \qquad \text{(red channel)}$$

$$g_i = \text{clip}(0.5 + \alpha \, Z_{i2} + \beta \, U_{i2}, \, 0, \, 1) \qquad \text{(green channel)}$$

$$b_i = \text{clip}\left(0.5 + \alpha \, \frac{Z_{i1} + Z_{i2}}{2}, \, 0, \, 1\right) \qquad \text{(blue channel)}$$

$$X_i(k, \ell, c) = G_i(k, \ell) \cdot (r_i, g_i, b_i)_c, \qquad \begin{matrix} c \in \{R, G, B\}, \\ (k, \ell) \in \{1, \dots, 28\}^2 \end{matrix} \qquad \text{(colour image)}$$

$$Y_i = r_i + g_i + b_i. \qquad \text{(outcome, details below)}$$

Returns the tuples $(Z_i, X_i, Y_i)$.

---

All encoders consist of three Conv2D layers, followed by additional feedforward layers, and conclude with a linear projection. Decoders mirror this architecture in reverse order. For our IRAE[2] and IRAE models, we set the bottleneck dimension to 10 which is larger than $k = 2$. For vanilla and IRAE[0], IRAE[1], the bottle neck is 2. The autoencoder with multiple HSIC regularization terms presents greater training challenges due to the complexity of term. To address this, we initialized IRAE[2] and IRAE with weights from the simpler IRAE[1] model. All of models are trained with 60k training samples and evaluated on 10k test set. More training details can be found in Table 11.

*Table 12.* Average Test Improvement Comparison of 5 Methods on MNIST Data (Mean $\pm$ Std). the reviewer wants the table to be transposed but idk if it fits the page

| Sample Size | 10000 | | | 30000 | | | 60000 | | |
|---|---|---|---|---|---|---|---|---|---|
| $\alpha$ | reconstructed(0) | intervened(0.2) | intervened(1.0) | reconstructed(0) | intervened(0.2) | intervened(1.0) | reconstructed(0) | intervened(0.2) | intervened(1.0) |
| Vanilla AE | -0.51 $\pm$ 0.03 | -0.50 $\pm$ 0.03 | -0.47 $\pm$ 0.04 | -0.51 $\pm$ 0.03 | -0.49 $\pm$ 0.03 | -0.44 $\pm$ 0.05 | -0.47 $\pm$ 0.02 | -0.46 $\pm$ 0.03 | -0.41 $\pm$ 0.06 |
| IRAE[0] | -0.73 $\pm$ 0.04 | -0.07 $\pm$ 0.15 | 0.04 $\pm$ 0.20 | -0.74 $\pm$ 0.04 | -0.11 $\pm$ 0.08 | -0.06 $\pm$ 0.12 | -0.71 $\pm$ 0.05 | -0.06 $\pm$ 0.26 | 0.05 $\pm$ 0.29 |
| IRAE[1] | -0.73 $\pm$ 0.04 | 0.07 $\pm$ 0.26 | 0.15 $\pm$ 0.29 | -0.73 $\pm$ 0.04 | -0.17 $\pm$ 0.11 | -0.13 $\pm$ 0.20 | -0.71 $\pm$ 0.04 | 0.14 $\pm$ 0.30 | 0.26 $\pm$ 0.36 |
| IRAE[2] | **-0.33 $\pm$ 0.05** | 0.72 $\pm$ 0.48 | 0.92 $\pm$ 0.47 | **-0.33 $\pm$ 0.05** | **0.38 $\pm$ 0.40** | **0.78 $\pm$ 0.43** | **-0.25 $\pm$ 0.05** | **0.98 $\pm$ 0.42** | **1.13 $\pm$ 0.44** |
| IRAE | **-0.33 $\pm$ 0.04** | **0.87 $\pm$ 0.43** | **1.10 $\pm$ 0.46** | **-0.33 $\pm$ 0.06** | 0.33 $\pm$ 0.44 | 0.69 $\pm$ 0.56 | **-0.25 $\pm$ 0.06** | 0.90 $\pm$ 0.52 | 1.05 $\pm$ 0.51 |

*Table 11.* Training Parameters for MNIST Simulations

| Parameter | Vanilla AE | IRAE[0] | IRAE[1] | IRAE[2] | IRAE |
|---|---|---|---|---|---|
| **Architecture** | | | | | |
| Kernel Size | | | 3 | | |
| Encoder channels | | | $16 \rightarrow 32 \rightarrow 64$ | | |
| Decoder channels | | | $64 \rightarrow 32 \rightarrow 16$ | | |
| Bottleneck dimension | 2 | 2 | 2 | 10 | 10 |
| **Optimization** | | | | | |
| Optimizer | | | Adam (default parameters in torch) | | |
| Learning rate | | | $1 \times 10^{-3}$ | | |
| Weight initialization | None | None | None | From IRAE[1] | From IRAE[1] |
| **Loss Weights** | | | | | |
| $\lambda$ | 0 | 10 | 10 | 10 | 10 |
| $\mu_1$ | 0 | 0 | 10 | 10 | 10 |
| $\mu_2$ | 0 | 0 | 0 | 10 | 10 |
| $\mu_3$ | 0 | 0 | 0 | 0 | 10 |
| | | | $c_1 = 0.8$, $c_2 = 0.15$, $c_3 = 0.05$ | | |
| Weights $\lambda, \mu_2, \mu_3$ took value 1 whenever non-zero, instead of 10 for experiments in the Case 2-4 DGPs. | | | | | |
| **Training Epochs (with early stopping of patience 5)** | | | | | |
| | 50 | 50 | 50 | 50* | 50* |

[*] Additional epochs after initializing with weights from IRAE[1]

*Remark* I.1 (Calculation of Color from Image). To calculate expected $Y_{\alpha u}$, we first perform 2-mean clustering on the image pixels and extract the red, green, blue values from the center of the colored cluster. Then, we take the sum of these values as $Y$. Note that is this similar to taking the average colors over the gray scale mask so the colors would be slightly smaller than the original colors. We tested the methods on the original image and the result is 0.2 smaller on average.

*Remark* I.2 (Calculation of Outcome Improvement). When calculating the outcome improvement of the intervention, take the difference between the kmeans calculation described in the previous paragraph applied to the image produced by the intervention and we subtract the outcome of the kmeans calculation when applied to the original image.

*Remark* I.3. We use a linear kernel for HSIC in order to perform benchmarking at a large scale in fast speed, which may not capture all nonlinear dependencies in this complex image representation setting. More complex independence statistics based on domain knowledge, could perhaps lead to more disentanglement, albeit they might also be harder to train. In subsequent section experiments we also examine a pairwise RBF Kernel based HSIC and we find that it does not lead to improved performance as compared to the linear kernel.

*Remark* I.4. We observe that this example does not perfectly align with the formulation in Equation (1). Here, the number of instruments is 2, which is fewer than the natural representation of D of 3 colors. We may be able to interpret the learned representation as a 2-dimensional subspace of the 3-dimensional color representation, but the mapping from Z to D is still not immediately invertible as assumed in the theory. Additionally, while our theoretical analysis assumes a mapping from color $D$ to outcome directly, our calculation employs k-means clustering on $X$ instead. Nevertheless, this example demonstrates that our method performs robustly even in settings beyond those covered by our theoretical guarantees, and offers potential future directions of theoretical investigation.

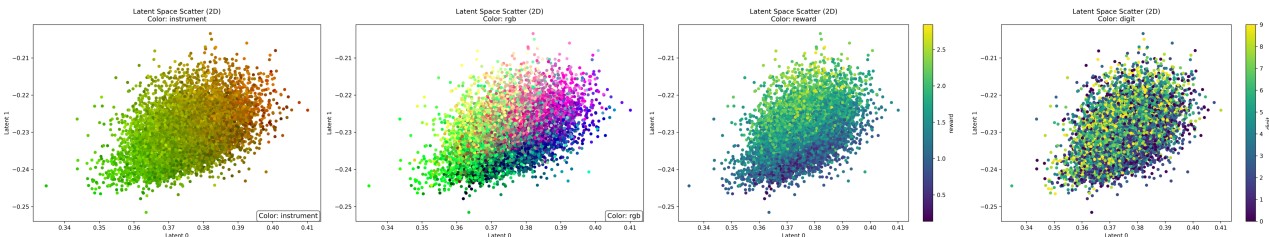

*Figure 5.* Alignment of recovered latent variables with instrument, true representation [R,G,B], reward (Y) and digit for the **IRAE** model (Case 1 DGP). Data points with similar instrument, color, and reward are grouped together in the latent space.

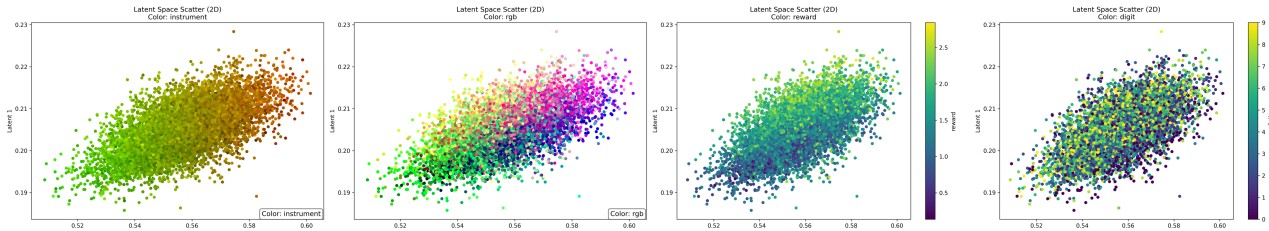

*Figure 6.* Alignment of recovered latent variables with instrument, true representation [R,G,B], reward (Y) and digit for the **IRAE[2]** model (Case 1 DGP).

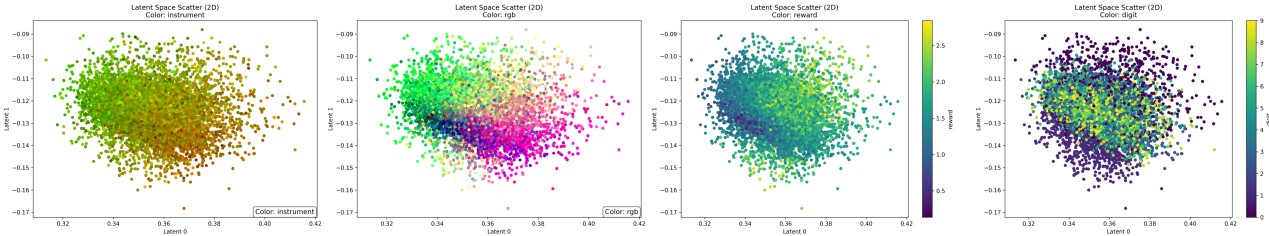

*Figure 7.* Alignment of recovered latent variables with instrument, true representation [R,G,B], reward (Y) and digit for the **IRAE[1]** model (Case 1 DGP).

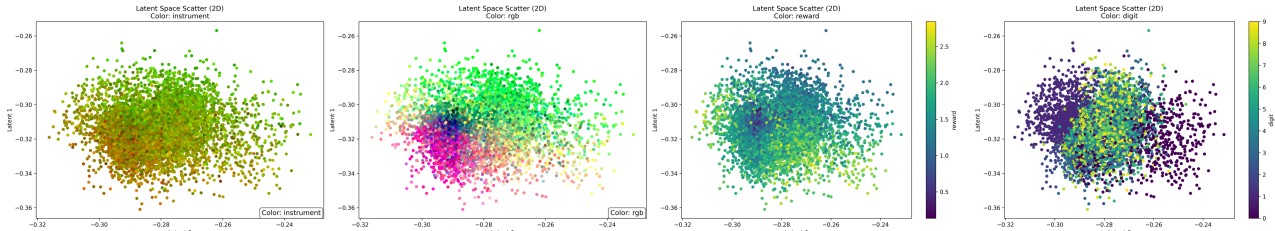

*Figure 8.* Alignment of recovered latent variables with instrument, true representation [R,G,B], reward (Y) and digit for the **IRAE[0]** model (Case 1 DGP).

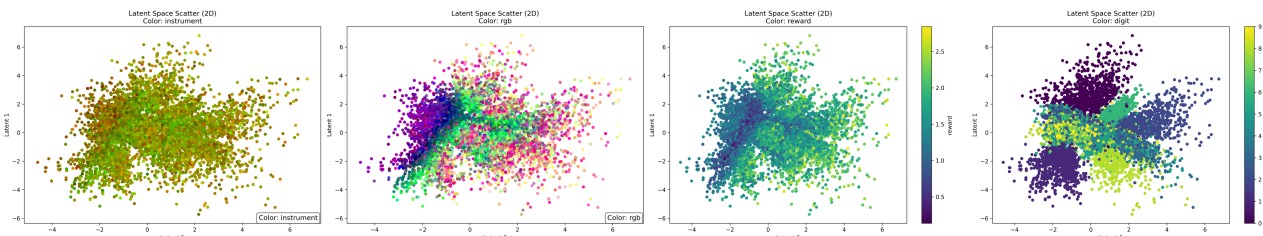

*Figure 9.* Alignment of recovered latent variables with instrument, true representation [R,G,B], reward (Y) and digit for the **Vanilla AE** model (Case 1 DGP).

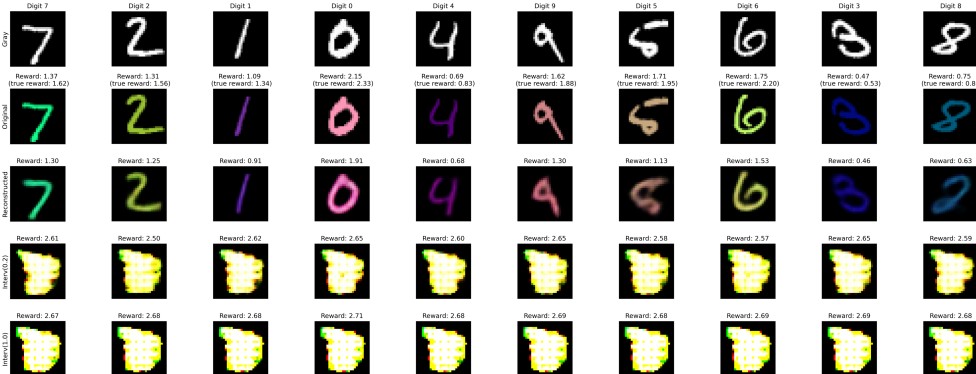

*Figure 10.* **IRAE** on Case 1 DGP for one random seed (random seed 22), with a Conv AutoEncoder, linear HSIC as independence criterion, **latent dimension 10**, regularization weights $\lambda = \mu_1 = \mu_2 = 10$ and training for 50 epochs with early stopping (patience 5 epochs) warm start from IRAE[1].

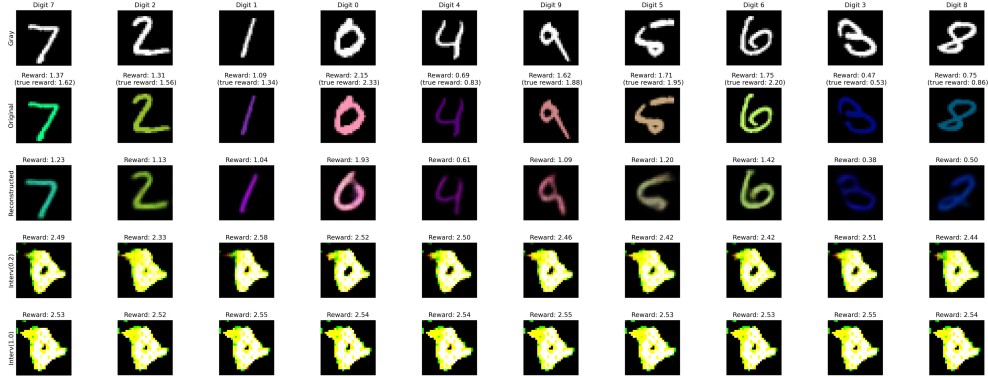

*Figure 11.* **IRAE[2]** on Case 1 DGP for one random seed (random seed 22), with a Conv AutoEncoder, linear HSIC as independence criterion, **latent dimension 10**, regularization weights $\lambda = \mu_1 = \mu_2 = 10$ and training for 50 epochs with early stopping (patience 5 epochs) warm start from IRAE[1].

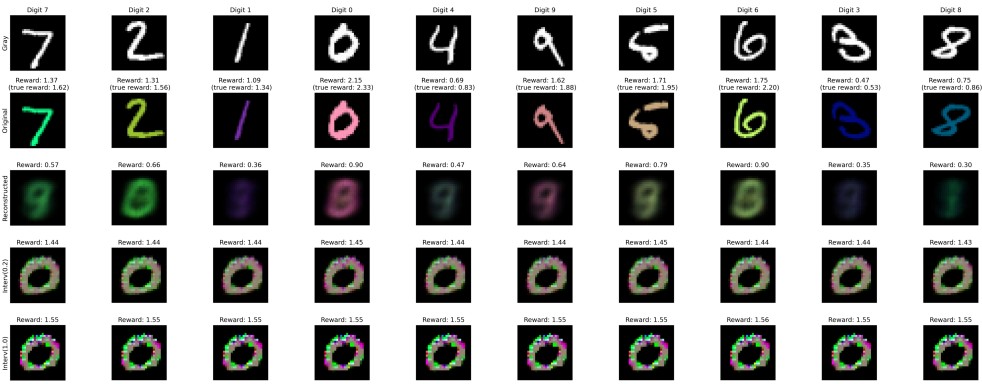

*Figure 12.* **IRAE[1]** on Case 1 DGP for one random seed (random seed 22), with a Conv AutoEncoder, linear HSIC as independence criterion, **latent dimension 2**, regularization weights $\lambda = \mu_1 = 10$ and training for 50 epochs with early stopping (patience 5 epochs) from scratch

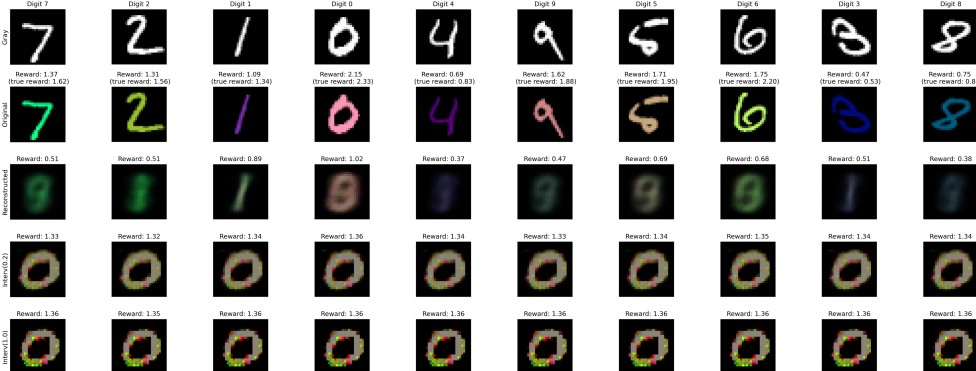

*Figure 13.* **IRAE[0]** on Case 1 DGP for one random seed (random seed 22), with a Conv AutoEncoder, linear HSIC as independence criterion, **latent dimension 2**, regularization weights $\lambda = \mu_1 = 10$ and training for 50 epochs with early stopping (patience 5 epochs) from scratch

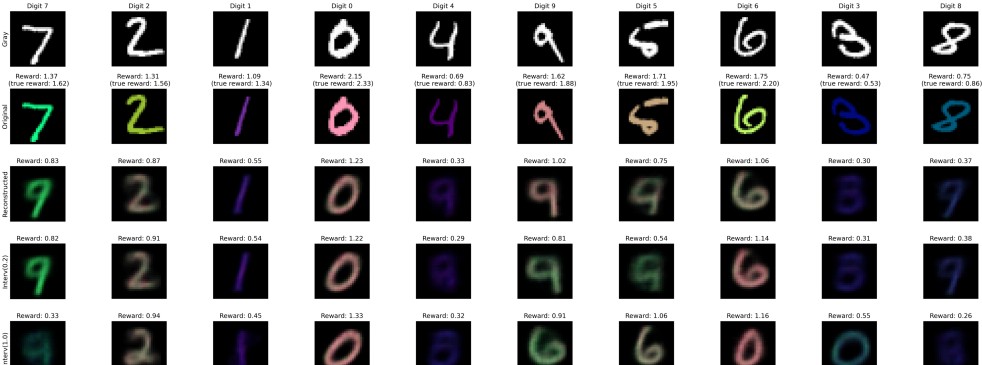

*Figure 14.* **Vanilla AE** on Case 1 DGP for one random seed (random seed 22), with a Conv AutoEncoder, **latent dimension 2**, regularization weights $\lambda = \mu_1 = 10$ and training for 50 epochs with early stopping (patience 5 epochs) from scratch

# J. Further MNIST Experiment

## J.1. MNIST Experiment 2

Building on the results from our MNIST experiments in Section 5, we conducted a more comprehensive evaluation by exploring additional hyperparameter configurations and data generating processes. Given that independence test statistics are often complex and challenging to train, we systematically investigated various model architectures, independence test statistics calculation, and initialization strategies to identify optimal configurations. To align with our theoretical requirements outlined in Theorem 2.2, we evaluated our approach on a supplementary dataset with three instruments, denoted as *Case 2 DGP*.

Our findings reveal that simpler dense architectures perform at least as well as, and often better than, more complex convolutional neural networks for this task. Furthermore, we observed that larger bottleneck dimensions in IRAE[2] and IRAE models better preserve the original digit morphology in treated images — a potentially valuable property when morphological features is confounded the outcome variable.

The full set of hyperparameters explored are included in Table 13. All of models are trained with 60k training samples and evaluated on 10k test set, for 40 random seeds. Regularization weights are 0 or 1. All models are trained with 50 epochs after initialization with early stopping of patience 5.

*Table 13.* Summary of parameters explored in MNIST Experiment 2

| Setting Category | Options | Description |
|---|---|---|
| **Data Generating Process** | DGP2 | Three Instruments |
| **Autoencoder Architecture** | Dense | **Encoder:** Dense layer $3 \times 28 \times 28 \to 512$, followed by linear projection to latent dimension 
 **Decoder:** Linear layer from latent dimension to 512, followed by dense layer $512 \to 3 \times 28 \times 28$ |
| | Convolution | **Encoder:** Three Conv2D layers with channel $16 \to 32 \to 64$ of kernel size 3, followed by a dense layer of size 256 and linear projection to latent dimension 
 **Decoder:** Linear layer from latent dimension to size 256, followed by dense layer and three Conv2D layers with channel $64 \to 32 \to 16$ of kernel size 3 |
| **Latent Dimension IRAE[2] and IRAE** | 10 
 32 | Used for IRAE[2] and IRAE models 
 Used for IRAE[2] and IRAE models |
| **Regularization Type** | Linear HSIC 
 Pairwise HSIC | Applied as independence measure on the entire vector 
 Applied between pairwise coordinates |
| **Weight Initialization IRAE[2] and IRAE** | Without warmstart 

 With warmstart | Training from randomly initialized weights for 50 epochs 
 Initializing with weights transferred from a pre-trained IRAE[1] model, and training for additional 50 epochs |

---

### Case 2 DGP

Draw DGP parameters $\alpha, \beta \sim \text{Unif}(0.1, 0.7)$. Then generate samples as:

$$G_i \in [0,1]^{28 \times 28} \qquad \text{(grayscale MNIST image)}$$

$$Z_i, \; U_i \sim \mathcal{N}(0, I_3), \qquad Z_i \perp\!\!\!\perp U_i \qquad \text{(instrument \& confounder)}$$

$$r_i = \text{clip}\big(0.5 + \alpha\, Z_{i1} + \beta\, U_{i1}, \, 0, \, 1\big) \qquad \text{(red channel)}$$

$$g_i = \text{clip}\big(0.5 + \alpha\, Z_{i2} + \beta\, U_{i2}, \, 0, \, 1\big) \qquad \text{(green channel)}$$

$$b_i = \text{clip}\big(0.5 + \alpha\, Z_{i3} + \beta\, U_{i3}, \, 0, \, 1\big) \qquad \text{(blue channel)}$$

$$X_i(k, \ell, c) = G_i(k, \ell) \cdot (r_i, g_i, b_i)_c, \qquad \begin{array}{l} c \in \{R, G, B\}, \\ (k, \ell) \in \{1, \ldots, 28\}^2 \end{array} \qquad \text{(colour image)}$$

$$Y_i = r_i + g_i + b_i. \qquad \text{(outcome)}$$

Returns the tuples $\big(Z_i, X_i, Y_i\big)$.

---

We highlight some findings from our exploration of the performance of our proposed methods across various hyperparameter dimensions:

**Architecture:** We found that simple dense layers can achieve better performance than convolutional architectures for this task, suggesting that Conv2D layers may be unnecessarily complex for this particular example.

**Data Generating Process:** Our experimental results demonstrate that the relative performance of our methods remains consistent across both DGP1 and DGP2.

**Latent Dimension:** When using larger latent dimensions (32), both the reconstructed and treated images preserved more of the original digit morphology although the improvement is smaller (c.f. Figures 16 to 21). This may be a desired property in some cases, especially in the case that the digit morphology is a confounder (not tested in our experiment) and has a direct effect on the outcome.

**Regularization Type:** While pairwise HSIC may theoretically capture more nonlinear dependencies, we found that it was often more difficult to train in practice. Linear HSIC consistently yielded better performance with greater training stability.

**Weight Initialization:** Dense architectures performed well without warm start initialization, while convolutional architectures benefited significantly from weight transfer. This difference likely stems from the higher complexity and larger parameter space of convolutional networks.

Overall, the best improvement model stems from the IRAE method with all regularizers, a Dense architecture, latent = 10, linear HSIC with no warm start.

| Arch. | Latent Dim | Reg Type | Warm Start | image | Vanilla AE | IRAE[0] | IRAE[1] | IRAE[2] | IRAE |
|---|---|---|---|---|---|---|---|---|---|
| dense | 10 | linear | False | reconstructed | -0.46 (0.02) | -0.67 (0.02) | -0.67 (0.02) | **-0.27 (0.01)** | -0.28 (0.02) |
| | | | | intervened(0.2) | -0.45 (0.02) | **1.4 (0.12)** | **1.4 (0.1)** | 1.39 (0.15) | 1.39 (0.2) |
| | | | | intervened(1.0) | -0.37 (0.02) | 1.54 (0.11) | 1.54 (0.09) | **1.57 (0.12)** | 1.54 (0.18) |
| | | | True | reconstructed | -0.46 (0.02) | -0.67 (0.02) | -0.67 (0.02) | **-0.36 (0.14)** | -0.39 (0.16) |
| | | | | intervened(0.2) | -0.45 (0.02) | **1.4 (0.12)** | **1.4 (0.1)** | 1.17 (0.53) | 1.12 (0.58) |
| | | | | intervened(1.0) | -0.37 (0.02) | **1.54 (0.11)** | **1.54 (0.09)** | 1.32 (0.5) | 1.18 (0.6) |
| | | pairwise | False | reconstructed | -0.46 (0.02) | -0.67 (0.02) | -0.68 (0.01) | **-0.3 (0.03)** | -0.35 (0.03) |
| | | | | intervened(0.2) | -0.45 (0.02) | **1.4 (0.12)** | **1.4 (0.14)** | -0.09 (0.37) | 0.12 (0.53) |
| | | | | intervened(1.0) | -0.37 (0.02) | **1.54 (0.11)** | 1.53 (0.13) | 0.09 (0.57) | 0.35 (0.58) |
| | | | True | reconstructed | -0.46 (0.02) | -0.67 (0.02) | -0.68 (0.01) | **-0.33 (0.1)** | -0.57 (0.22) |
| | | | | intervened(0.2) | -0.45 (0.02) | **1.4 (0.12)** | **1.4 (0.14)** | 1.31 (0.24) | 0.87 (0.78) |
| | | | | intervened(1.0) | -0.37 (0.02) | **1.54 (0.11)** | 1.53 (0.13) | 1.49 (0.15) | 1.12 (0.59) |
| | 32 | linear | False | reconstructed | -0.46 (0.02) | -0.67 (0.02) | -0.67 (0.02) | **-0.14 (0.02)** | **-0.14 (0.01)** |
| | | | | intervened(0.2) | -0.45 (0.02) | **1.4 (0.12)** | **1.4 (0.1)** | 0.74 (0.34) | 0.66 (0.38) |
| | | | | intervened(1.0) | -0.37 (0.02) | **1.54 (0.11)** | **1.54 (0.09)** | 1.43 (0.31) | 1.35 (0.38) |
| | | | True | reconstructed | -0.46 (0.02) | -0.67 (0.02) | -0.67 (0.02) | **-0.26 (0.12)** | **-0.26 (0.15)** |
| | | | | intervened(0.2) | -0.45 (0.02) | **1.4 (0.12)** | **1.4 (0.1)** | 1.08 (0.36) | 1.03 (0.5) |
| | | | | intervened(1.0) | -0.37 (0.02) | **1.54 (0.11)** | **1.54 (0.09)** | 1.29 (0.42) | 1.19 (0.49) |
| | | pairwise | False | reconstructed | -0.46 (0.02) | -0.67 (0.02) | -0.68 (0.01) | **-0.13 (0.01)** | -0.19 (0.02) |
| | | | | intervened(0.2) | -0.45 (0.02) | **1.4 (0.12)** | **1.4 (0.14)** | -0.15 (0.05) | -0.23 (0.09) |
| | | | | intervened(1.0) | -0.37 (0.02) | **1.54 (0.11)** | 1.53 (0.13) | -0.2 (0.18) | -0.26 (0.3) |
| | | | True | reconstructed | -0.46 (0.02) | -0.67 (0.02) | -0.68 (0.01) | **-0.19 (0.05)** | -0.31 (0.15) |
| | | | | intervened(0.2) | -0.45 (0.02) | **1.4 (0.12)** | **1.4 (0.14)** | 0.07 (0.41) | 0.02 (0.47) |
| | | | | intervened(1.0) | -0.37 (0.02) | **1.54 (0.11)** | 1.53 (0.13) | 0.42 (0.65) | 0.4 (0.68) |
| conv | 10 | linear | False | reconstructed | -0.37 (0.02) | -0.6 (0.06) | -0.6 (0.05) | **-0.21 (0.03)** | -0.22 (0.03) |
| | | | | intervened(0.2) | -0.36 (0.03) | 0.21 (0.34) | 0.4 (0.4) | **0.98 (0.23)** | 0.91 (0.32) |
| | | | | intervened(1.0) | -0.31 (0.07) | 0.4 (0.56) | 0.69 (0.58) | **1.25 (0.55)** | 1.18 (0.57) |
| | | | True | reconstructed | -0.37 (0.02) | -0.6 (0.06) | -0.6 (0.05) | **-0.2 (0.04)** | **-0.2 (0.03)** |
| | | | | intervened(0.2) | -0.36 (0.03) | 0.21 (0.34) | 0.4 (0.4) | **1.0 (0.45)** | 0.9 (0.51) |
| | | | | intervened(1.0) | -0.31 (0.07) | 0.4 (0.56) | 0.69 (0.58) | **0.89 (0.75)** | 0.64 (0.75) |
| | | pairwise | False | reconstructed | -0.37 (0.02) | -0.6 (0.06) | -0.6 (0.05) | **-0.22 (0.05)** | -0.27 (0.08) |
| | | | | intervened(0.2) | -0.36 (0.03) | 0.21 (0.34) | 0.04 (0.45) | 0.47 (0.42) | **0.49 (0.47)** |
| | | | | intervened(1.0) | -0.31 (0.07) | 0.4 (0.56) | 0.12 (0.57) | 0.86 (0.47) | **0.89 (0.55)** |
| | | | True | reconstructed | -0.37 (0.02) | -0.6 (0.06) | -0.6 (0.05) | **-0.26 (0.07)** | -0.3 (0.08) |
| | | | | intervened(0.2) | -0.36 (0.03) | 0.21 (0.34) | 0.04 (0.45) | **0.82 (0.47)** | 0.54 (0.51) |
| | | | | intervened(1.0) | -0.31 (0.07) | 0.4 (0.56) | 0.12 (0.57) | **1.08 (0.55)** | 0.75 (0.64) |
| | 32 | linear | False | reconstructed | -0.37 (0.02) | -0.6 (0.06) | -0.6 (0.05) | **-0.1 (0.03)** | -0.11 (0.04) |
| | | | | intervened(0.2) | -0.36 (0.03) | 0.21 (0.34) | 0.4 (0.4) | **0.7 (0.33)** | 0.56 (0.38) |
| | | | | intervened(1.0) | -0.31 (0.07) | 0.4 (0.56) | 0.69 (0.58) | **1.26 (0.39)** | 1.11 (0.5) |
| | | | True | reconstructed | -0.37 (0.02) | -0.6 (0.06) | -0.6 (0.05) | **-0.1 (0.03)** | **-0.1 (0.04)** |
| | | | | intervened(0.2) | -0.36 (0.03) | 0.21 (0.34) | 0.4 (0.4) | 1.05 (0.49) | **1.13 (0.38)** |
| | | | | intervened(1.0) | -0.31 (0.07) | 0.4 (0.56) | 0.69 (0.58) | **1.11 (0.57)** | 1.09 (0.64) |
| | | pairwise | False | reconstructed | -0.37 (0.02) | -0.6 (0.06) | -0.6 (0.05) | **-0.11 (0.03)** | -0.14 (0.06) |
| | | | | intervened(0.2) | -0.36 (0.03) | **0.21 (0.34)** | 0.04 (0.45) | 0.02 (0.26) | -0.02 (0.31) |
| | | | | intervened(1.0) | -0.31 (0.07) | **0.4 (0.56)** | 0.12 (0.57) | 0.21 (0.49) | 0.1 (0.55) |
| | | | True | reconstructed | -0.37 (0.02) | -0.6 (0.06) | -0.6 (0.05) | **-0.14 (0.08)** | -0.19 (0.07) |
| | | | | intervened(0.2) | -0.36 (0.03) | 0.21 (0.34) | 0.04 (0.45) | 0.4 (0.56) | **0.5 (0.45)** |
| | | | | intervened(1.0) | -0.31 (0.07) | 0.4 (0.56) | 0.12 (0.57) | 0.68 (0.68) | **0.82 (0.59)** |

*Figure 15.* Experimental results for the **Case 2** data generating process. Mean improvement and standard deviation of improvement is reported. *reconstructed* refers to the mean outcome improvement of the reconstructed image from the autoencoder with no intervention in the latents, as compared to the original image. *intervened(α)* refers to the mean outcome improvement of the image produced by intervening on the latents in direction $\alpha \cdot u$, where $u = \theta/\|\theta\|$ and $\theta$ is estimated by 2SLS in latent space.

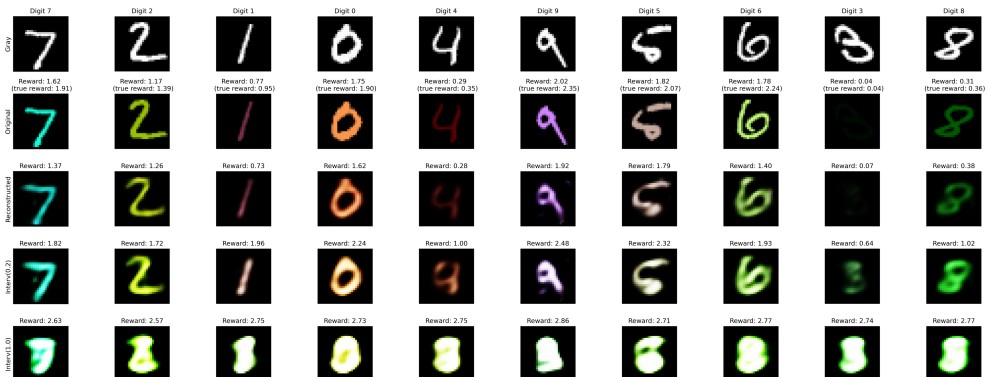

*Figure 16.* **IRAE** on Case 2 DGP for one random seed (random seed 22), with a Dense AutoEncoder, linear HSIC as independence criterion, **latent dimension 32**, regularization weights $\lambda = \mu_1 = \mu_2 = \mu_3 = 1$ and training for 50 epochs with early stopping (patience 5 epochs) from scratch (no warm start from IRAE1).

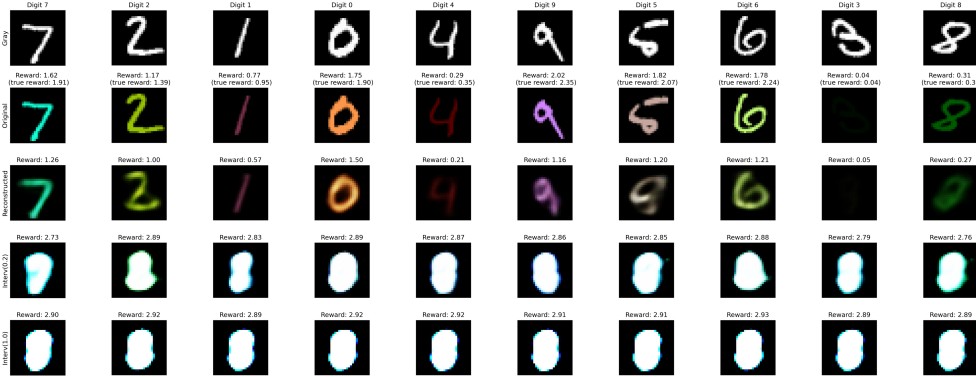

*Figure 17.* **IRAE** on Case 2 DGP for one random seed (random seed 22), with a Dense AutoEncoder, linear HSIC as independence criterion, **latent dimension 10**, regularization weights $\lambda = \mu_1 = \mu_2 = \mu_3 = 1$ and training for 50 epochs with early stopping (patience 5 epochs) from scratch (no warm start from IRAE1).

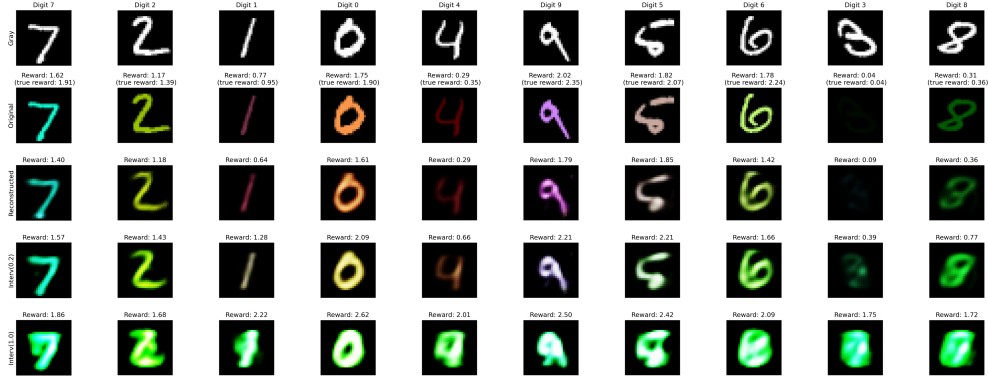

*Figure 18.* **IRAE[2]** on Case 2 DGP for one random seed (random seed 22), with a Dense AutoEncoder, linear HSIC as independence criterion, **latent dimension 32**, regularization weights $\lambda = \mu_1 = \mu_2 = 1$ and training for 50 epochs with early stopping (patience 5 epochs) from scratch (no warm start from IRAE[1]).

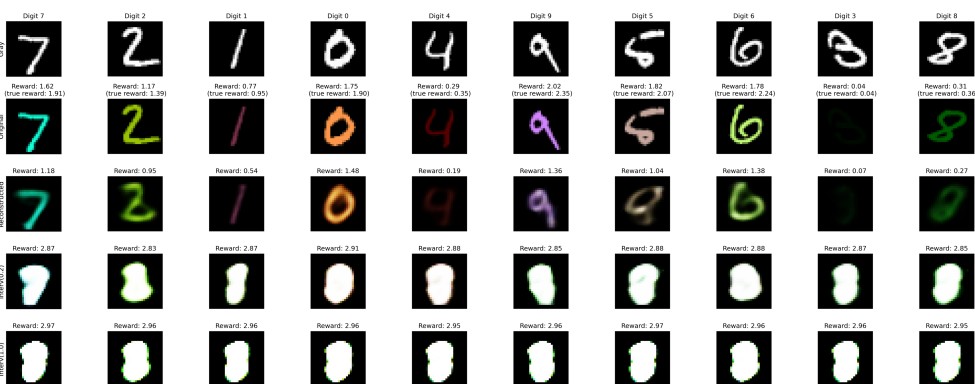

*Figure 19.* **IRAE[2]** on Case 2 DGP for one random seed (random seed 22), with a Dense AutoEncoder, linear HSIC as independence criterion, **latent dimension 10**, regularization weights $\lambda = \mu_1 = \mu_2 = 1$ and training for 50 epochs with early stopping (patience 5 epochs) from scratch (no warm start from IRAE[1]).

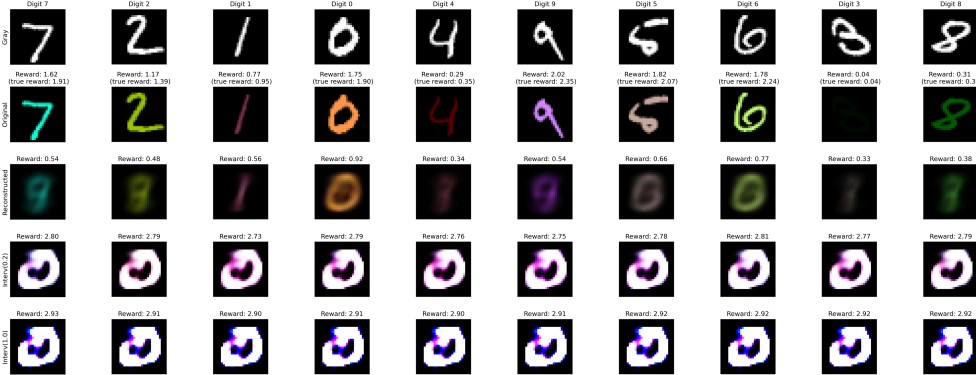

*Figure 20.* **IRAE[1]** on Case 2 DGP for one random seed (random seed 22), with a Dense AutoEncoder, linear HSIC as independence criterion, **latent dimension 3 = number of instruments**, regularization weights $\lambda = \mu_1 = 1$ and $\mu_2 = \mu_3 = 0$ and training for 50 epochs with early stopping (patience 5 epochs).

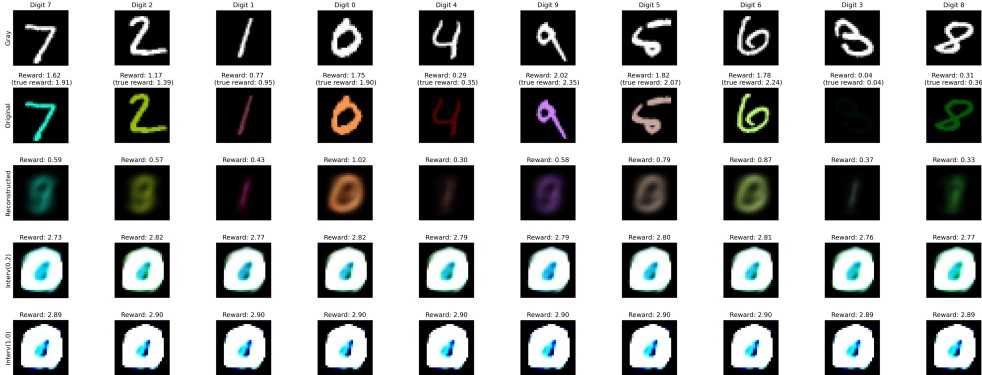

*Figure 21.* **IRAE[0]** on Case 2 DGP for one random seed (random seed 22), with a Dense AutoEncoder, linear HSIC as independence criterion, **latent dimension 3 = number of instruments**, regularization weights $\lambda = \mu_1 = 1$ and $\mu_2 = \mu_3 = 0$ and training for 50 epochs with early stopping (patience 5 epochs).

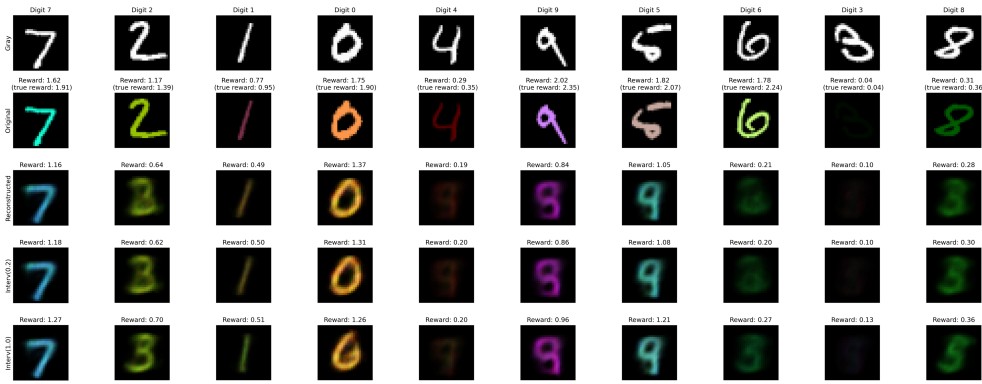

*Figure 22.* **Vanilla AE** on Case 2 DGP for one random seed (random seed 22), with a Dense AutoEncoder, linear HSIC as independence criterion, **latent dimension 3 = number of instruments**, regularization weights $\lambda = \mu_1 = \mu_2 = \mu_3 = 0$ and training for 50 epochs with early stopping (patience 5 epochs).

## J.2. Case 3: Confounded Outcome

We examine the following confounded outcome generating process, where the instruments now affect the colors in a more convoluted intertwined manner. We denote this as *Case 3 DGP*.

All of models are trained with 60k training samples and evaluated on 10k test set, for 40 random seeds. Regularization weights are 0 or 1. All models are trained with 50 epochs after initialization with early stopping of patience 5.

---

**Case 3 DGP**

Draw DGP parameters $\alpha, \beta \sim \text{Unif}(0.1, 0.7)$. Then generate samples as:

$$G_i \in [0, 1]^{28 \times 28} \qquad \text{(grayscale MNIST image)}$$

$$Z_i, \ U_i \sim \mathcal{N}(0, I_3), \qquad Z_i \perp\!\!\!\perp U_i \qquad \text{(instrument \& confounder)}$$

$$r_i = \text{clip}(0.5 + \alpha \, Z_{i1} + \beta \, U_{i1}, \, 0, \, 1) \qquad \text{(red channel)}$$

$$g_i = \text{clip}(0.5 + \alpha \, Z_{i2} + \beta \, U_{i2}, \, 0, \, 1) \qquad \text{(green channel)}$$

$$b_i = \text{clip}(0.5 + \alpha \, Z_{i3} + \beta \, U_{i3}, \, 0, \, 1) \qquad \text{(blue channel)}$$

$$X_i(k, \ell, c) = G_i(k, \ell) \cdot (r_i, g_i, b_i)_c, \qquad \begin{matrix} c \in \{R, G, B\}, \\ (k, \ell) \in \{1, \dots, 28\}^2 \end{matrix} \qquad \text{(colour image)}$$

$$Y_i = r_i + g_i + b_i - U_{i1} - U_{i2} - U_{i3}. \qquad \text{(confounded outcome)}$$

Returns the tuples $(Z_i, X_i, Y_i)$.

---

In this confounding setting, we found that IRAE[0], IRAE[1], IRAE[2], IRAE still led to improved outcome, whereas Vanilla AE did not.

| Arch | Latent Dim | Reg Type | Warm Start | image | Vanilla AE | IRAE[0] | IRAE[1] | IRAE[2] | IRAE |
|---|---|---|---|---|---|---|---|---|---|
| dense | 10 | linear | False | reconstructed | -0.46 (0.02) | -0.67 (0.02) | -0.67 (0.02) | **-0.27 (0.01)** | **-0.27 (0.02)** |
| | | | | intervened(0.2) | -0.45 (0.02) | **1.4 (0.12)** | **1.4 (0.1)** | 1.38 (0.15) | 1.36 (0.15) |
| | | | | intervened(1.0) | -0.37 (0.03) | 1.54 (0.11) | 1.54 (0.09) | 1.57 (0.12) | **1.58 (0.08)** |
| | 32 | linear | False | reconstructed | -0.46 (0.02) | -0.67 (0.02) | -0.67 (0.02) | **-0.14 (0.02)** | **-0.14 (0.02)** |
| | | | | intervened(0.2) | -0.45 (0.02) | **1.4 (0.12)** | **1.4 (0.1)** | 0.74 (0.34) | 0.63 (0.35) |
| | | | | intervened(1.0) | -0.37 (0.03) | **1.54 (0.11)** | **1.54 (0.09)** | 1.42 (0.32) | 1.34 (0.39) |
| conv | 10 | linear | False | reconstructed | -0.37 (0.02) | -0.6 (0.06) | -0.6 (0.05) | **-0.21 (0.03)** | -0.22 (0.03) |
| | | | | intervened(0.2) | -0.36 (0.03) | 0.21 (0.34) | 0.4 (0.4) | **0.98 (0.23)** | 0.83 (0.41) |
| | | | | intervened(1.0) | -0.31 (0.07) | 0.4 (0.56) | 0.69 (0.58) | 1.25 (0.54) | **1.28 (0.51)** |
| | 32 | linear | False | reconstructed | -0.37 (0.02) | -0.6 (0.06) | -0.6 (0.05) | **-0.1 (0.03)** | **-0.1 (0.03)** |
| | | | | intervened(0.2) | -0.36 (0.03) | 0.21 (0.34) | 0.4 (0.4) | **0.7 (0.33)** | 0.64 (0.38) |
| | | | | intervened(1.0) | -0.31 (0.07) | 0.4 (0.56) | 0.69 (0.58) | **1.26 (0.39)** | 1.11 (0.42) |

*Figure 23.* Experimental results for the **Case 3** data generating process. Mean improvement and standard deviation of improvement is reported.

## J.3. Case 4: Confounded DGP with One Outcome Relevant Dimension

We examine the following confounded outcome generating process, where the instruments now affect the colors in a more convoluted intertwined manner. Moreover, only the red channel is relevant for the outcome and the outcome is confounded. We denote this as *Case 4 DGP*.

All of models are trained with 60k training samples and evaluated on 10k test set, for 40 random seeds. Regularization weights are 0 or 1. All models are trained with 50 epochs after initialization with early stopping of patience 5.

| Arch | Latent Dim | Reg Type | Warm Start | image | Vanilla AE | IRAE[0] | IRAE[1] | IRAE[2] | IRAE |
|---|---|---|---|---|---|---|---|---|---|
| dense | 10 | linear | False | reconstructed | -0.16 (0.01) | -0.22 (0.01) | -0.22 (0.01) | **-0.09 (0.01)** | **-0.09 (0.01)** |
| | | | | intervened(0.2) | -0.15 (0.01) | **0.51 (0.03)** | 0.5 (0.03) | **0.51 (0.02)** | **0.51 (0.02)** |
| | | | | intervened(1.0) | -0.1 (0.02) | **0.55 (0.01)** | **0.55 (0.02)** | **0.55 (0.01)** | **0.55 (0.01)** |
| | 32 | linear | False | reconstructed | -0.16 (0.01) | -0.22 (0.01) | -0.22 (0.01) | **-0.05 (0.01)** | **-0.05 (0.01)** |
| | | | | intervened(0.2) | -0.15 (0.01) | **0.51 (0.03)** | 0.5 (0.03) | 0.5 (0.01) | 0.49 (0.03) |
| | | | | intervened(1.0) | -0.1 (0.02) | **0.55 (0.01)** | **0.55 (0.02)** | 0.54 (0.01) | 0.54 (0.01) |
| conv | 10 | linear | False | reconstructed | -0.13 (0.01) | -0.2 (0.02) | -0.2 (0.02) | **-0.07 (0.03)** | **-0.07 (0.03)** |
| | | | | intervened(0.2) | -0.13 (0.01) | 0.26 (0.12) | 0.28 (0.14) | **0.45 (0.03)** | 0.44 (0.05) |
| | | | | intervened(1.0) | -0.11 (0.04) | 0.42 (0.19) | 0.44 (0.21) | **0.54 (0.01)** | 0.53 (0.02) |
| | 32 | linear | False | reconstructed | -0.13 (0.01) | -0.2 (0.02) | -0.2 (0.02) | **-0.04 (0.03)** | **-0.04 (0.02)** |
| | | | | intervened(0.2) | -0.13 (0.01) | 0.26 (0.12) | 0.28 (0.14) | **0.45 (0.04)** | **0.45 (0.05)** |
| | | | | intervened(1.0) | -0.11 (0.04) | 0.42 (0.19) | 0.44 (0.21) | **0.53 (0.01)** | 0.51 (0.08) |

*Figure 24.* Experimental results for the **Case 4** data generating process. Mean improvement and standard deviation of improvement is reported.

---

**Case 4 DGP**

Draw DGP parameters $\alpha, \beta \sim \mathrm{Unif}(0.1, 0.7)$. Then generate samples as:

$$G_i \in [0,1]^{28 \times 28} \qquad \text{(grayscale MNIST image)}$$

$$Z_i,\ U_i \sim \mathcal{N}(0, I_3), \qquad Z_i \perp\!\!\!\perp U_i \qquad \text{(instrument \& confounder)}$$

$$r_i = \mathrm{clip}\big(0.5 + \alpha\,(Z_{i1} - Z_{i2}) + \beta\,U_{i1},\ 0,\ 1\big) \qquad \text{(red channel)}$$

$$g_i = \mathrm{clip}\big(0.5 + \alpha\,(Z_{i2} - Z_{i3}) + \beta\,U_{i2},\ 0,\ 1\big) \qquad \text{(green channel)}$$

$$b_i = \mathrm{clip}\big(0.5 + \alpha\,(Z_{i3} - Z_{i1}) + \beta\,U_{i3},\ 0,\ 1\big) \qquad \text{(blue channel)}$$

$$X_i(k, \ell, c) = G_i(k, \ell) \cdot (r_i, g_i, b_i)_c, \qquad \begin{aligned} &c \in \{R, G, B\}, \\ &(k, \ell) \in \{1, \dots, 28\}^2 \end{aligned} \qquad \text{(colour image)}$$

$$Y_i = r_i - U_{i1}. \qquad \text{(confounded outcome)}$$

Returns the tuples $\big(Z_i, X_i, Y_i\big)$.

---

We demonstrate in this data generating process the importance of running an instrumental variable regression in the latent space. We see below that if instead we had run OLS regressing the outcome on the identified latent factors, then the direction would be erroneous and the interventional images will not be moving the image towards more red colors.

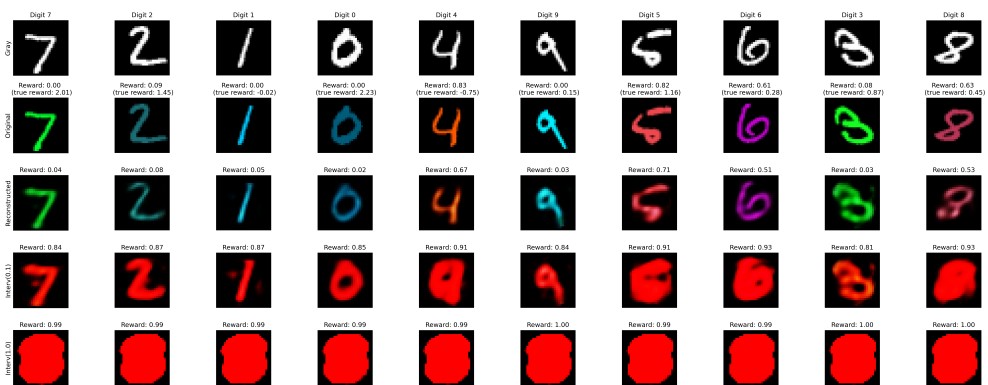

*Figure 25.* **IRAE** on **Case 4 DGP** for one random seed, with a Dense AutoEncoder, linear HSIC as independence criterion, **latent dimension 32**, regularization weights $\lambda = \mu_1 = \mu_2 = \mu_3 = 1$ and training for 50 epochs with early stopping (patience 5 epochs) from scratch (no warm start from IRAE1). Interventional images are intervened in the **direction identified by 2SLS** in the latent space with instrument $Z$, treatment $D$ and outcome $Y$. The outcome is larger when the color of the image is changed to red.

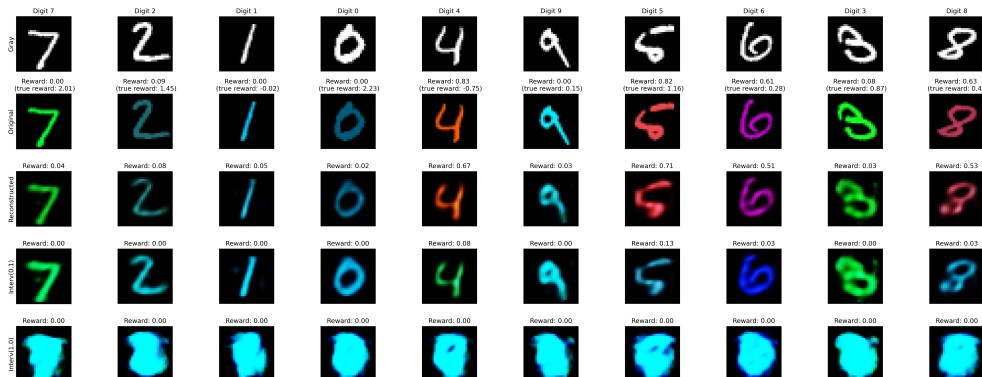

*Figure 26.* **IRAE** on **Case 4 DGP** for one random seed, with a Dense AutoEncoder, linear HSIC as independence criterion, **latent dimension 32**, regularization weights $\lambda = \mu_1 = \mu_2 = \mu_3 = 1$ and training for 50 epochs with early stopping (patience 5 epochs) from scratch (no warm start from IRAE[1]). Interventional images are intervened in the **direction identified by OLS**($Y \sim D$) in the latent space. The outcome is larger when the color of the image is changed to red.

