# OpenReview forum: "Learning Treatment Representations for Downstream Instrumental Variable Regression"
_ICML.cc/2026/Conference — ICML 2026 regular_

### Official Review · Reviewer_rJ5c · 2026-03-11

**Soundness:** 3
**Presentation:** 3
**Significance:** 3
**Originality:** 2
**Overall Recommendation:** 5
**Confidence:** 3

**Summary:**

This paper presents a method for learning treatment representations which un-confound downstream instrumental variable (IV) regression. Traditional IV regression requires an instrument for each endogenous component. However, instruments are generally too low dimensional for this. The authors assume a structural causal model where the observed data and outcome depend on latent variables and learn a decomposition of observed features (treatment) into these latent variables to learn to perturb the outcome. Ability to perturb the outcome is demonstrated via experiments.

**Compliance With Llm Reviewing Policy:**

Affirmed.

**Final Justification:**

The rebuttal addressed my questions and concerns. I think this will be an interesting contribution to the community.

**Key Questions For Authors:**

- How does the method do at recovering treatment effects?
- Are the recovered perturbations reasonable?
- Why does D appear twice in Figure 2?
- Is the method overall to be called IGRL or LIRR or IRAE?

**Limitations:**

Yes

**Strengths And Weaknesses:**

**Strengths**
- The exposition throughout section 1 is very clear. Assumptions are well laid out throughout. (presentation, soundness)
- The method is very interesting. Algorithms are well detailed. Theory supports that which is not assumed. (soundness, originality)

**Weaknesses**
- It could be helpful to add V to Figure 1. (presentation)
- Nomenclature is inconsistent (IGRL, LIRR, vs. IRAE) and exposition connecting the theory and algorithms is at times challenging to follow due to the addition of new variables. (presentation)
- Figure 2’s D vs. tilde D difference is not clear. (presentation)
- It seems necessary to demonstrate recovery of theta, i.e. treatment effects in the SCM to showcase the utility of IV regression with these methods. (soundness, significance)
- Outcome perturbation is not a completely motivated showcase for the ability of the method to perturb the data to make treatments exogenous. It would be helpful to better explain the specific utility of having such a model, why this requires learning un-confounded treatment representation, etc, even just a more complete explanation of the example in Section 2. (significance, presentation)
- The learned perturbations are not shown to be reasonable or actionable for any experiments. (soundness, significance)
- Some IV regression downstream methods (DeepIV) do not need all variables to be exogenous. (significance)

---

> ### Author Rebuttal · Authors · 2026-03-31
>
> We thank the reviewer for the constructive feedback and for recognizing the clarity of our assumptions and algorithm. We provide the following clarifications regarding our objectives and the utility of the learned representations:
>
>
> **1. Recovery of Treatment Effects ($\theta$) vs. Causal Direction Discovery**
>
> The reviewer asks about the recovery of treatment effect $\theta$. We clarify that in high-dimensional settings with sparse instruments, point-identification of individual coefficients for every raw feature is often ill-posed. In our framework, a "bundle" of treatments is collapsed into a single latent dimension $D$, meaning a single causal estimate is assigned to an entire axis of variation. Therefore, our goal is more aligned with causal direction discovery. We identify the latent treatment axes that are both instrument-relevant and causally linked to the outcome. These causal directions can either identify immediate paths for policy improvement or, in more conservative settings, provide a principled basis for future randomized experiments by focusing resources on the most likely causal drivers.
>
>
> **2. Practical Utility and Actionability of Perturbations**
>
> The reviewer questions if the recovered perturbations are reasonable. Because our architecture utilizes a decoder to map the latent $D$ back to the original input space, any counterfactual perturbation is physically grounded and actionable. For example, in our pneumonia case study, a shift in $D$ translates back into the original feature space $X$ as a coordinated change (e.g., a $1$ indicating the addition of a treatment or a specific dosage increase, while $0$ indicating the opposite).
>
> The reliability of these perturbations depends on the model's ability to accurately reconstruct $X$. While downstream IV handles confounding, our latent partitioning ($D$ and $V$) is still essential: it allows the IRAE to learn an invertible mapping between the instrument-informed representation $D$ and the high-dimensional observed features $X$. Without the nuisance latent $V$, the model would be forced to compress all variation into learned $D$, likely violating the exclusion restriction or failing to reconstruct $X$ accurately. As seen in our Colored-MNIST experiments (both the primary experiment and the ablation in Appendix J.2), increasing the dimensions of the learned latent $V$ significantly improves digit reconstruction.
>
>
> **3. Comparison with DeepIV**
>
> The reviewer notes that some methods do not require all variables to be exogenous. We agree and have included an extension in Appendix D to incorporate observed exogenous variables.
>
>
> **4. Clarification on Nomenclature and Figures**
>
> We use IGRL (Instrument-Guided Representation Learning) as the umbrella term for the framework, while LIRR (linear) and IRAE (nonlinear) are the specific model implementations we proposed. $D$ represents the "true" latent factor in the SCM, while $\tilde{D}$ is our learned representation. We will add caption to this visualization to clarify the mapping. We will also incorporate the exogenous factor $V$ into Figure 1 to better illustrate the latent decomposition.

---

> > ### Author Rebuttal · Reviewer_rJ5c · 2026-04-02
> >
> > This is an interesting contribution to IV learning. The rebuttal addresses my confusions and concerns. I am happy to raise my score.

---

> > > ### Author Response · Authors · 2026-04-07
> > >
> > > We appreciate the reviewer's feedback. We would be happy to provide additional details or address any further questions you may have.

---

### Official Review · Reviewer_uD78 · 2026-03-13

**Soundness:** 2
**Presentation:** 3
**Significance:** 2
**Originality:** 2
**Overall Recommendation:** 2
**Confidence:** 2

**Summary:**

The paper focuses on learning low-dimensional treatment representations that preserve instrument-relevant variation for downstream IV analysis in settings with high-dimensional, structured treatments and unobserved confounding. The central argument is that learning a treatment representation without using the instrument can discard instrument-relevant variation in X, creating omitted-treatment bias and invalidating downstream IV analysis. To address this, the paper proposes Instrument-Guided Representation Learning (IGRL) in a linear method (LIRR) and a nonlinear autoencoder-based extension (IRAE).

**Compliance With Llm Reviewing Policy:**

Affirmed.

**Key Questions For Authors:**

1. Can the authors provide empirical evidence that unsupervised representation learning frequently removes instrument-relevant treatment variation in real-world datasets?
2. How sensitive are the theoretical guarantees to violations of the independence assumptions among latent variables?

**Limitations:**

no, but I don't know how to improve.

**Strengths And Weaknesses:**

Strengths:
The paper provides both a linear and nonlinear formulation, which helps readers understand the idea in a simpler setting before moving to the more complex representation-learning framework.

Weaknesses:
1. The practical importance of the problem is somewhat unclear. While it is possible that unsupervised compression removes instrument-relevant information, the paper does not convincingly demonstrate how often this occurs in real applications or why existing flexible IV methods cannot be applied directly to the original treatment variables.
2. The framing around unobserved confounding is somewhat confusing. IV methods are typically introduced precisely because unobserved confounding exists; therefore, discussing "handling unobserved confounding in IV settings" risks sounding circular unless the focus is clearly on how representation learning may interfere with IV identification.

---

> ### Author Rebuttal · Authors · 2026-03-31
>
> We thank the reviewer for the assessment. We appreciate the opportunity to clarify the necessity of our approach and the nuances of unobserved confounding in high-dimensional representation learning.
>
> **1. Unsupervised Compression Failures**
>
> The reviewer asks for evidence that unsupervised methods frequently discard instrument-relevant variation. Unsupervised methods (e.g. PCA, Vanilla AE) are designed to capture directions of maximum variance. However, in high-dimensional real-world data, the nuisance noise or confounding signal ($V$ in our notation) often has higher variance than the instrument-responsive causal signal ($D$). Our benchmarks in Appendices G and H demonstrate this failure explicitly. Among the tested scenarios, when noise follows a non-isotropic Gaussian distribution, a common case in real-world datasets, unsupervised methods prioritize high-variance dimensions. This effectively removes the treatment variation related to instruments required for IV identification, and hence leading to minimal downstream outcome improvements. Our instrument-guided objective encourages the model to learn lower-variance but causally-essential representations.
>
>
> **2. Clarification on Unobserved Confounding and Circularity**
>
> Our focus is not only using IV for confounding, but specifically how representation learning can interfere with IV identification. In high-dimensional $X$, features are often a mixture of instrument-relevant components ($D$) and exogenous features ($V$). If a representation learning step inadvertently bundles $V$ into the latent space while ignoring $D$ (which contains the variation correlated with $Z$), it can create a direct path from the instrument to the outcome. This omitted variable bias (unobserved treatment) violates the exclusion restriction, invalidating the downstream IV analysis (examples included in Appendix G and H). We explicitly partition the latent space into $D$ and $V$ using our IRAE loss to avoid such violation.
>
> **3. Comparison to Flexible IV Applied Directly to Raw Treatment $X$**
>
> Methods such as DeepIV, when applied directly to high-dimensional raw features (e.g., dense clinical logs), reduce to direct treatment assignments or perturbations on raw high-dimensional $X$. In these settings, we typically have insufficient data support for such granular interventions; changing a single clinical value in isolation often results in off-manifold points. By mapping to a lower-dimensional latent $D$, we identify the common groups in $X$ effectively spanned by the instrument. This ensures our incremental intervention stay within the data-supported manifold, making the results both statistically robust and actionable for practitioners. While a domain expert might perform such grouping manually in simple cases, our method provides an automated, data-driven approach for complex high-dimensional signals (e.g., image pixels, radio frequencies, text descriptions) where such groupings are not obvious.
>
> **4. Sensitivity to Independence Assumptions**
>
> The joint independence assumption ($U \perp V \perp Z$) is a strong condition for full latent recovery. However, as discussed in Appendix D, this can be relaxed to conditional joint independence given observed covariates. This relaxation provides a practical path for complex datasets where dependencies between latent components can be explained by observable factors.

---

> > ### Author Rebuttal · Reviewer_uD78 · 2026-04-06
> >
> > Thanks for the clarification. The rebuttal addresses my concerns. However, I still feel this work is very difficult to realize in applications.
> > I am happy to raise my score to 3.

---

> > > ### Author Response · Authors · 2026-04-07
> > >
> > > We thank the reviewer for the score increase and the constructive dialogue. Regarding the practical realization of this work, we would like to clarify that our framework was specifically inspired by the complexities of real-world clinical pathways (a simplified example is discussed in Section 2).
> > >
> > > In a typical hospital setting, the treatment $X$ is high-dimensional, consisting of thousands of specific billing items—e.g., a 0.5mg pill vs. a 5% IV solution, or specific surgical supplies like gauze and scalpels. Conversely, the available instruments $Z$ (e.g., Physician ID or time of visit) are sparse. Standard IV methods struggle to identify the individual causal effect of a single "0.5mg pill" in isolation, as such granular perturbations are often statistically ill-posed (requires large sample size and overlap) or medically uninformative (e.g., when a specific medication only occurs in tandem with a specific procedure).
> > >
> > > Our method is designed for exactly this scenario: it identifies a low-dimensional, instrument-driven latent $D$—such as "Aggressive Surgical" vs. "Conservative Oral Medication"—from this high-dimensional noise. This allows clinicians to identify actionable causal gradients for patient outcomes, such as survival or length of stay.
> > >
> > > To demonstrate this feasibility, we included a simulation of hospital data in our experimental section (Section 5) modeled after these exact constraints. Furthermore, we are currently applying this method to records from a major research hospital, collaborating with clinicians to interpret the learned latent directions and identify optimal recovery pathways. This ongoing real-world application underscores that our approach is not only theoretically sound but practically necessary.

---

### Official Review · Reviewer_Mxgo · 2026-03-13

**Soundness:** 3
**Presentation:** 3
**Significance:** 3
**Originality:** 3
**Overall Recommendation:** 4
**Confidence:** 3

**Summary:**

This paper studies causal inference with high-dimensional, confounded treatments by proposing a framework for learning low-dimensional treatment representations suitable for downstream instrumental variable regression. The key idea is to use the instrument $Z$ to guide representation learning, so that the learned latent factors capture the instrument-relevant components of the treatment while avoiding the omitted-treatment-bias issues of naive unsupervised dimension reduction. The paper proposes a linear method (LIRR), based on regression and singular value decomposition, and a nonlinear method (IRAE), based on an instrument-regularized autoencoder with independence-promoting regularizers. Under a set of structural assumptions, the authors show that the learned representations recover the latent treatment factors up to an invertible linear transformation, which in turn supports the identification of outcome-improving intervention directions via downstream IV analysis. Experiments on a semi-synthetic hospital dataset and a nonlinear MNIST-based benchmark suggest advantages over standard dimension reduction and unregularized autoencoding baselines.

**Compliance With Llm Reviewing Policy:**

Affirmed.

**Final Justification:**

Authors have addressed my concerns.

**Key Questions For Authors:**

1. Regarding hyperparameter selection in IRAE: The proposed loss function introduces multiple regularization weights (λ,μ1,μ2,μ3) that balance reconstruction fidelity against various independence constraints. Could the authors provide practical guidance or a systematic methodology for tuning these hyperparameters in real-world applications, particularly when domain knowledge about the strength of the instrument is limited?
2. Regarding experimental comparisons and robustness: The linear simulation study (Appendix G) compares LIRR only against PCA. Would the authors consider extending this comparison to include other supervised dimensionality reduction methods, such as Partial Least Squares (PLS) to better isolate the value of using Z as the supervisory signal? Additionally, given that DeepIV was evaluated on the MNIST benchmark, it would strengthen the semi-synthetic medical experiment to include a comparison with DeepIV applied directly to the high-dimensional treatment X.
3. Regarding the dimensionality assumption in the MNIST experiments: In the Case 1 DGP, the true latent factor D is 3-dimensional (RGB), but the instrument Z is only 2-dimensional. This appears to violate the theoretical assumption that the matrix A must have full row rank (i.e., dim⁡(Z)≥dim⁡(D)). Could the authors discuss whether this violation affects the interpretability of the results, and whether the strong performance in this setting should be interpreted as evidence of robustness beyond the theoretical guarantees or as a limitation of the experimental design?

**Limitations:**

yes

**Strengths And Weaknesses:**

## Strengths

**Strength:**
1. The paper studies an important problem in causal inference: how to perform IV analysis when the treatment is high-dimensional and confounded. The motivation for learning instrument-aware treatment representations, rather than relying on naive unsupervised dimension reduction, is clear and well justified.

2. The proposed framework is conceptually appealing. By separating the treatment into an instrument-relevant latent component and an instrument-independent residual component, it reduces the downstream IV analysis to a lower-dimensional latent space. The accompanying theoretical results, including identifiability up to a linear transformation under structural assumptions, are a meaningful strength.

3. The nonlinear IRAE method incorporates IV-motivated statistical constraints directly into the representation learning objective through regularization terms such as HSIC-based penalties. This provides a principled connection between the causal assumptions and the learned representation, going beyond purely black-box dimensionality reduction approaches.

## Weaknesses

1. The nonlinear IRAE objective is fairly complex, combining reconstruction with multiple regularization terms weighted by several hyperparameters (e.g., $\lambda, \mu_1, \mu_2, \mu_3$). This creates a nontrivial optimization and model-selection problem, and the paper does not yet provide a clear ablation or tuning guideline showing how robust the method is to these design choices. The choice of independence statistic is itself another sensitive component: the paper mainly uses linear-kernel HSIC for scalability, while also noting that more expressive kernels such as RBF-HSIC did not improve performance and were harder to train.

2. The main training recipe for IRAE[2] and IRAE relies on warm-start initialization from a simpler IRAE[1] model. This suggests that optimization may be sensitive to initialization and training strategy, but the paper does not clearly quantify how much performance degrades without warm-starting or how robust the method is across different initialization schemes. This increases the practical complexity of applying the method.

---

> ### Author Rebuttal · Authors · 2026-03-31
>
> We thank the reviewer for their positive assessment of our work, particularly for noting the conceptual appeal and the connection between our causal assumptions and the IRAE objective. We address your specific questions regarding optimization, baselines, and dimensionality below:
>
> **1. Hyperparameter Selection and Optimization Guidelines**
>
> We acknowledge that tuning multiple loss terms is challenging. In practice, we follow a magnitude balancing heuristic: weights are adjusted so that the reconstruction loss and independence penalties maintain similar magnitudes during the early training phase. This ensures that the instrument-guided constraints are active without overwhelming the latent manifold formation. While we used linear-kernel HSIC in the experiment presented in the main paper, we also tested pairwise HSIC in Appendix J.2, which also yielded notable outcome improvement compared to unregularized baselines. For practitioners, IRAE is designed to be modular; alternative statistics can be substituted based on domain knowledge of expected dependencies.
>
> **2. Robustness to Initialization (Warm-starting)**
>
> Contrary to the suggestion that warm-starting is a strict requirement for stability, our ablation study in Appendix J.2 demonstrates that for dense autoencoder architectures, training without a warm-start performs equally well, and in some cases better, than the staged approach. While warm-starting can be a useful stabilizer in highly complex architectures (e.g. convolution layers), it is not a prerequisite for the method’s success, which reduces the practical complexity for the user.
>
> **3. Comparison with Partial Least Squares (PLS) and DeepIV**
>
> We extended our linear study to include PLS. In our pneumonia benchmark, PLS is competitive since treatment noise $V$ aligns with the outcome $Y$ directions. We additionally modified our linear benchmarks (Appendix G) to create an opposing covariance structure, where outcome is subtracted by sum of $V$. Under these more challenging, yet common, real-world conditions, LIRR consistently outperforms PLS, as PLS is susceptible to the confounding between treatment and outcome that our instrument-guided approach specifically filters out.
>
> Result of Linear Benchmark (modified Appendix G, of dimension $m=50$)
> | Method | DGP 1 | DGP 2 | DGP 3 |
> |---|---|---|---|
> | **LIRR** | **1.06±2.61** | **2.59±3.51** | **2.42±3.87** |
> | PCA | 0.78±3.91 | 1.40±3.20 | 1.19±4.12 |
> | PLS | 0.46±0.60 | 0.75±1.58 | 1.56±1.91 |
>
> Regarding DeepIV on the pneumonia experiment: since the underlying data-generating process is linear, a non-parametric model such as DeepIV likely introduces unnecessary estimation variance and training complexity. Our initial run (shown below) shows only marginal improvement. While these figures reflect limited hyperparameter tuning during the rebuttal period, we expect that further optimization will yield minimal gain for above reasoning.
>
> Result of Pneumonia Benchmark
> | Model Type | Saved time (±SD) |
> |---|---|
> | Ours) LIRR | 37.46 ± 10.71 |
> | (NEW) PLS | 36.97 ± 7.81 |
> | (NEW) DeepIV | 0.61 ± 2.18 |
>
> The finalized benchmark results for all aforementioned methods will be included in the revised manuscript.
>
> **4. Dimensionality Assumption in MNIST (Case 1)**
>
> The reviewer is correct that the 2D instrument ($Z$) and 3D latent ($D$) in Case 1 violate the full-row-rank assumption ($dim(Z) \ge dim(D)$). This was a design choice to facilitate 2D visualization of the learned manifold and test the framework's robustness to rank-deficiency. The strong performance in this setting suggests that the method identifies the maximal identifiable subspace of the treatment that is responsive to the instrument. We provide a fully rank-sufficient experiment in Appendix J.2 to confirm that the method performs well when the assumption is fully met.

---

> > ### Author Rebuttal · Reviewer_Mxgo · 2026-04-04
> >
> > Thanks. I maintain my positive score

---

> > > ### Author Response · Authors · 2026-04-07
> > >
> > > We appreciate the reviewer's feedback, which has helped expand our benchmark. We would be happy to provide additional details or address any further questions you may have.

---

### Official Review · Reviewer_9nTy · 2026-03-19

**Soundness:** 3
**Presentation:** 2
**Significance:** 2
**Originality:** 3
**Overall Recommendation:** 4
**Confidence:** 3

**Summary:**

This paper proposes a framework for instrumental variables with highdimensional treatments and unobserved confounding. For this, the authors propose to learn an instrumented guided representation for improved IV regression. The authors provide theoretical analyses of their method and semi-synthetic experiments.

**Compliance With Llm Reviewing Policy:**

Affirmed.

**Final Justification:**

In the rebuttal, the open questions have mostly been addressed. While the concerns regarding practical relevance due to hard assumptions and optimization challenges remain, I still think this work is interesting and I would vote for acceptance of the paper.

**Key Questions For Authors:**

1.	The authors state “Our Pneumonia and MNIST experiments further support this practical applicability, demonstrating that the method is robust even to minor violations.” What are the violations exactly and where can I see the robustness and how can this be interpreted?
2.	What is the main motivation for the soft intervention setting (unlike full point intervention for arbitrary treatments)? Is it to avoid the problem of overlap violations which easily come from the high dimensional treatment space?

**Limitations:**

Not really. I think the strong assumptions and the optimization challenges should be mentioned here at least.

**Strengths And Weaknesses:**

### Strengths:
- Causal inference with highdimensional treatments and unobserved confounding is an interesting and underexplored setting and the relevance is nicely motivated.
- The idea of using instrument guided representations instead of unsupervised dimensionality reduction to avoid omitted variable bias makes sense and is intuitive.
- The paper provides proper theoretical background of their method and gives also empirical insights into the learned representations in the experiments in addition to the benchmarking.

### Weaknesses:
- The main weakness is probably the hard assumptions required for this approach. However, I appreciate that these are properly stated. Nevertheless, robustness against minor violations should be discussed in more detail and maybe also investigated more empirically.
- The loss terms of the IRAE with independences seem hard to optimize and tune for real-world applications where tuning cannot be performed with ground-truth counterfactual evaluation.
- I think the setting and motivation of the incremental/soft intervention could be explained more clearly. I guess the main motivation for this (unlike full point intervention for arbitrary treatments) is to avoid the problem of overlap violations which easily comes from the high dimensional treatment space? Further, since the goal here is to just learn an intervention that improves over the existing policy, this seams more like a policy learning problem than a typical IV/potential outcome/treatment effect setting. While there is nothing wrong with that, I think the current formulation and structure of the paper is a bit confusing to people wit a background in IVs ( also the used formulation in the abstract “identification of optimal outcome-prediction directions” is a bit confusing). Maybe a direct connection to policy learning would be nice.
- The experimental evaluation and presentation here could be improved, e.g., more benchmarking and the plots of the representations are a bit hard to understand.
- The discussion should be extended to provide a short conclusion and limitations.
----
Overall I think the hard  assumptions and the hard-to-optimize IRAE training objective are challenges in real-world applications and may limit the usefulness of the approach drastically. However, I overall like the idea and motivation of the method and the authors provide extensive theoretical background, so this still might be of interest for the research community as a starting point for more robust methods in this setting.

---

> ### Author Rebuttal · Authors · 2026-03-31
>
> We thank the reviewer for the constructive feedback and for recognizing the intuition and theoretical grounding of our instrument-guided representation approach. We address the review below:
>
> **1. Robustness to Minor Violations of Assumptions and Associated Benchmark**
>
> Our experiments specifically test the proposed model's stability under the following assumption violations:
>
> Invertibility (Assump 2.1): In our Colored-MNIST experiments, the mapping is inherently non-perfectly-invertible due to continuous color gradients and background noise. Despite approximating digit color (and hence outcome $Y$) via 2-mean clustering during evaluation, the model successfully recovers a causal representation that leads to significant downstream improvement in the outcome. Similarly, in the Pneumonia dataset, the observed interventions involve thresholding and clinical "clipping" that violates perfect invertibility. Nevertheless, the learned latent captures the relevant variation for downstream IV regression. Additionally, Appendix J shows that even when latent dimensions are intentionally restricted (preventing perfect reconstruction), downstream outcome improvement remains significant.
>
> Row-Rank-Deficiency (Assump 2.2): In Colored-MNIST Experiment 1 (main), we use a 2D instrument ($Z$) for a 3D treatment ($D$, RGB). Despite this theoretical violation of the full-row-rank condition, the model identifies the 2D subspace of the treatment that is responsive to the instrument and still outperforms unsupervised baselines.
>
> Lastly, we included a discussion in Appendix D on relaxing joint independence ($U \perp V \perp Z$) through conditioning on observable covariates $W$. This conditional joint independence ($U \perp Z \perp V \mid W$) offers a practical path for complex real-world datasets, and only requires modifying the independence regularization in IRAE loss to conditional independence regularization.
>
> **2. Practical Optimization and Tuning (IRAE)**
>
> We acknowledge that tuning multiple loss terms is challenging, both with or without ground-truth counterfactuals. In practice, we follow a magnitude balancing heuristic: weights are adjusted so that the reconstruction loss and independence penalties maintain similar magnitudes during the early training phase. This ensures that the instrument-guided constraints are active without overwhelming the latent manifold formation. Furthermore, our framework maintains a level of statistical verifiability similar to classical methods; practitioners can still compute standard diagnostics, such as the first-stage $F$-statistic, on the learned latent representations to verify the strength and relevance of the instrument.
>
> **3. Motivation for Soft Interventions and Connection to Policy Learning, Overlap**
>
> The reviewer is correct that this framework is naturally suited for policy learning, specifically in identifying "directions of improvement" rather than static effects. In high-dimensional spaces, "point interventions" (setting $X$ to an arbitrary $x$) frequently suffer from a lack of overlap/support. By identifying the latent directions ($D$) that are instrument-relevant and moving only small steps away from the current treatment, we focus on incremental interventions within the manifold supported by the data, avoiding the need for extreme and unreliable extrapolation.
>
> **4. Presentation and Benchmarking**
>
> In the revised manuscript, we will expand the description of the experimental section and add explicit pointers to the systematic benchmarks in the Appendix (specifically results from Appendices G, H, and J). We will also add a formal Conclusion and Limitations section summarizing these experimental findings and computational considerations discussed above.

---

> > ### Author Rebuttal · Reviewer_9nTy · 2026-04-03
> >
> > Thank you for the rebuttal. The open questions have mostly been addressed. While the concerns regarding practical relevance due to hard assumptions and optimization challenges remain, I still think this work is interesting and I maintain my borderline positive score.

---

> > > ### Author Response · Authors · 2026-04-07
> > >
> > > We appreciate the reviewer's feedback, which has helped improve the clarity and rigor of our work. We would be happy to provide additional details or address any further questions you may have.

---

### Decision · Program_Chairs · 2026-04-30

**Decision:**

Accept (regular)

**Comment:**

The paper proposed a framework that constructs treatment representations by explicitly incorporating instrumental variables. The studied problem is very important and relevant. Most of the concerns are well addressed during rebuttals, so I would like to recommend accept on this paper.